# Mean Estimation with User-level Privacy under Data Heterogeneity

**Rachel Cummings**[*]
Department of Industrial Engineering and Operations Research
Columbia University
New York, NY 10027
rac2239@columbia.edu

**Vitaly Feldman**
Apple
Cupertino, CA 95014

**Audra McMillan**
Apple
Cupertino, CA 95014
audra.mcmillan@apple.com

**Kunal Talwar**
Apple
Cupertino, CA 95014
ktalwar@apple.com

## Abstract

A key challenge in many modern data analysis tasks is that user data is heterogeneous. Different users may possess vastly different numbers of data points. More importantly, it cannot be assumed that all users sample from the same underlying distribution. This is true, for example in language data, where different speech styles result in data heterogeneity. In this work we propose a simple model of heterogeneous user data that differs in both distribution and quantity of data, and we provide a method for estimating the population-level mean while preserving user-level differential privacy. We demonstrate asymptotic optimality of our estimator and also prove general lower bounds on the error achievable in our problem.

## 1 Introduction

Many practical problems in statistical data analysis and machine learning deal with the setting in which each user generates multiple data points. In such settings the distribution of each user's data may be (somewhat) different and, furthermore, users may possess vastly different numbers of samples. This issue is one the key challenges in federated learning [20] leading to considerable interest in models and algorithms that address the issue.

As an example, consider the task of next-word prediction for a keyboard. Different users typing on a keyboard may have different styles of writing, leading to different distributions. There are aspects of the language that are common to all users, and likely additional aspects of style that are common to large groups of users. Thus while each user has their own data distribution, there are commonalities to all the distributions, and additional commonalities amongst distributions corresponding to subsets of users. Modeling and learning such relationships between users' distributions is crucial for building a better global model, as well as for personalizing models for users.

The focus of this work is on differentially private algorithms for such settings. We assume that there is an unknown global meta-distribution $\mathcal{D}$. For each user $i$, a personal data distribution $\mathcal{D}_i$ is chosen randomly from $\mathcal{D}$ (for example, by sampling a set of parameters that define $\mathcal{D}_i$). Each user then receives some number $k_i$ of i.i.d. samples from $\mathcal{D}_i$. The goal is to solve an analysis task relative to $\mathcal{D}$, with an eye towards better modeling of each $\mathcal{D}_i$ even when $k_i$ is small. This abstract setting

---

[*]Part of this work was completed while the author was at Apple.

36th Conference on Neural Information Processing Systems (NeurIPS 2022).

can model many practical settings where the relationships between the $\mathcal{D}_i$'s take different forms. Indeed the standard loss in federated learning is the unweighted average over users of a per-user loss function [20, Sec. 3.3.2], which corresponds to learning $\mathcal{D}$. Little theoretical work has been done in this setting and even the most basic statistical tasks are poorly understood. Thus we start by focusing on the fundamental problem of mean estimation in this setting. Specifically, in our model, $\mathcal{D}$ is a distribution on the interval $[0, 1]$ with unknown mean $p$ and unknown variance $\sigma_p^2$. Further, we assume that $\mathcal{D}_i$ is simply a Bernoulli distribution with mean $p_i \sim \mathcal{D}$.

While the general $\mathcal{D}_i$ setting is of interest, there are many settings where users generate Boolean signals. For example, each sample from the Bernoulli distribution could represent whether or not the user has clicked on an ad. Another common example is model evaluation, where the user produces a Bernoulli sample by engaging or not engaging with a feature (e.g., phone keyboard next word suggestion, crisis helpline link, search engine knowledge panels, sponsored link in search results, etc.). As a concrete example, a language model is used to make the next word suggestions on a phone keyboard. A new version of this model would be first tested to measure the average suggestion acceptance rate over users. Each user would thus generate a set of independent Bernoulli r.v.'s with each individual mean $p_i$ corresponding to the model accuracy for the user. Heterogeneity comes from different users typing differently (and hence model accuracy varying across users) and using the keyboard with different frequency. Note that the distribution of model accuracies among users is the meta distribution $\mathcal{D}$ in our work. More generally, measuring the average accuracy of a classification model among a large group of users is an important task in itself. Such models are deployed in privacy-sensitive applications such as health and finance. The resulting statistics may need to be shared with third parties or other teams within a company, raising potential user privacy concerns.

Our main contribution is a differentially private algorithm that estimates $p$ and $\sigma_p$ in this setting. We first study this question in an idealized setting with known $\sigma_p$ and no privacy constraints. Here the optimal non-private estimator for $p_i$ is simple and linear: it is a weighted linear combination of the individual user means with weights that depend on the $k_i$'s and on $\sigma_p$. The variance of this estimate is $\sigma_{ideal}^2 \approx (\sum_i \min(k_i, \sigma_p^{-2}))^{-1}$. This expression has a natural interpretation: this is the variance from using $\min(k_i, \sigma_p^{-2})$ samples from user $i$ and averaging all the Bernoulli samples thus obtained. The restriction on using at most $\sigma_p^{-2}$ samples from each user ensures that the estimator is not too affected by their individual mean $p_i$.

Even in the case where it is known that $\sigma_p^2 = 0$, the solution is non-trivial and, to the best of our knowledge, no optimal private algorithm was previously known. In this case, each user samples from the same distribution, but there may be deviations in the number of samples that each user holds. In the absence of privacy constraints, this setting poses no additional complexity over the case where each user has a single data point, since the data points all come from the same distribution. However, with the requirement of user-level differential privacy, additional care needs to be taken to hide *all* samples from any individual user. In this case, we already need to employ many of the technical tools developed in this work, as we show in Appendix C.

We show that under mild assumptions, there is no asymptotic price to privacy (and to not knowing $\sigma_p$). We provide a differentially private estimator for $p$ with variance $O(\sigma_{ideal}^2)$. Interestingly, the estimator achieving this bound in the private setting is non-linear. Further, we show that $\sigma_{ideal}^2$ is near-optimal, under some mild technical conditions.

Our technical results highlight several of the challenges associated with ensuring user-level privacy when data is heterogeneous. For example, the optimal choice of weights for each user contribution itself depends on $p$ and $\sigma_p$ that we are trying to estimate. Further, we show a novel approach to proving lower bounds for private statistical estimation in the heterogenous setting. Our approach builds on the proof of the Cramér-Rao lower bound in statistics, and we show how privacy terms can be incorporated in this approach to show near optimality of our algorithms for nearly every setting of $k_i$'s. These tools and insights should be useful for modeling and designing algorithms for more involved data analysis tasks.

Our work lays the foundation for similar model-driven exploration in other settings. There have been attempts to handle heterogeneity by phrasing the problem as meta-learning or multi-task learning [20, Sec 3.3.3], which implicitly makes some assumptions about the different distributions. Our goal is to start with a more principled approach that makes explicit the assumptions on the relationship between different distributions and use that to derive the algorithm. For example, if were to model the $D_i$'s

as having means coming from a mixture of Gaussians, the estimation of cluster means would be a necessary step in an EM-type algorithm. Our choice of $\mathcal{D}_i$'s being Bernoulli is meant to capture discrete distribution learning problems that have been extensively studied in private federated settings. Our techniques are general and would extend naturally to real-valued settings where, e.g., $\mathcal{D}_i$ is a Gaussian with mean $p_i$ and known variance. While we make minimal assumptions on $\mathcal{D}$, our results asymptotically match the lower bounds for the case of $\mathcal{D}$ being Gaussian with known variance. Our techniques also extend in natural ways to higher dimensions.

Our main results involve two estimators; a non-realisable estimator $\widehat{p}_\epsilon^{\text{ideal}}$ that assumes that the mean and variance of $\mathcal{D}$ are known to the estimator, and a realisable estimator $\widehat{p}_\epsilon^{\text{realistic}}$ that is private with respect to the user's samples, but not with respect to each user's number of samples $k_i$. Let $\widehat{p}_i$ be the mean of the $k_i$ samples from user $i$. The estimator $\widehat{p}_\epsilon^{\text{realistic}}$ requires as input initial, less accurate $(\epsilon, \delta)$-DP mean and variance estimators $\texttt{mean}_{\epsilon,\delta}$ and $\texttt{variance}_{\epsilon,\delta}$. The main results of this paper can be (informally) summarised as follows:

- **Near optimality of $\widehat{p}_\epsilon^{\text{ideal}}$ [Theorem 5.1].** For any parameterized family of distributions $p \mapsto \mathcal{D}_p$, if the Fisher information of $\widehat{p}_i$ is inversely proportional to the variance of $\widehat{p}_i$ for all $i$, and each $\widehat{p}_i$ is sufficiently well concentrated (sub-Gaussian is sufficient) then $\widehat{p}_\epsilon^{\text{ideal}}$ is minimax optimal, up to logarithmic factors, among all unbiased estimators of $p$ in the range $p \in [1/3, 2/3]$. The proof of this result involves a Cramér-Rao style proof which may be of independent interest.

- **Near optimality of $\widehat{p}_\epsilon^{\text{realistic}}$ [Theorem 4.1].** Under mild conditions on the accuracy of $\texttt{mean}_{\epsilon,\delta}$ and $\texttt{variance}_{\epsilon,\delta}$, and assuming the $\max$ and and median $k_i$ are within a factor of $(n\epsilon/\log n) - 1$, then $\text{Var}(\widehat{p}_\epsilon^{\text{realistic}}) = O(\text{Var}(\widehat{p}_\epsilon^{\text{ideal}}))$.

- **Lower bound in terms of $k_i$ [Corollary 5.5].** We give an explicit formula for the minimax optimal error in terms of the sequence $k_1, \cdots, k_n$ and variance $\sigma_p^2$.

Our main algorithmic results require concentration of the meta-distribution $\mathcal{D}$. We note that in practice, this is not an unreasonable assumption. For example, in the case of model evaluation, it may be be reasonable to assume that a general model has similar accuracy for the vast majority of users, or formally, that the model accuracy is well-concentrated.

## 1.1 Related Work

Frequency estimation in the example-level privacy model has been well-studied in the central [12, 11] and local models [18, 14, 7, 1, 2]. Similarly, private mean estimation has been well studied in both central [12, 16] and local models [9, 8, 5] of privacy. These works have focused on providing example-level privacy (rather than user-level) in settings with homogeneous data, i.e., i.i.d. samples.

[23] recently studied the problem of learning discrete distributions in the homogeneous cases (same distribution and same number of samples per user) with user-level differential privacy, and [22] extended such results to other statistical tasks. These works also consider the setting with different number of samples per user although only via a reduction to same number of samples by discarding the data of users that have less than the median number of samples and effectively only using the median number of samples from all the other users. This approach can be asymptotically suboptimal for many natural distributions of $k_i$'s and is also likely to be worse in practice. Previously, [27] showed how to build a (user-level) differentially private recommendation system, and [25] showed how to train a language model with user-level differential privacy.

User-level differential privacy in the context of heterogeneous data distributions has been studied in the constant $k_i$ setting [31]. Much of the complexity in our setting arises from variation in the $k_i$ values, which makes it challenging to maintain user-level privacy while leveraging the additional data points from users with a large number of data points. The challenges to optimization due to data heterogeneity have also been studied; [34, 15], and Eichner et al. [13] study the approach of using different models for different groups from a convex optimization point-of-view. Mathematically, similar issues are addressed in meta-analysis [6, 33], where the heterogeneity comes from different studies instead of different users. The non-private approach of inverse variance weighting that we recap in Section 3 is standard in that context.

## 2 Model and Preliminaries

Let $\mathcal{D}$ be a distribution on $[0,1]$ with (unknown) mean $p$ and variance $\sigma_p^2$. We assume a population of $n \in \mathbb{N}$ users, where each user $i \in [n]$ has a hidden variable $p_i \sim \mathcal{D}$ and $k_i \in \mathbb{N}$ samples $x_i^1, \ldots, x_i^{k_i} \sim_{i.i.d.} Ber(p_i)$. That is, the samples of user $i$ are i.i.d. from a Bernoulli distribution with parameter $p_i$, which we will denote $\mathcal{D}_i = Ber(p_i)$. Assume without loss of generality that individuals are sorted by their $k_i$, so that $k_1 \geq \cdots \geq k_n$. The samples $x_i^j$ and hidden variables $p_i$ of each user are unknown to the analyst, and we start with assuming that the $k_i$'s are publicly known. In Appendix D, we extend our results to the general case where the $k_i$'s are also private.

The analyst's goal is to estimate the population mean $p$ with an estimator of minimum variance in a manner that is differentially private with respect to user data ($p_i$ and $\{x_i^j\}$). Each user provides their own estimate of their $p_i$ to the analyst based on their data $x_i$: $\widehat{p}_i = \frac{1}{k_i} \sum_{j=1}^{k_i} x_i^j$. The analyst can then aggregate these (possibly along with other information) into her estimate of $p$.

Let us first give some intuition for the distribution of these $\widehat{p}_i$. Let $\mathcal{D}(k)$ be the distribution that first samples $p_i \sim \mathcal{D}$, then samples $x_1, \cdots, x_k \sim Ber(p_i)$ and finally outputs $\widehat{p}_i = \frac{1}{k} \sum_{i=1}^k x_i$. The following lemma (proven in Appendix A) shows that the variance of $\widehat{p}_i$ is larger than $\sigma_p^2$ and transitions from $p(1-p)$ to $\sigma_p^2$ as $k$ increases (equivalently as $\widehat{p}_i$ concentrates around $p_i$).

**Lemma 2.1.** *For all distributions $\mathcal{D}$ supported on $[0,1]$ with mean $p$ and variance $\sigma_p^2$, $\sigma_p^2 \leq p(1-p)$. Further, $\mathbb{E}[\mathcal{D}(k)] = p$ and $\mathrm{Var}(\mathcal{D}(k)) = \frac{1}{k}p(1-p) + \left(1 - \frac{1}{k}\right)\sigma_p^2$.*

We assume that $k_i$ and $p_i$ are independent, so the amount of data an individual has is independent of her data distribution. This is crucial for the problem setup: in order for learning from the heterogeneous population to be advantageous, there must a common meta-distribution is shared across all individuals in the population, rather than a meta-distribution only for each fixed $k_i$.

### 2.1 Differential Privacy

Differential privacy (DP) [12] informally limits the inferences that can be made about an individual as a result of computations on a large dataset containing their data. The definition of DP requires a pairwise *neighbouring relation* between datasets, and DP algorithms ensure that differences between all pairs of neighboring datasets should be hidden by the private algorithm.

In our setting where users have multiple data points, we must distinguish between *user-level* and *event-level* DP. The former considers $D$ and $D'$ neighbours if they differ on all data points associated with a single user, whereas the latter considers $D$ and $D'$ neighbours only if they differ on a *single* data point, regardless of the number of data points contributed by that user. Naturally, user-level DP provides substantially stronger privacy guarantees, and is often more challenging to achieve from a technical perspective. In this work, we will provide user-level DP guarantees.

**Definition 2.2** (User-level $(\epsilon, \delta)$-Differential Privacy [12]). *Given $\epsilon \geq 0$, $\delta \in [0,1]$ and a neighbouring relation $\sim$, a randomized mechanism $\mathcal{M} : \mathcal{X}^{k_1} \times \cdots \times \mathcal{X}^{k_n} \to \mathcal{Y}$ is $(\epsilon, \delta)$-differentially private if for all neighboring datasets $D \sim D' \in \mathcal{X}^{k_1} \times \cdots \times \mathcal{X}^{k_n}$, and all events $E \subseteq \mathcal{Y}$, $\Pr[\mathcal{M}(D) \in E] \leq e^\epsilon \cdot \Pr[\mathcal{M}(D') \in E] + \delta$, where the probabilities are taken over the random coins of $\mathcal{M}$. When $\delta = 0$, we refer to this as $\epsilon$-differential privacy.*

One standard tool for achieving $\epsilon$-differential privacy is the *Laplace Mechanism*. For a given function $f$ to be evaluated on a dataset $D$, the Laplace Mechanism first computes $f(D)$ and then adds Laplace noise which depends on the *sensitivity* of $f$, defined for real-valued functions as $\Delta f = \max_{D, D' \text{ neighbors}} |f(D) - f(D')|$. The Laplace Mechanism outputs $\mathcal{M}_L(D, f, \epsilon) = f(D) + Lap(\Delta f/\epsilon)$ and is $(\epsilon, 0)$-DP.

## 3 A Non-Private Estimator

We begin by illustrating the procedure for computing an optimal estimator $\widehat{p}$ in the non-private setting. The general structure of the estimator will be the same in both settings. The analyst will compute the

---
**Algorithm 1** Non-private Heterogeneous Mean Estimation
---
**Input:** number of users $n$, number of samples held by each user $(k_1, \ldots, k_n)$ $(k_i \geq k_{i+1})$, user-level estimates $(\widehat{p}_1, \cdots, \widehat{p}_n)$.

1: **Initial Estimates**
2: $\widehat{p}^{\text{initial}} = \sum_{i=9n/10}^{n} x_i^1$
3: $\widehat{\sigma}_p^2 = \frac{1}{\log n (\log n - 1)} \sum_{i,j \in [\log n]} (\widehat{p}_i - \widehat{p}_j)^2$

4: **Defining weights**
5: **for** $i = \log n$ to $9n/10$ **do**
6:      Compute $\widehat{\sigma}_i^2 = \frac{1}{k_i}(\widehat{p}^{\text{initial}} - (\widehat{p}^{\text{initial}})^2) + (1 - \frac{1}{k_i})\widehat{\sigma}_p^2$.
7:      $\widehat{w}_i = \frac{1/\widehat{\sigma}_i^2}{\sum_{j=\log n}^{9n/10} 1/\widehat{\sigma}_j^2}$

8: **Final Estimate**
9: **return** $\widehat{p}^{\text{realistic}} = \sum_{i=\log n}^{n} \widehat{w}_i \widehat{p}_i$
---

population-level mean estimate $\widehat{p}$ as a weighted linear combination of the user-level estimates $\widehat{p}_i$.[2] The key question is how to derive the weights so that individuals with more reliable estimates (i.e., larger $k_i$) have more influence over the final result.

Let $\sigma_i^2$ be the variance of $\widehat{p}_i$. In an idealized setting where the $\sigma_i^2$ are all known, the analyst can minimize the variance of the estimator by weighting each user's estimate $\widehat{p}_i$ proportionally to the inverse variance of their estimate. The weights are normalised to ensure the estimate is unbiased. This approach yields the following estimator, which is optimal in the non-private setting [17]:

$$\widehat{p}^{\text{ideal}} = \sum_{i=1}^{n} w_i^* \widehat{p}_i \text{ where } w_i^* = \frac{1/\sigma_i^2}{\sum_{j=1}^{n} 1/\sigma_j^2}. \tag{1}$$

In practice, the $\sigma_i^2$s are unknown, so the analyst must rely on estimates to assign weights. Fortunately, the user-level variance $\sigma_i^2$ can be expressed as a function of $k_i$ and the population statistics $p$ and $\sigma_p^2$, as shown in Lemma 2.1:

$$\sigma_i^2 = \frac{1}{k_i}(p - p^2) + (1 - \frac{1}{k_i})\sigma_p^2. \tag{2}$$

Now, $p$ and $\sigma_p^2$ are also unknown but since they are population statistics, we can use simple estimators to obtain initial estimates. These initial statistics can then be used to define the weights, resulting in a refined estimate of the mean $p$. Specifically, as outlined in Algorithm 1, we split users into three groups. The $\log n$ individuals with the most data are used to produce an estimate of $\text{Var}(\mathcal{D}(k_{\log n}))$, which serves as a proxy for $\sigma_p^2$. The $1/10$th of individuals with the least data are used to produce an initial estimate of the mean $p$. The remaining $9n/10 - \log n$ individuals are used to produce the final estimate. We split the individuals into separate groups to ensure the initial estimates and the final estimate are independent so we can easily obtain variance bounds on the final estimate. The specific sizes of the three groups are heuristic, the exact fraction $1/10$ is not necessary. Under some mild conditions on $\mathcal{D}$, and if $n$ is large enough, the error incurred by $\widehat{p}^{\text{realistic}}$ is within a constant factor of the error incurred by the ideal estimator $\widehat{p}^{\text{ideal}}$.[3]

## 4 A Framework for Private Estimators

We now turn to our main result, which is a framework for designing differentially private estimators for the mean $p$ of the meta-distribution $\mathcal{D}$. We discussed in Section 3 the need for initial estimates of $p$ and $\sigma_p^2$ to weight the contributions of the users. In the non-private setting, there are canonical, optimal choices of these estimators; the empirical mean and empirical variance. In the private setting, these choices are not canonical, and different estimators may perform better in different settings. There is a considerable literature exploring various mean and variance estimators for the homogeneous,

---

[2]In the non-private setting, this restriction is without loss of generality since the optimal estimator takes this form. In the private setting this is still near-optimal; see Section 5 for more details.

[3]This can be observed by viewing the non-private setting as a simplified version of the setting studied in Section 5, which proves near-optimality of (truncated) linear estimators for this problem.

single-data-point-per-user setting. As such, we leave the choice of the specific initial mean and variance estimators as parameters of the framework. This allows us to focus on the nuances of the heterogeneous setting, not addressed in prior work. In Appendix F, we give a specific pair of private mean and variance estimators that provably perform well in our framework.

As in the previous section, we will define two estimators: a ideal estimator $\widehat{p}_\epsilon^{\text{ideal}}$ (only implementable if all the $\sigma_i^2$ are known), and a realisable estimator $\widehat{p}_\epsilon^{\text{realistic}}$. The main result in this section (Theorem 4.1) is that under some mild conditions and assuming $n$ is sufficiently large, there exists an $(\epsilon, \delta)$-DP estimator $\widehat{p}_\epsilon^{\text{realistic}}$ (Algorithm 2) such that for some constant $C$, $\text{Var}(\widehat{p}_\epsilon^{\text{realistic}}) \leq C \cdot \text{Var}(\widehat{p}_\epsilon^{\text{ideal}})$.

## 4.1 The Complete Information Private Estimator

As in Section 3, we begin with a discussion of the ideal estimator if the $\sigma_i$ were known. This ideal estimator $\widehat{p}_\epsilon^{\text{ideal}}$ has a similar form to $\widehat{p}^{\text{ideal}}$ with some crucial differences. The first main distinction is that Laplace noise is added to achieve DP, where the standard deviation of the noise must be scaled to the sensitivity of the statistic. A natural solution would be to add noise directly to the non-private estimator $\widehat{p}^{\text{ideal}}$, but the sensitivity of this statistic is too high. In fact, the worst case sensitivity of $\widehat{p}^{\text{ideal}}$ is 1, which would result in the noise that completely masks the signal. Thus, the first change we make is to limit the weight of any individual's contribution by setting $w_i = \frac{\min\{1/\sigma_i^2, T/\sigma_i\}}{\sum_{j=1}^n \min\{1/\sigma_j^2, T/\sigma_j\}}$ for some truncation parameter $T$. Intuitively, the parameter $T$ controls the trade-off between variance of the weighted sum of individual estimates (which is minimized by assigning high weight to low variance estimators) and variance of the noise added for privacy (which is minimized by assigning roughly equal weight to all users).

We make one final modification to lower the sensitivity of the statistic. Inspired by the Gaussian mean estimator of [21], we truncate the individual contributions $\widehat{p}_i$ into a sub-interval of $[0, 1]$. The truncation intervals $[a_i, b_i]$ are chosen to be as small as possible (to reduce the sensitivity and hence the noise added for privacy), while simultaneously ensuring that $\widehat{p}_i \in [a_i, b_i]$ with high probability (to avoid truncating relevant information for the estimation). In order to achieve this, we need a tail bound on the distribution $\mathcal{D}$. To maintain generality for now, we assume there exists a known function $f_\mathcal{D}^k(n, \sigma_p^2, \beta)$ that gives high-probability concentration guarantees of $\widehat{p}_i$ around $p$, and is defined such that $\Pr\left(\forall i, |\widehat{p}_i - p| \leq f_\mathcal{D}^{k_i}(n, \sigma_p^2, \beta)\right) \geq 1 - \beta$. Appendix G presents a more detailed discussion of the structure of these concentration functions and how they may be estimated if they are unknown to the analyst.

We can now describe the full information, or *ideal*, estimator $\widehat{p}_\epsilon^{\text{ideal}}$:

$$\widehat{p}_\epsilon^{\text{ideal}} = \sum_{i=1}^n w_i^* [\widehat{p}_i]_{a_i}^{b_i} + \text{Lap}(\tfrac{\max_i w_i^* |b_i - a_i|}{\epsilon}), \tag{3}$$

where $[\widehat{p}_i]_{a_i}^{b_i}$ denotes the projection of $\widehat{p}_i$ onto the interval $[a_i, b_i]$ and

$$a_i = p - f_\mathcal{D}^{k_i}(n, \sigma_p^2, \beta), \quad b_i = p + f_\mathcal{D}^{k_i}(n, \sigma_p^2, \beta), \quad \text{and} \quad w_i^* = \frac{\min\{1/\sigma_i^2, T^*/\sigma_i\}}{\sum_{j=1}^n \min\{1/\sigma_j^2, T^*/\sigma_j\}}. \tag{4}$$

We would like to choose the truncation parameter $T^*$ to minimise the variance of the resulting estimator:

$$\text{Var}(\widehat{p}_\epsilon^{\text{ideal}}) = \sum_{i=1}^n (w_i^*)^2 \text{Var}([\widehat{p}_i]_{a_i}^{b_i}) + \max_i \frac{(w_i^*)^2 |b_i - a_i|^2}{\epsilon^2}. \tag{5}$$

Although we do not know $\text{Var}([\widehat{p}_i]_{a_i}^{b_i})$ exactly, we do know that $[\widehat{p}_i]_{a_i}^{b_i} = \widehat{p}_i$ with high probability, and thus we can approximate $\text{Var}([\widehat{p}_i]_{a_i}^{b_i})$ with $\sigma_i$. Throughout the remainder of the paper, we will assume that $\beta$ is chosen such that $\frac{1}{2}\sigma_i^2 \leq \text{Var}([\widehat{p}_i]_{a_i}^{b_i})$. Thus, we will approximate the optimal truncation parameter by

$$T^* = \arg\min_T \sum_{i=1}^n (w_i^*)^2 \sigma_i^2 + \max_i \frac{(w_i^*)^2 |b_i - a_i|^2}{\epsilon^2}$$

$$= \arg\min_T \frac{1}{(\sum_{j=1}^n \min\{1/\sigma_j^2, T/\sigma_i\})^2} \left(\sum_{i=1}^n \min\{1/\sigma_i^2, T^2\} + \max_i \frac{\min\{1/\sigma_i^4, T^2/\sigma_i^2\}|b_i - a_i|^2}{\epsilon^2}\right). \tag{6}$$

We'll show in Section 5 that under some conditions on the Fisher information of $\mathcal{D}(k)$, $\widehat{p}_\epsilon^{\text{ideal}}$ is optimal up to logarithmic factors among all private unbiased estimators for heterogeneous mean estimation.

**Example 1.** *As a simple example, suppose that $p \in (\frac{1}{3}, \frac{2}{3})$, $\sigma_p = 1/\sqrt{n}$, and $k_i = \lceil \frac{n}{i} \rceil$. In this case, an asymptotically optimal non-private estimator averages all the $\sum k_i = O(n \log n)$ available samples. It can be shown that this gives us an unbiased estimator with standard deviation $\Theta(\frac{1}{\sqrt{n \log n}})$. A naive sensitivity-based noise addition method will give us privacy error $O(\frac{1}{\varepsilon \log n})$, since the weight of the first user in this average is $\Theta(1/\log n)$. Our truncation-based algorithm will truncate the ith user's contribution to a range of width $\sqrt{\frac{\log n}{k_i}} \approx \sqrt{\frac{i \log n}{n}}$. Applying our algorithm would then give us privacy error $\Theta(\frac{1}{\varepsilon \sqrt{n \log n}})$. In other words, for constant $\varepsilon$, privacy does not have an asymptotic cost. We remark that in this case, any uniform weighted average will incur asymptotically larger standard deviation $\Omega(\frac{1}{\sqrt{n}})$.*

## 4.2 Realizable Private Heterogeneous Mean Estimation

Our goal in this section is to design a realizable estimator $\widehat{p}_\epsilon^{\text{realistic}}$ that is competitive with the ideal estimator $\widehat{p}_\epsilon^{\text{ideal}}$. As in the non-private setting, we divide the individuals into three groups. The first group, consisting of the $n/10$ individuals with the lowest $k_i$ will be used to compute the initial mean estimate $\widehat{p}_\epsilon^{\text{initial}}$. The $\log n$ individuals with the largest $k_i$ will be used to compute the initial variance estimate $\widehat{\sigma}_p^2$. These initial estimates will be plugged into expressions to compute $\widehat{\sigma}_i^2$, $\widehat{a}_i$, and $\widehat{b}_i$ for the remaining individuals $\log n + 1 \leq i \leq 9n/10$. As in the non-private setting, the specific sizes of these groups are heuristic. The important thing is that the size of the first two groups are large enough that the resulting mean and variance estimates are sufficiently accurate, and the last group contains $\Theta(n)$-users whose $k_i$ is above the median.

Since the estimate $\widehat{p}_\epsilon^{\text{initial}}$ used in $\widehat{a}_i$ and $\widehat{b}_i$ may had additional error up to $\alpha$, we shift these estimates by an additive $\alpha$ to account for this error. Next, all of these intermediate estimates and the user-level mean estimates $\widehat{p}_i$ from users $\log n + 1 \leq i \leq 9n/10$ will be used to compute the optimal weight cutoff $\widehat{T}^*$, the optimal weights $\widehat{w}_i^*$ for each user $\log n + 1 \leq i \leq 9n/10$, and finally the estimator $\widehat{p}_\epsilon^{\text{realistic}}$ as a weighted sum of the truncated user-level estimates $[\widehat{p}_i]_{\widehat{a}_i}^{\widehat{b}_i}$ plus Laplace noise. This procedure is presented in full detail in Algorithm 2.

For the remainder of this section, we turn to establishing the accuracy requirements of $\texttt{mean}_{\epsilon,\delta}$ and $\texttt{variance}_{\epsilon,\delta}$ that ensure that the error of $\widehat{p}_\epsilon^{\text{realistic}}$ is within a constant factor of the error of $\widehat{p}_\epsilon^{\text{ideal}}$.

**Theorem 4.1.** *For any $\epsilon > 0$, $\delta \in [0,1]$, Algorithm 2 is $(\epsilon, \delta)$-DP. If,*

- *$\texttt{mean}_{\epsilon,\delta}$ is such that given $n/10$ samples from $\mathcal{D}$, with probability $1 - \beta$ $|p - \widehat{p}_\epsilon^{\text{initial}}| \leq f_\mathcal{D}^{k_i}(n, \sigma_p^2, \beta)$ and $\widehat{p}_\epsilon^{\text{initial}}(1 - \widehat{p}_\epsilon^{\text{initial}}) \in \left[\frac{1}{2}p(1-p), \frac{3}{2}p(1-p)\right]$,*

- *$\texttt{variance}_{\epsilon,\delta}$ is such that given $\log n$ samples from $\mathcal{D}(k)$, with probability $1 - \beta$, $\widehat{\sigma}_p^2 \in [\text{Var}(\mathcal{D}(k)), 8\text{Var}(\mathcal{D}(k))]$,*

- *the $k_i$s are such that $\frac{k_1}{k_{n/2}} \leq \frac{n/2 - \log n}{\log n}$,*

*then with probability $1 - 2\beta$, $\text{Var}(\widehat{p}_\epsilon^{\text{realistic}}) \leq C \cdot \text{Var}(\widehat{p}_\epsilon^{\text{ideal}})$ for some absolute constant $C$.*

A full proof of Theorem 4.1 is given in Appendix B, we present intuition and a proof sketch here. The first two conditions of Theorem 4.1 ensure that the mean and variance estimates are sufficiently accurate to use in the remainder of the algorithm. Notice that the initial estimates do not need to be especially accurate. In fact, provided $p$ is not too close to 0 or 1, the DP mean estimator that simply adds noise to the sample mean achieves the right accuracy (see Lemma F.1 for details). In Appendix F, we also give a DP variance estimator that achieves the desired accuracy guarantee using only $\log n$ samples, under some mild conditions (Lemma F.4). Thus the set of mean and variance estimators that satisfy the accuracy requirements of Theorem 4.1 are non-empty. We note that the constants $1/2$, $3/2$ and $8$ in Theorem 4.1 are not intrinsic; any constant multiplicative factors will suffice. We also note that the specific sizes of the three groups outlined in Algorithm 2 are heuristic and can be varied to ensure that the initial estimator achieves the required accuracy.

---

**Algorithm 2** Private Heterogeneous Mean Estimation

---

**Input:** $(\epsilon, \delta)$-DP mean estimator $\texttt{mean}_{\epsilon,\delta}$, $(\epsilon, \delta)$-DP variance estimator $\texttt{variance}_{\epsilon,\delta}$, number of users $n$, number of samples held by each user $(k_1, \ldots, k_n$ $s.t.$ $k_i \geq k_{i+1})$, user-level estimates $(\widehat{p}_1, \cdots, \widehat{p}_n)$, error guarantee on $\texttt{mean}_{\epsilon,\delta}$ $\alpha > 0$, and desired high probability bound $\beta \in [0, 1]$.

1: **Initial Estimates**
2: $\widehat{p}_\epsilon^{\text{initial}} = \texttt{mean}_{\epsilon,\delta}(\text{x}_{9n/10+1}^1, \cdots, \text{x}_n^1)$
3: $\widehat{\sigma}_p^2 = \texttt{variance}_{\epsilon,\delta}(\widehat{p}_1, \cdots, \widehat{p}_{\log n})$

4: **Defining weights and truncation**
5: **for** $i = \log n + 1$ to $9n/10$ **do**
6:      Compute $\widehat{\sigma}_i^2 = \frac{1}{k_i}(\widehat{p}_\epsilon^{\text{initial}} - (\widehat{p}_\epsilon^{\text{initial}})^2) + (1 - \frac{1}{k_i})\widehat{\sigma}_p^2$.
7:      $\widehat{a}_i = \widehat{p}_\epsilon^{\text{initial}} - \alpha - f_\mathcal{D}^{k_i}(n, \widehat{\sigma_p^2}, \beta)$
8:      $\widehat{b}_i = \widehat{p}_\epsilon^{\text{initial}} + \alpha + f_\mathcal{D}^{k_i}(n, \widehat{\sigma_p^2}, \beta)$
9:
10: $\widehat{T}^* = \arg\min_T \dfrac{(\sum_{i=\log n+1}^{9n/10} \min\{\frac{1}{\widehat{\sigma}_i^2}, T^2\} + \max_{\log n+1 \leq i \leq 9n/10} \frac{\min\{1/\widehat{\sigma}_i^4, T^2/\widehat{\sigma}_i^2\}|\widehat{b}_i - \widehat{a}_i|^2}{\epsilon^2})}{(\sum_{i=\log n+1}^{9n/10} \min\{1/\widehat{\sigma}_j^2, T/\widehat{\sigma}_i\})^2}$
11: **for** $i = \log n + 1$ to $9n/10$ **do**
12:      $\widehat{w}_i^* = \dfrac{\min\{1/\widehat{\sigma}_i^2, \widehat{T}^*/\widehat{\sigma}_i\}}{\sum_{j=\log n+1}^{9n/10} \min\{1/\widehat{\sigma}_j^2, \widehat{T}^*/\widehat{\sigma}_i\}}$

13: **Final Estimate**
14: $\Lambda = \max_{i \in [\log n+1, 9n/10]} \dfrac{\min\{1/\widehat{\sigma}_i^2, \widehat{T}^*/\widehat{\sigma}_i\}|\widehat{b}_i - \widehat{a}_i|}{\sum_{j=\log n+1}^{9n/10} \min\{1/\widehat{\sigma}_j^2, \widehat{T}^*/\widehat{\sigma}_i\}}$
15: Sample $Y \sim \text{Lap}\left(\frac{\Lambda}{\epsilon}\right)$
16: **return** $\widehat{p}_\epsilon^{\text{realistic}} = \sum_{i=\log n+1}^{9n/10} \widehat{w}_i^* [\widehat{p}_i]_{\widehat{a}_i}^{\widehat{b}_i} + Y$

---

The final assumption ensures that the $\log n$ users with the most data can not estimate the mean of meta-distribution alone. Note that up to logarithmic factors, this condition simply requires that the number of data points held by the user with the most data is at most $n$ times the number of data points of the median user. If $n$ is large, then this is unlikely to be a limiting factor.

The main distinction between $\widehat{p}_\epsilon^{\text{ideal}}$ and $\widehat{p}_\epsilon^{\text{realistic}}$ is the use of the output of the estimators $\texttt{mean}_{\epsilon,\delta}$ and $\texttt{variance}_{\epsilon,\delta}$ to estimate $\sigma_i^2$, $a_i$ and $b_i$. Thus, the main component of the proof of Theorem 4.1 is to show that the conditions stated in the theorem are enough to ensure that $\widehat{\sigma}_i^2$, $\widehat{a}_i$ and $\widehat{b}_i$ are sufficiently accurate.

**Lemma 4.2.** *Given $\widehat{p}_\epsilon^{\text{initial}}$, $\widehat{\sigma}_p^2$, and $k_i$, define $\widehat{\sigma}_i^2 = \frac{1}{k_i}\widehat{p}_\epsilon^{\text{initial}}(1 - \widehat{p}_\epsilon^{\text{initial}}) + \frac{k_i-1}{k_i}\widehat{\sigma}_p^2$. Under the conditions of Theorem 4.1, for all $i > \log n$, we have $\widehat{\sigma}_i^2 \in \left[\frac{1}{2}\sigma_i^2, 9.5\sigma_i^2\right]$ and $|\widehat{b}_i - \widehat{a}_i| \leq 4|b_i - a_i|$.*

A detailed proof of Lemma 4.2 is presented in Appendix B. Lemma 4.2 implies that the individual variance estimates used in the weights, and the truncation parameters are accurate up to constant multiplicative factors. The main ingredient left then is to show that using only a subset of the population in the final estimate only affects the performance up to a multiplicative factor. Under the assumption that $\frac{k_{\max}}{k_{\text{med}}} \leq \frac{n/2 - \log n}{\log n}$, where $\sigma_{k_{\max}}^2 = \text{Var}(\widehat{p}_1)$ and $\sigma_{k_{\text{med}}}^2 = \text{Var}(\widehat{p}_{n/2})$ then

$$\sigma_{k_{\text{med}}}^2 = \frac{1}{k_{\text{med}}}p(1-p) + (1 - \frac{1}{k_{\text{med}}})\sigma_p^2 \leq \frac{n/2 - \log n}{\log n}\sigma_{k_{\max}}^2. \tag{7}$$

We use this to show that for any truncation parameter $T$, $\sum_{i=1}^n \min\{\frac{1}{\sigma_i^2}, \frac{T}{\sigma_i}\} \leq 4\sum_{i=\log n+1}^{9n/10} \min\{\frac{1}{\sigma_i^2}, \frac{T}{\sigma_i}\}$. Using this, along with the bounds on estimated quantities from Lemma 4.2, we show that with high probability, the variance of the our estimator $\widehat{p}_\epsilon^{\text{realistic}}$ is within a constant factor of $\text{Var}(\widehat{p}_\epsilon^{\text{ideal}})$, as given in Equation (5).

We remark that this framework is amenable to being performed in a federated manner if one has private federated mean and variance estimators. Steps (6) - (8) and Step (12) can be performed locally. Steps (10) and the final sum in Step (16) would need to be altered to fit the federated framework.

We'll see in Appendix D that it is sufficient to replace Step (10) with an estimate of $\frac{1}{\sigma_{\log n}}$ (the inverse standard deviation of the user with the $\log n$-th most data). The final step is then a simple addition with output perturbation, which can be performed in a federated manner (e.g., [24, 20]).

In Appendix D, we extend this result to the case where $k_i$s are private and unknown to the analyst (Algorithm 3, Theorem D.1). We'll need considerably more machinery in this setting where both the sensitivity of the final estimator and the truncation parameter $T$ are data dependent.

# 5   Near Optimality and Lower Bounds

In Section 4, we showed that the variance of our realisable private estimator $\widehat{p}_\epsilon^{\text{realistic}}$ is within a constant of that of the complete information estimator $\widehat{p}_\epsilon^{\text{ideal}}$. In this section, we will show that in fact, $\widehat{p}_\epsilon^{\text{realistic}}$ performs as well (up to logarithmic factors) as the true optimal private estimator. We'll also give a lower bound on the performance of the optimal estimator in terms of the $k_i$. This will give us some intuition into the types of distributions of $k_i$'s that benefit from this refined analysis.

## 5.1   Minimax Optimality of $\widehat{p}_\epsilon^{\text{realistic}}$

The goal of this section is to show that the estimator $\widehat{p}_\epsilon^{\text{realistic}}$ discussed in Section 4.2 is minimax optimal up to logarithmic factors. In light of Theorem 4.1, it suffices to show that the estimator $\widehat{p}_\epsilon^{\text{ideal}}$ is minimax optimal up to logarithmic factors. Let $\mathcal{P}$ be a parameterized family of distributions $p \mapsto \mathcal{D}_p$, where $\mathbb{E}[\mathcal{D}_p] = p$ and $\mathcal{D}_p$ is supported on $[0, 1]$. For $p \in [0, 1]$ and $k \in \mathbb{N}$, let $\phi_{p,k}$ be the probability density function of $\mathcal{D}_p(k)$.

Our lower bound will show that the estimation error must consist of a statistical term and a privacy term. Such a lower bound thus must generalize a statistical lower bound. We will rely on the Cramer-Rao approach to proving statistical lower bounds; as we show, it is particularly amenable to incorporating a privacy term. This approach relates the variance of any unbiased estimator of the mean of a distribution to the inverse of the Fischer information; the proof naturally extends to the case where we are given samples from a set of distributions with the same mean but different variances, as is the case in our setting. For many distributions of interest, e.g., Gaussian and Bernoulli, the Fischer information of a single sample is the inverse of the variance, and we make that assumption for $\mathcal{D}_p$. We also assume that the $\mathcal{D}_p$ has sub-Gaussian tails. Thus, as long as the set of permissible meta-distributions includes distributions with this property, e.g., included truncated Gaussians, our lower bound applies.

**Theorem 5.1.** *Let $\mathcal{P}$ be a parameterized family of distributions $p \mapsto \mathcal{D}_p$ and suppose that for all $p \in [0, 1]$ and $k \in \mathbb{N}$, the Fisher information of $\phi_{p,k}$ is inversely proportional to the variance, $\mathrm{Var}(\mathcal{D}_p(k))$:*

$$\int (\tfrac{\partial}{\partial p} \log \phi_{p,k}(x))^2 \phi_{p,k}(x) dx = O(\tfrac{1}{\mathrm{Var}(\mathcal{D}_p(k))}), \tag{8}$$

*and for all $p$, $n > 0$, $k \in \mathbb{N}$ and $\beta \in [1/3, 2/3]$, $f_{\mathcal{D}_p}^k(n, \sigma_p^2, \beta) = \tilde{O}(\mathrm{Var}(\mathcal{D}_p(k)))$, then*

$$\max_{p \in [1/3, 2/3]} [\mathrm{Var}_{\forall i \in [n], x_i \sim \mathcal{D}(k_i), M}(\widehat{p}_\epsilon^{\text{ideal}})] = \tilde{O}(\min_{M, \text{ unbiased}} \max_{p \in [1/3, 2/3]} [\mathrm{Var}_{\forall i \in [n], x_i \sim \mathcal{D}(k_i), M}(M)]).$$

*Further, under the conditions of Theorem 4.1,*

$$\max_{p \in [1/3, 2/3]} [\mathrm{Var}_{\forall i \in [n], x_i \sim \mathcal{D}(k_i), M}(\widehat{p}_\epsilon^{\text{realistic}})] = \tilde{O}(\min_{M, \text{ unbiased}} \max_{p \in [1/3, 2/3]} [\mathrm{Var}_{\forall i \in [n], x_i \sim \mathcal{D}(k_i), M}(M)]).$$

We will prove Theorem 5.1 in three steps. The following class of noisy linear estimators, NLE, will act as an intermediary in our proof. The notation $\sigma_i$ denotes $\mathrm{Var}(x_i)$, which accounts for the randomness in generating $x_i$.

$$\text{NLE} = \{M_{\text{NL}}(\mathbf{x}; \mathbf{w}) = \textstyle\sum_{i=1}^n w_i x_i + \mathrm{Lap}(\tfrac{\max_i w_i \sigma_i}{\epsilon}) \,\big|\, w_i \in [0, 1], \textstyle\sum_{i=1}^n w_i = 1\}.$$

Similar to $\widehat{p}_\epsilon^{\text{ideal}}$, this class of estimators is not realizable since we only have access to an estimate of $\sigma_i = \mathrm{Var}(\mathcal{D}_p(k_i))$. Additionally, the estimators in NLE are not necessarily $\epsilon$-DP.

The proof Theorem 5.1 has three main steps outlined below. The proof of each Lemma is contained in Appendix E. The first step is shown in Lemma 5.2, which shows that the weights used in $\widehat{p}_\epsilon^{\text{ideal}}$ are optimal (i.e., variance-minimizing) among all estimators in the set NLE.

**Lemma 5.2.** *Given $\widehat{p}_i \sim \mathcal{D}_p(k_i)$ with variance $\sigma_i^2$ for all $i \in [n]$ and $w \in [0,1]^n$ such that $\sum_{i=1}^n w_i = 1$, let $\widehat{p} = \sum_{i=1}^n w_i \widehat{p}_i + \mathrm{Lap}(\frac{\max_i w_i \sigma_i}{\epsilon})$. The variance of $\widehat{p}$ is minimized by the following weights: $\tilde{w}_i^* = \frac{\min\{1/\sigma_i^2, T/\sigma_i\}}{\sum_{j=1}^n \min\{1/\sigma_j^2, T/\sigma_j\}}$ for some $T$.*

Since the threshold $T^*$ in $\widehat{p}_\epsilon^{\mathrm{ideal}}$ was chosen to minimize $\mathrm{Var}(\widehat{p}_\epsilon^{\mathrm{ideal}})$, then we know that the weights $w_i^*$ in $\widehat{p}_\epsilon^{\mathrm{ideal}}$ are optimal.

Now, let us turn to the second – and main – component of the proof of Theorem 5.1. Lemma 5.3 formalises the statement that an estimator inside the class NLE is minimax optimal among unbiased estimators. That is, for any unbiased estimator $M$, there exists an estimator $M_{\mathrm{NL}} \in$ NLE with lower worst-case variance.

**Lemma 5.3.** *Let $\mathcal{P}$ be a parameterized family of distributions $p \mapsto \mathcal{D}_p$ and suppose that $M$ : $[0,1]^n \to [0,1]$ is an $\epsilon$-DP estimator such that for all $p \in [1/3, 2/3]$, (1) $M$ is unbiased, $\mu_M(p) = p$, and (2) the Fisher information of $\phi_{p,k_i}$ is inversely proportional to the variance $\mathrm{Var}(\mathcal{D}_p(k_i))$, $\int (\frac{\partial}{\partial p} \log \phi_{p,k_i}(x_i))^2 \phi_{p,k_i}(x_i) dx_i = O(\frac{1}{\mathrm{Var}(\mathcal{D}_p(k_i))})$, then there exists an estimator $M_{\mathrm{NL}} \in$ NLE such that*
$$\max_{p \in [1/3,2/3]} [\mathrm{Var}_{\forall i \in [n], x_i \sim \mathcal{D}(k_i), M_{\mathrm{NL}}}(M_{\mathrm{NL}})] \leq O(\max_{p \in [1/3,2/3]} [\mathrm{Var}_{\forall i \in [n], x_i \sim \mathcal{D}(k_i), M}(M)]).$$

The final component needed for the proof of Theorem 5.1 is a translation from the estimators in NLE, which are not $\epsilon$-DP to the corresponding $\epsilon$-DP estimator. For any weight vector $\mathbf{w}$, we can define an $\epsilon$-DP estimator by truncating the data point $x_i$ and calibrating the noise appropriately:

$$M_{\mathrm{TNL}}(x_1, \cdots, x_n; \mathbf{w}) = \sum_{i=1}^n w_i [x_i]_{p - f_{\mathcal{D}}^{k_i}(n, \sigma_p^2, \beta)}^{p + f_{\mathcal{D}}^{k_i}(n, \sigma_p^2, \beta)} + \mathrm{Lap}(\frac{\max_i 2 w_i f_{\mathcal{D}}^{k_i}(n, \sigma_p^2, \beta)}{\epsilon}). \tag{9}$$

Provided $f_{\mathcal{D}}^{k_i}(n, \sigma_p^2, \beta) \approx \mathrm{Var}(\mathcal{D}(k_i))$, the estimators $M_{\mathrm{TNL}}$ have approximately the same variance as the corresponding element of NLE, but are slightly biased. This is formalized in the following lemma.

**Lemma 5.4.** *For any distribution $\mathcal{D}$, $n > 0$ and $\beta \in [0,1]$, if for all $k_i$, $f_{\mathcal{D}}^{k_i}(n, \sigma_p^2, \beta) = \tilde{O}(\mathrm{Var}(\mathcal{D}(k_i))$ then for any $\mathbf{w} \in [0,1]^n$ such that $\sum_{i=1}^n w_i = 1$, we have $\mathrm{Var}(M_{\mathrm{TNL}}(\cdot \; ; \mathbf{w})) = \tilde{O}(\mathrm{Var}(M_{\mathrm{NL}}(\cdot \; ; \mathbf{w})))$. Further, the bias of $M_{\mathrm{TNL}}$ is at most $\beta$.*

Finally, we have the tools to prove the main theorem in this section, Theorem 5.1:

$$\min_{M \text{ unbiased}} \max_{p \in [1/3,2/3]} [\mathrm{Var}_{\mathcal{D}_p}(M)] = \Omega(\min_{M \in \mathrm{NLE}} \max_{p \in [1/3,2/3]} [\mathrm{Var}_{\mathcal{D}_p}(M)]) = \Omega(\max_{p \in [1/3,2/3]} [\mathrm{Var}_{\mathcal{D}_p}(p_\epsilon^{\mathrm{NLE}})])$$

$$= \tilde{\Omega}(\max_{p \in [1/3,2/3]} [\mathrm{Var}_{\mathcal{D}_p}(\widehat{p}_\epsilon^{\mathrm{ideal}})])$$

$$= \tilde{\Omega}(\max_{p \in [1/3,2/3]} [\mathrm{Var}_{\mathcal{D}_p}(\widehat{p}_\epsilon^{\mathrm{realistic}})])$$

where $p_\epsilon^{\mathrm{NLE}} \in$ NLE has the same weights as $\widehat{p}_\epsilon^{\mathrm{ideal}}$. The equalities follow from Lemmas 5.3, 5.2, 5.4, and Theorem 4.1, respectively.

## 5.2 Minimax Lower Bound on Estimation Rate

In addition to establishing the near optimality of $\widehat{p}_\epsilon^{\mathrm{realistic}}$, we will also give a lower bound on minimax rate of estimation in terms of the parameters $k_1, \cdots, k_n$ and $\sigma_p^2$. Note that we can view the truncation of the weights $w_i$ as establishing an effective upper bound on $k_i$. Given $k_1, \cdots, k_n \in \mathbb{N}$, and $\epsilon > 0$, let $k^* = \arg\min_k \frac{\frac{k}{\epsilon^2} + \sum_{i=1}^n \min\{k_i, k\}}{(\sum_{i=1}^n \min\{k_i, k\})^2}$. Intuitively, in the case that $\sigma_p = 0$, we want to use as many samples as possible, but one user contributing many samples leads to larger sensitivity and thus privacy cost. Limiting to $k_{max}$ the number of samples per user allows us to limit the sensitivity to be about $w_{max}(1/\sqrt{k_{max}})$. Since $w_i$ is proportional to the number of samples used, the variance when using at most $k$ samples per user is the above expression being minimized. Our lower bound below is close to this value for reasonable $k_i$'s.

**Corollary 5.5.** *Given $k_1, \cdots, k_n \in \mathbb{N}$, and $\sigma_p$, there exists a family of distributions $\mathcal{D}_p$ such that*

$$\min_{M, \text{ unbiased}} \max_{p \in [1/3,2/3]} \mathrm{Var}_{\forall i \in [n], x_i \sim \mathcal{D}_p(k_i)} [M(x_1, \cdots, x_n)] \geq \tilde{\Omega}(\min\{\frac{\frac{k^*}{\epsilon^2} + \sum_{i=1}^n \min\{k_i, k^*\}}{(\sum_{i=1}^n \min\{k_i, \sqrt{k_i k^*}\})^2}, \frac{\sigma_p^2}{n}\}).$$

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
