# A    Proofs from Section 2

**Lemma 2.1.** *For all distributions $\mathcal{D}$ supported on $[0,1]$ with mean $p$ and variance $\sigma_p^2$, $\sigma_p^2 \leq p(1-p)$.
Further, $\mathbb{E}[\mathcal{D}(k)] = p$ and $\mathrm{Var}(\mathcal{D}(k)) = \frac{1}{k}p(1-p) + \left(1 - \frac{1}{k}\right)\sigma_p^2$.*

*Proof of Lemma 2.1.* Firstly, note that,

$$\sigma_p^2 = \mathbb{E}_{x \sim \mathcal{D}}[x^2] - p^2 \leq \mathbb{E}_{x \sim \mathcal{D}}[x] - p^2 = p(1-p),$$

where the inequality follows from the fact that $\mathcal{D}$ is supported on $[0,1]$.

Next,

$$\mathbb{E}[x_i] = \int_{x=0}^{1} \Pr(p_i = x)\Pr(\mathrm{Ber}(x) = 1)dx = \int_{x=0}^{1} \Pr(p_i = x)x\,dx = p,$$

which by linearity of expectation implies that $\mathbb{E}[\mathcal{D}(k)] = p$.

By the Law of Total Variation, the variance of $\widehat{p}_i$ is:

$$\begin{aligned}
\mathrm{Var}(\widehat{p}_i) &= \mathbb{E}_{p_i}[\mathrm{Var}_{x_i}(\widehat{p}_i|p_i)] + \mathrm{Var}_{p_i}(\mathbb{E}_{x_i}[\widehat{p}_i|p_i]) \\
&= \mathbb{E}_{p_i}[\frac{1}{k_i}p_i(1 - p_i)] + \mathrm{Var}_{p_i}(p_i) \\
&= \frac{1}{k_i}(p - \sigma_p^2 - p^2) + \sigma_p^2 \\
&= \frac{1}{k_i}(p - p^2) + (1 - \frac{1}{k_i})\sigma_p^2. \\
&= \frac{1}{k_i}\mathrm{Var}(\mathrm{Ber}(p)) + (1 - \frac{1}{k_i})\sigma_p^2.
\end{aligned}$$

$\square$

# B    Proofs from Section 4.2

**Theorem 4.1.** *For any $\epsilon > 0$, $\delta \in [0,1]$, Algorithm 2 is $(\epsilon, \delta)$-DP. If,*

- *$\mathtt{mean}_{\epsilon,\delta}$ is such that given $n/10$ samples from $\mathcal{D}$, with probability $1 - \beta$ $|p - \widehat{p}_\epsilon^{\mathrm{initial}}| \leq f_{\mathcal{D}}^{k_i}(n, \sigma_p^2, \beta)$ and $\widehat{p}_\epsilon^{\mathrm{initial}}(1 - \widehat{p}_\epsilon^{\mathrm{initial}}) \in [\frac{1}{2}p(1-p), \frac{3}{2}p(1-p)]$,*

- *$\mathtt{variance}_{\epsilon,\delta}$ is such that given $\log n$ samples from $\mathcal{D}(k)$, with probability $1 - \beta$, $\widehat{\sigma}_p^2 \in [\mathrm{Var}(\mathcal{D}(k)), \ 8\mathrm{Var}(\mathcal{D}(k))]$,*

- *the $k_i$s are such that $\frac{k_1}{k_{n/2}} \leq \frac{n/2 - \log n}{\log n}$,*

*then with probability $1 - 2\beta$, $\mathrm{Var}(\widehat{p}_\epsilon^{\mathrm{realistic}}) \leq C \cdot \mathrm{Var}(\widehat{p}_\epsilon^{\mathrm{ideal}})$ for some absolute constant $C$.*

First, let us show that the conditions of Theorem 4.1 imply that the variance and truncation parameter estimates of each individual data subject are correct up to constant factors.

**Lemma 4.2.** *Given $\widehat{p}_\epsilon^{\mathrm{initial}}$, $\widehat{\sigma}_p^2$, and $k_i$, define $\widehat{\sigma}_i^2 = \frac{1}{k_i}\widehat{p}_\epsilon^{\mathrm{initial}}(1 - \widehat{p}_\epsilon^{\mathrm{initial}}) + \frac{k_i - 1}{k_i}\widehat{\sigma}_p^2$. Under the conditions of Theorem 4.1 for all $i > \log n$, we have $\widehat{\sigma}_i^2 \in [\frac{1}{2}\sigma_i^2, 9.5\sigma_i^2]$ and $|\widehat{b}_i - \widehat{a}_i| \leq 4|b_i - a_i|$.*

*Proof of Lemma 4.2.* Note that $\widehat{\sigma}_p^2$ is actually an estimate of the variance of $\mathcal{D}(k_{\log n})$ since it has access to samples from this distribution rather than $\mathcal{D}$ itself. Therefore, $\widehat{\sigma}_p^2 \in [\mathrm{Var}(\mathcal{D}(k_{\log n})), 8 \cdot \mathrm{Var}(\mathcal{D}(k_{\log n}))]$ implies $\widehat{\sigma}_p^2 \in [\sigma_p^2, 8\left(\frac{1}{k_{\log n}}p(1-p) + \sigma_p^2\right)]$. Then for every

$i \geq \log n$ (i.e., with $k_i \leq k_{\log n}$),

$$\widehat{\sigma}_i^2 = \frac{1}{k_i}\widehat{p}_\epsilon^{\text{initial}}(1 - \widehat{p}_\epsilon^{\text{initial}}) + \frac{k_i - 1}{k_i}\widehat{\sigma}_p^2$$

$$\geq \frac{1}{k_i}\frac{1}{2}p(1-p) + \frac{k_i - 1}{k_i}\sigma_p^2$$

$$\geq \frac{1}{2}\left(\frac{1}{k_i}p(1-p) + \frac{k_i - 1}{k_i}\sigma_p^2\right)$$

$$= \frac{1}{2}\sigma_i^2,$$

where the first inequality follows from the accuracy conditions on $\texttt{mean}_{\epsilon,\delta}$ and $\texttt{variance}_{\epsilon,\delta}$ in Theorem 4.1, and the last equality follows from the definition of $\sigma_i^2$ in Lemma 2.1. Also,

$$\widehat{\sigma}_i^2 = \frac{1}{k_i}\widehat{p}_\epsilon^{\text{initial}}(1 - \widehat{p}_\epsilon^{\text{initial}}) + \frac{k_i - 1}{k_i}\widehat{\sigma}_p^2$$

$$\leq \frac{1}{k_i}\frac{3}{2}p(1-p) + 8\frac{k_i - 1}{k_i}\left(\frac{1}{k_{\log n}}p(1-p) + \sigma_p^2\right)$$

$$= \left(\frac{3}{2} + 8\frac{k_i - 1}{k_{\log n}}\right)\frac{1}{k_i}p(1-p) + 8\frac{k_i - 1}{k_i}\sigma_p^2$$

$$\leq 9.5\left(\frac{1}{k_i}p(1-p) + \frac{k_i - 1}{k_i}\sigma_p^2\right)$$

$$= 9.5\sigma_i^2,$$

where again, the first inequality follows from the accuracy conditions on $\texttt{mean}_{\epsilon,\delta}$ and $\texttt{variance}_{\epsilon,\delta}$ in Theorem 4.1, and the last equality follows from the definition of $\sigma_i^2$ in Lemma 2.1. The intermediate steps are simply algebraic manipulations. These two facts give us the desired bounds on $\widehat{\sigma}_i^2$.

Next we turn to the truncation parameters $\widehat{a}_i$ and $\widehat{b}_i$. Using the definition of $\widehat{a}_i$ in Algorithm 2, we have,

$$\widehat{a}_i = \widehat{p}_\epsilon^{\text{initial}} - \alpha - f_{\mathcal{D}}^{k_i}(n, \widehat{\sigma_p^2}, \beta/2)$$

$$\leq p - f_{\mathcal{D}}^{k_i}(n, \widehat{\sigma_p^2}, \beta/2))$$

$$\leq p - f_{\mathcal{D}}^{k_i}(n, \sigma_p^2, \beta)$$

$$= a_i,$$

where the two inequalities respectively follow from the accuracy conditions on $\texttt{mean}_{\epsilon,\delta}$ and $\texttt{variance}_{\epsilon,\delta}$ in Theorem 4.1. A symmetric result that $\widehat{b}_i \geq b_i$ follows similarly.

Finally,

$$|\widehat{b}_i - \widehat{a}_i| = 2\alpha + 2f_{\mathcal{D}}^{k_i}(n, \widehat{\sigma_p^2}, \beta)$$

$$\leq 2f_{\mathcal{D}}^{k_i}(n, \sigma_p^2, \beta) + 2f_{\mathcal{D}}^{k_i}(n, \sigma_p^2, \beta)$$

$$= 4f_{\mathcal{D}}^{k_i}(n, \sigma_p^2, \beta)$$

$$= 4|b_i - a_i|.$$

The inequalities again follows from the accuracy conditions on $\texttt{mean}_{\epsilon,\delta}$ and $\texttt{variance}_{\epsilon,\delta}$ in Theorem 4.1. $\qquad\square$

*Proof of Theorem 4.1.* To see that Algorithm 2 is differentially private, consider the three cohorts into which users are placed. The first cohort, containing the $n/10$ users with the smallest $k_i$ will have their data used in $\texttt{mean}_{\epsilon,\delta}$, which is $(\epsilon, \delta)$-DP. Similarly, the second cohort containing the $\log n$ users with the largest $k_i$ will have their data used in $\texttt{variance}_{\epsilon,\delta}$, which is also $(\epsilon, \delta)$-DP. The intermediate estimators of $\widehat{\sigma}_i^2$, $\widehat{T}^*$, $\widehat{a}_i$, $\widehat{b}_i$, and sensitivity $\Lambda$ are all computed as post-processing on the private outputs of these initial estimation subroutines and on the public $k_i$s, and thus do not incur

any additional privacy cost. The third cohort contains the middle users $i \in [\log n + 1, 9n/10]$. These users' data are only used in the final estimate, which is an $(\epsilon, 0)$-DP instantiation of the Laplace Mechanism [12].

Since these cohorts are disjoint and private algorithms are applied to each cohort's data separately, parallel composition applies, and the overall privacy parameters are the maximum of those experienced by any cohort, so the overall algorithm is $(\epsilon, \delta)$-DP.

For accuracy of the $\widehat{p}_\epsilon^{\text{realistic}}$ estimator produced by Algorithm 2, first notice that under the assumption that $\frac{k_{\max}}{k_{\text{med}}} \leq \frac{n/2 - \log n}{\log n}$, if $\sigma^2_{k_{\max}} = \text{Var}(\widehat{p}_1)$ and $\sigma^2_{k_{\text{med}}} = \text{Var}(\widehat{p}_{n/2})$ then

$$\sigma^2_{k_{\text{med}}} = \frac{1}{k_{\text{med}}} p(1-p) + \left(1 - \frac{1}{k_{\text{med}}}\right) \sigma^2_p \leq \frac{n/2 - \log n}{\log n} \frac{1}{k_{\max}} p(1-p) + \left(1 - \frac{1}{k_{\max}}\right) \sigma^2_p \leq \frac{n/2 - \log n}{\log n} \sigma^2_{k_{\max}}.$$

Therefore, for any truncation parameter $T$,

$$\frac{1}{2} \sum_{i=1}^{n} \min\left\{\frac{1}{\sigma_i^2}, \frac{T}{\sigma_i}\right\} \leq \sum_{i=1}^{n/2} \min\left\{\frac{1}{\sigma_i^2}, \frac{T}{\sigma_i}\right\}$$

$$= \sum_{i=1}^{\log n} \min\left\{\frac{1}{\sigma_i^2}, \frac{T}{\sigma_i}\right\} + \sum_{i=\log n+1}^{n/2} \min\left\{\frac{1}{\sigma_i^2}, \frac{T}{\sigma_i}\right\}$$

$$\leq \log n \cdot \min\left\{\frac{1}{\sigma^2_{k_{\max}}}, \frac{T}{\sigma_{k_{\max}}}\right\} + \sum_{i=\log n+1}^{n/2} \min\left\{\frac{1}{\sigma_i^2}, \frac{T}{\sigma_i}\right\}$$

$$\leq (n/2 - \log n) \cdot \min\left\{\frac{1}{\sigma^2_{k_{\text{med}}}}, \frac{T}{\sigma_{k_{\text{med}}}}\right\} + \sum_{i=\log n+1}^{n/2} \min\left\{\frac{1}{\sigma_i^2}, \frac{T}{\sigma_i}\right\}$$

$$\leq 2 \sum_{i=\log n+1}^{n/2} \min\left\{\frac{1}{\sigma_i^2}, \frac{T}{\sigma_i}\right\}$$

$$\leq 2 \sum_{i=\log n+1}^{9n/10} \min\left\{\frac{1}{\sigma_i^2}, \frac{T}{\sigma_i}\right\}, \tag{10}$$

where the first, second, and fourth inequalities follow from our assumed ordering on the $k_i$s. The third inequality comes from our assumption on $k_{\max}$ and $k_{\text{med}}$, and the final inequality follows from the fact that the summands $\min\{\frac{1}{\sigma_i^2}, \frac{T}{\sigma_i}\}$ are positive so adding more terms only increases the sum.

Therefore,

$$
\mathrm{Var}(\widehat{p}_\epsilon^{\mathrm{realistic}}) = \frac{1}{(\sum_{j=\log n+1}^{9n/10}\min\{1/\widehat{\sigma_j}^2,\frac{\widehat{T}^*}{\widehat{\sigma_i}}\})^2}\left(\sum_{i=\log n+1}^{9n/10}\min\{\frac{1}{\widehat{\sigma_i}^4},\frac{\widehat{T}^{*2}}{\widehat{\sigma_i}^2}\}\sigma_i^2 + \max_i \frac{\min\{\frac{1}{\widehat{\sigma_i}^4},\frac{\widehat{T}^{*2}}{\widehat{\sigma_i}^2}\}|\widehat{b_i}-\widehat{a_i}|^2}{\epsilon^2}\right)
$$

$$
\leq \frac{1}{(\sum_{j=\log n+1}^{9n/10}\min\{1/\widehat{\sigma_j}^2,\frac{\widehat{T}^*}{\widehat{\sigma_i}}\})^2}\left(\sum_{i=\log n+1}^{9n/10}\min\{\frac{1}{\widehat{\sigma_i}^4},\frac{\widehat{T}^{*2}}{\widehat{\sigma_i}^2}\}2\widehat{\sigma_i}^2 + \max_i \frac{\min\{\frac{1}{\widehat{\sigma_i}^4},\frac{\widehat{T}^{*2}}{\widehat{\sigma_i}^2}\}|\widehat{b_i}-\widehat{a_i}|^2}{\epsilon^2}\right)
$$

$$
\leq 2\frac{1}{(\sum_{j=\log n+1}^{9n/10}\min\{1/\widehat{\sigma_j}^2,\frac{\widehat{T}^*}{\widehat{\sigma_i}}\})^2}\left(\sum_{i=\log n+1}^{9n/10}\min\{\frac{1}{\widehat{\sigma_i}^2},\widehat{T}^{*2}\} + \max_i \frac{\min\{\frac{1}{\widehat{\sigma_i}^4},\frac{\widehat{T}^{*2}}{\widehat{\sigma_i}^2}\}|\widehat{b_i}-\widehat{a_i}|^2}{\epsilon^2}\right)
$$

$$
\leq 2\frac{1}{(\sum_{j=\log n+1}^{9n/10}\min\{1/\widehat{\sigma_j}^2,\frac{T^*}{\widehat{\sigma_i}}\})^2}\left(\sum_{i=\log n+1}^{9n/10}\min\{\frac{1}{\widehat{\sigma_i}^2},T^{*2}\} + \max_i \frac{\min\{\frac{1}{\widehat{\sigma_i}^4},\frac{T^{*2}}{\widehat{\sigma_i}^2}\}|\widehat{b_i}-\widehat{a_i}|^2}{\epsilon^2}\right)
$$

$$
\leq 2\frac{1}{(\sum_{j=\log n+1}^{9n/10}\min\{1/10\sigma_j^2,\frac{\sqrt{2}T^*}{\sigma_i}\})^2}\left(\sum_{i=\log n+1}^{9n/10}\min\{\frac{2}{\sigma_i^2},T^{*2}\} + \max_i \frac{\min\{\frac{4}{\sigma_i^4},\frac{2T^{*2}}{\sigma_i^2}\}6|b_i-a_i|^2}{\epsilon^2}\right)
$$

$$
\leq 480\frac{1}{(\sum_{j=\log n+1}^{9n/10}\min\{1/\sigma_j^2,\frac{T^*}{\sigma_i}\})^2}\left(\sum_{i=\log n+1}^{9n/10}\min\{\frac{1}{\sigma_i^2},T^{*2}\} + \max_i \frac{\min\{\frac{1}{\sigma_i^4},\frac{T^{*2}}{\sigma_i^2}\}|b_i-a_i|^2}{\epsilon^2}\right)
$$

$$
\leq 480\frac{1}{\frac{1}{16}(\sum_{j=1}^{n}\min\{1/\sigma_j^2,\frac{T^*}{\sigma_i}\})^2}\left(\sum_{i=1}^{n}\min\{\frac{1}{\sigma_i^2},T^{*2}\} + \max_i \frac{\min\{\frac{1}{\sigma_i^4},\frac{T^{*2}}{\sigma_i^2}\}|b_i-a_i|^2}{\epsilon^2}\right)
$$

$$
= 7680 \cdot \mathrm{Var}(\widehat{p}_\epsilon^{\mathrm{ideal}})
$$

The first equality simply follows from the definition of the estimator and basic properties of the variance, as well as the fact that $\mathrm{Var}([\widehat{p}_i]_{a_i}^{b_i}) \leq \sigma_i$. The first inequality follows from the fact that $\sigma_i^2 \leq 2\widehat{\sigma_i}^2$, which was shown in Lemma 4.2. The second inequality is simply pulling out the constant to the front. The third inequality follows from the definition of $\widehat{T}$ as the optimiser of the variance using the approximations $\widehat{\sigma_i}^2$, $\widehat{b_i}$ and $\widehat{a_i}$. The fourth inequality follows from the fact that $\widehat{\sigma_i}^2 \in \left[\frac{1}{2}\sigma_i^2, 10\sigma_i^2\right]$ and $|\widehat{b_i}-\widehat{a_i}| \leq 4|b_i-a_i|$, as shown in Lemma 4.2, and will hold with probability $1-2\beta$, by taking a union bound over the $\beta$ failure probabilities from each of the $\mathtt{mean}_{\epsilon,\delta}$ and $\mathtt{variance}_{\epsilon,\delta}$ subroutines. The fifth inequality simply pulls out the constants ($480=2*10*4*6$). The final inequality follows from Equation (10) above. The final equality follows from definition of $\widehat{p}_\epsilon^{\mathrm{ideal}}$ and the assumption that $\frac{1}{2}\sigma_i^2 \leq \mathrm{Var}([\widehat{p}_i]_{a_i}^{b_i})$.

$\square$

## C  Special Case: The constant $p_i$ case.

In the previous section, we considered the setting where there was heterogeneity in both the users' distributions (i.e., the $p_i$s were not constant), as well as the number of data points that they each held (i.e., the $k_i$s were not constant). In the absence of variation in the $p_i$, each user is sampling from the same distribution $\mathrm{Ber}(p)$. When privacy is not a concern, this setting reduces to the single-data-point-per-user setting where the sample size is increased to $\sum_{i=1}^{n} k_i$. However, under the constraint of user-level differential privacy, this setting is distinct from the single-data-point-per-user setting, since we need to protect the entirety of each users data set. In fact, much of the complexity of Algorithm 2 is required even in this simpler case. In particular, the truncated inverse variance weighting is still required in this case when there is variation in the $k_i$. In fact, the only step of Algorithm 2 that is not required is Step **??**, since already know that $\sigma_p^2 = 0$. Since there is no variance in $\mathcal{D}$, the high probability bound $f_{\mathcal{D}}^{k_i}(n, \widehat{\sigma_p^2}, \beta)$ is just due to the randomness in the binomial distribution $\mathrm{Bin}(k_i, p)$, which comes from averaging $k_i$ samples drawn from $\mathrm{Ber}(p)$.

When $\sigma_p^2 = 0$, $\sigma_i$ has the simple formula $\sigma_i = \frac{\sqrt{p(1-p)}}{k_i}$ and we can directly translate from the truncation threshold $T$ on $\sigma_i$ to a truncation threshold $k$ on $k_i$, $T = \frac{\sqrt{p(1-p)}}{k}$. Further, if we assume that all the $k_i$ are large enough ($\min k_i \geq 2\ln(1/\delta)/p$) then we also have the simple formula $f_{\mathcal{D}}^{k_i}(n, \widehat{\sigma_p^2}, \beta) = \sqrt{\frac{3p\ln(2/\beta)}{k_i}}$. We can plug these into Equation (6) (recall that $T^*$ is defined as the truncation threshold that minimizes the variance of $\widehat{p}_\epsilon^{\text{ideal}}$) to obtain the following formula for the variance of $\widehat{p}_\epsilon^{\text{ideal}}$, and hence $\widehat{p}_\epsilon^{\text{realistic}}$:

$$\min_{k} \frac{p(1-p)\sum_{i=1}^{n}\min\{k_i,k\} + 6p\ln(2/\beta)\frac{k}{\epsilon^2}}{(\sum_{j=1}^{n}\min\{k_i,\sqrt{k_i k}\})^2}. \tag{11}$$

Even in the private setting, one can reduce to the single-data-point-per-user setting by reducing the sample size by a factor of 2, and forcing the $n/2$ users with the most data points to produce their estimate $\hat{p}_i$ using only $k_{\text{med}}$ (the median $k_i$) data points. Then each estimate $\hat{p}_i$ is a sample from the same distribution and we can compute their mean. To the best of our knowledge, all the prior work in the private literature that handles variations in $k_i$ follows this formula. However, not only does this algorithm reduce the sample size by a factor of 2, it also unnecessarily hinders the contribution of users with many data points. As a simple example, suppose that all the users have a single data point, except for $\sqrt{n}$ users, which have $n$ data points. Then the algorithm which forces $n/2$ of the users to use the median number of data points has an error rate of $\Theta(\frac{1}{n} + \frac{1}{n^2\epsilon^2})$ assuming that $p$ is bounded away from 0 or 1. Letting $k = n$ in Equation 11 implies that that the truncated inverse variance weighted algorithm in the previous section is better able to utilise the data of the users with high $k_i$s, resulting in an error rate of $O(\frac{1}{n^{3/2}} + \frac{1}{n^2\epsilon^2})$.

## D   Extension: private-size user-level differential privacy setting

When defining user-level DP where users have heterogeneous quantities of data, we also need to distinguish between settings where the number of data points held by each user is protected information, and settings where it is publicly known. We'll refer to the former as *private-size user-level differential privacy*, where the entry that differs between neighboring databases can have arbitrarily different number of data points, and the latter as *public-size user-level differential privacy*, where the amount of data held by each user is the same in neighboring databases. Formally, let $D_i = \{x_i^1, \cdots, x_i^{k_i}\}$ be the data of user $i$ for each $i \in [n]$. For private-size user-level differential privacy, we say $D$ and $D'$ are neighbours if there exists an index $i$ such that for all $j \in [n]\backslash\{i\}$, $D_j = D_j'$. For public-size user-level differential privacy, we say $D$ and $D'$ are neighbours if they are neighbours under private-size user-level differential privacy and additionally $|D_i| = |D_i'|$ for all $i \in [n]$.

Let us now turn to our problem in the private-size user-level differential privacy setting, where the $k_i$s are private information and require formal privacy protections. We will need to make several changes to Algorithm 2 to make it private under this stronger notion of privacy. Under public-size user-level privacy, the quantities $\hat{T}^*$ (the weight truncation parameter) and $\Lambda$ (the sensitivity of the final estimate) in Algorithm 2 do not pose privacy concerns since they only depend on the data points $\hat{p}_i$ through the $\widehat{p}_\epsilon^{\text{initial}}$ and $\widehat{\sigma}_i^2$, which are both produced differentially privately. However, both these quantities depend on the $k_i$ directly, and hence care needs to be taken when using them under private-size user-level DP.

In Algorithm 3, we outline the extension of Algorithm 2 to satisfy private-size user-level differential privacy. The core component of the final estimate is described in Algorithm 4. The function $\mathcal{M}$ as described in Algorithm 4 incorporates the truncation in a slightly different (but equivalent) manner to Algorithm 2, but is otherwise the same, without the addition of noise.

The first key difference is how the weights are truncated. Observe that choosing a truncation parameter $T$ is equivalent to choosing an integer $k$ such that $T = 1/\text{Var}(\mathcal{D}(k))$. So $\widehat{k_{\log n}}$ plays the role in Algorithm 3 that $T^*$ plays in Algorithm 2. The statistic $\widehat{k_{\log n}}$ is a private estimate of the $\log n$-th order statistic of the set $\{k_1, \cdots, k_n\}$. Since the only users that participate in the final estimate (and hence have their data truncated) all have $k_i < k_{\log n}$, this algorithm attempts to find the smallest truncation parameter such that no data is actually truncated. We will show that provided $\epsilon$ is not too small, this level of truncation is sufficient.

**Algorithm 3** Private Heterogeneous Mean Estimation $\hat{p}_\epsilon^{\text{realistic, unknown size}}$

**Input:** $(\epsilon, \delta)$-DP mean estimator $\texttt{mean}_{\epsilon,\delta}$, $(\epsilon, \delta)$-DP variance estimator $\texttt{variance}_{\epsilon,\delta}$, $\epsilon$-DP estimator of the $\ell$th order statistic $\texttt{EM}_\epsilon(\cdot; \ell)$, number of users $n$, number of samples held by each user $(k_1, \ldots, k_n \ s.t. \ k_i \geq k_{i+1})$, an upper bound on the total number of data points held by a single user $k_{\max}$, user-level estimates $(\widehat{p}_1, \cdots, \widehat{p}_n)$, a constant $C > 0$, error guarantee on $\texttt{mean}_{\epsilon,\delta}$ $\alpha > 0$, and desired high probability bound $\beta \in [0, 1]$.

1: **Initial Estimates**
2: $\quad \widehat{p}^{\text{initial}} = \texttt{mean}_{\epsilon,\delta}(\mathrm{x}_{9\mathrm{n}/10+1}^1, \cdots, \mathrm{x}_\mathrm{n}^1)$ $\hfill$ Initial mean estimate
3: $\quad \widehat{\sigma}_p^2 = \texttt{variance}_{\epsilon,\delta}(\widehat{p}_1, \cdots, \widehat{p}_{\log n})$ $\hfill$ Initial variance estimate

4: **Compute Sensitivity Proposal**
5: $\quad \widehat{k_T} = \texttt{EM}_\epsilon(k_1, \cdots, k_n; \frac{2\log n}{\epsilon}, k_{\max})$ $\hfill$ Compute weight truncation
6: **for** $i \in [\log n + 1, n/2]$ **do**
7: $\quad\quad k_i' = \min\{k_i, \widehat{k_T}\}$
8: $\quad\quad \widehat{\sigma_i'}^2 = \frac{1}{k_i'}(\widehat{p}_\epsilon^{\text{initial}} - (\widehat{p}_\epsilon^{\text{initial}})^2) + (1 - \frac{1}{k_i'})\widehat{\sigma}_p^2.$
9: $\quad\quad v_i = \frac{1}{\widehat{\sigma_i'}^2}$ $\hfill$ Compute truncated, unnormalised weights
10: $\quad \widehat{\sigma_{\min}}^2 = \frac{1}{\widehat{k_T}}(\widehat{p}_\epsilon^{\text{initial}} - (\widehat{p}_\epsilon^{\text{initial}})^2) + (1 - \frac{1}{\widehat{k_T}})\widehat{\sigma}_p^2.$
11: $\quad \widehat{N} = \sum_{j=\log n+1}^{9n/10} v_i + \texttt{Lap}\left(\frac{1}{\epsilon\widehat{\sigma_{\min}}^2}\right) - \frac{1}{\epsilon\widehat{\sigma_{\min}}^2}\ln(2\delta)$ $\hfill$ Compute noisy normalisation term
12: $\quad \Upsilon = \frac{\log(1/\delta)}{\epsilon} + \frac{\ln(1/\delta)\ln(1/\gamma)}{\epsilon}$
13: $\quad \Lambda = 8\left(\sqrt{\Upsilon + 1}\log n + \frac{\log(1/\beta)}{\sqrt{\Upsilon+1}}\right)\frac{f_\mathcal{D}^{k_{\max}}(n,\widehat{\sigma_p}^2,\beta)}{\widehat{\sigma_{\max}}^2\widehat{N}}$ $\hfill$ Compute local sensitivity proposal

14: **Propose-Test-Release on** $\mathcal{M}(\cdot; \widehat{k_{\log n}}, n, \widehat{p}_\epsilon^{\text{initial}}, \widehat{\sigma}_p^2, \alpha)$
15: $\quad D_T = \{(\widehat{p}_i, k_i)\}_{i \in [\log n+1:9n/10]}$
16: $\quad \kappa^* = \arg\max\{\kappa \in \mathbb{N} \mid \forall D' \text{ s.t. } D' \text{ is a } \kappa\text{-neighbor of } D_T, \Delta(\mathcal{M}(\cdot; \widehat{k_T}, n, \widehat{p}_\epsilon^{\text{initial}}, \widehat{\sigma}_p^2, \alpha); D') \leq \Lambda\}$ $\hfill$ Compute distance to high sensitivity dataset
17: $\quad \tilde{\kappa} = \kappa^* + \texttt{Lap}(1/\epsilon)$
18: **if** $\tilde{\kappa} < \frac{\log(1/\delta)}{\epsilon}$ **then**
19: $\quad$ **return** $\hat{p}_\epsilon^{\text{realistic, unknown size}} = \widehat{p}_\epsilon^{\text{initial}}$ $\hfill$ Return initial estimate if proposed local sens. too small
20: **else**
21: $\quad$ Sample $Y \sim \texttt{Lap}\left(\frac{\Lambda}{\epsilon}\right)$ $\hfill$ Sample noise added for privacy
22: $\quad$ **return** $\hat{p}_\epsilon^{\text{realistic, unknown size}} = \mathcal{M}(D_T; \widehat{k_T}, n, \widehat{p}_\epsilon^{\text{initial}}, \widehat{\sigma}_p^2, \alpha) + Y$ $\hfill$ Final estimate

The second significant change in Algorithm 3 is how the sensitivity parameter $\Lambda$ is chosen. The final statistic is more sensitive under the view of private-size user level privacy; the weight of every user can change as a result of a single user changing the amount of data they hold (due to the normalisation constant changing). Thus, the formula for the scale of the noise given in Algorithm 3 is higher than the noise added in Algorithm 2. Further, $\Lambda$ as defined in Algorithm 3 is not an upper bound on the local sensitivity for *all* databases, although with high probability it is an upper bound on the local sensitivity of all databases that lie in a neighbourhood of $D$. So, we use a standard framework from the differential privacy literature called propose-test-release (PTR) to privately verify that $\Lambda$ is indeed an upper bound on the local sensitivity of all databases in a neighbourhood of $D$, which allows us to safely add noise proportional to $\Lambda$ to privatise the final statistic.

There are several existing algorithms in the literature that can be used to privately estimate the $\frac{\log n}{\epsilon}$-th order statistic $\widehat{k_T}$. A simple algorithm [10, 32, 19, 3, 4] that estimates the order statistic using the common differential privacy framework called the exponential mechanism [28] is sufficient up to a constant factor. For a full description of this algorithm, as well as its accuracy guarantees see [4]. In order for this algorithm to produce accurate results, we need an upper bound on the maximum number of data points a single user can have; we will call this number $k_{\max}$.

---

**Algorithm 4** Truncated weighted mean, $\mathcal{M}(\cdot; k_{\max}, n, \hat{p}, \hat{\sigma_p}^2, \alpha)$

---

**Input:** number of users $n$, number of samples held by each user $(k_1, \dots, k_n)$, user-level estimates $(\widehat{p}_1, \cdots, \widehat{p}_n)$, upper bound $k_{\max}$, lower bound $k_{\min}$, mean estimate $\hat{p}$, variance estimate $\hat{\sigma}_p^2$, accuracy on mean estimate $\alpha$

---

1: **for** $i \in [n]$ **do**
2: $\quad \widehat{a}_i = \hat{p} - \alpha - f_{\mathcal{D}}^{k_i}(n, \hat{\sigma_p}^2, \beta)$
3:
4: $\quad \widehat{b}_i = \hat{p} + \alpha + f_{\mathcal{D}}^{k_i}(n, \hat{\sigma_p}^2, \beta)$
5:
6: $\quad k_i' = \min\{k_i, k_{\max}\}$
7: $\quad \widehat{\sigma}_i^2 = \frac{1}{k_i'}(\hat{p} - (\hat{p})^2) + (1 - \frac{1}{k_i'})\hat{\sigma}_p^2.$
8: $\quad v_i = \frac{1}{\widehat{\sigma}_i^2}$

9: Return $\dfrac{\sum_{i \in [n]} v_i [\widehat{p}_i]_{\widehat{a}_i}^{\widehat{b}_i}}{\sum_{i \in [n]} v_i}$

---

A $\kappa$-neighbour of $D$ is a database $D'$ of size $n$ that differs from $D$ on the data of at most $\kappa$ data subjects. Given a function $f$ from the set of databases to $\mathbb{R}$, and a database $D$, the local sensitivity of $f$ at $D$ is defined by $\Delta(f; D) = \max_{D' \text{ neighbour of } D} |f(D) - f(D')|$.

**Theorem D.1.** *For any $\epsilon > 0$ and $\delta \in [0, 1]$, Algorithm 3 is $(3\epsilon, 2\delta)$-DP. If the conditions of Theorem 4.1 hold and*

- *$\epsilon \geq \frac{2 \log n}{n}$,*

- *$k_{\max} \leq \sqrt{n}$ and $\beta \leq 1/\sqrt{n}$,*

- *if $k_{med}$ is the median $k_i$ value then $\frac{k_{\max}}{k_{med}} \leq$ $\min\left\{ \frac{1}{2}(n - \Upsilon - 1), \frac{n-1}{\Upsilon+1}, \frac{\epsilon^3(n/2 - \log n - 1)}{\log n \log(1/\delta) \log(1/\beta)}, \frac{(n/4 - 1)\epsilon}{3 \ln(2/\delta)}, \frac{n/2 - \frac{\log n}{\epsilon}}{\frac{\log n}{\epsilon}} \right\},$*

- *$\max\{\alpha, \sigma_{k_{\max}}\} \leq f_{\mathcal{D}}^{k_{\max}}(n, \hat{\widehat{\sigma}}_p^2, \beta)$, and*

- *for any set $I \subset \{1, m\}$ then with probability $1 - \beta$, $\left| \frac{\sum_{i \in I} v_i \widehat{p}_i}{\sum_{i \in I} v_i} - p \right| \leq 2\mathrm{Var}\left( \frac{\sum_{i \in I} v_i \widehat{p}_i}{\sum_{i \in I} v_i} \right) \log(1/\beta),$*

*then with probability $1 - 4\beta$, $\mathrm{Var}(\hat{p}_\epsilon^{realistic, unknown size}) \leq \tilde{O}\left( \mathrm{Var}(\widehat{p}^{realistic}) + \frac{\mathrm{Var}(\widehat{p}_\epsilon^{realistic})}{\sqrt{\epsilon}} \right)$*

Theorem D.1 implies that when $\epsilon$ is sufficiently large ($\epsilon \geq \frac{\log n}{n}$) then the variance of $\hat{p}_\epsilon^{\text{realistic, unknown size}}$ is within a constant of $\hat{p}_\epsilon^{\text{realistic}}$, except that the final noise added for privacy is $\tilde{O}(1/\sqrt{\epsilon})$ larger. While the conditions of this theorem may seem intimidating, note that none of them are particularly stringent. The first condition $\epsilon \geq 2 \log n/n$ is a reasonable assumption on $\epsilon$ (we note that essentially no learning is possible with $\epsilon < 1/n$). The second condition on $k_{\max}$ is a reasonable assumption if the number of users $n$ is large, as may be the case in practice. The third condition, up to logarithmic factors, can be viewed as $k_{\max}/k_{\text{med}} \leq \epsilon^3 n$. This is a stronger assumption than was required for Theorem 4.1 but provided reasonable values of $\epsilon$s, it is not prohibitive in practice. The final condition is simply a concentration bound on the weighted sums; this condition holds for Gaussians.

The proof that Algorithm 3 is $(3\epsilon, 2\delta)$-DP is fairly routine so we will present it first. The population is broken into three cohorts. Let us consider each cohort individually. First, consider the $\log n$ individuals with the most data. They participate in private releases in lines (3) ($(\epsilon, \delta)$-DP), and (5) ($\epsilon$-DP). Using the simple composition rule of differential privacy [12], Algorithm 3 is $(2\epsilon, \delta)$-DP with respect to these users.

Next, consider the $1/10$th of users with the least data. These users participate in lines (2) ($(\epsilon, \delta)$-DP) and (5) ($\epsilon$-DP). Again using the simple composition rule of differential privacy, Algorithm 3 is $(2\epsilon, \delta)$-DP with respect to these users.

Finally, let us consider the the group consisting of users $i \in [\log n + 1, 9n/10]$. These users first participate in line (5) ($\epsilon$-DP). The post-processing inequality of differential privacy states that we can now use these statistics in the subsequent computations without paying additionally for their privacy. Lines (7) - (9) are pre-processing for the computation of $\tilde{N}$. The algorithm releasing $\tilde{N}$ is a simple application of the Laplace mechanism since each $v_i \in [0, \frac{1}{\widehat{\sigma_{\min}}^2}]$, and hence is $\epsilon$-differentially private. The computation of $\Lambda$ in line (13) does not additionally touch the users data. The final estimate $\widehat{p}_\epsilon^{\text{realistic}}$ is an application of the propose-test-release framework on the function $\mathcal{M}(\cdot ; \widehat{k_T}, n, \widehat{p}_\epsilon^{\text{initial}}, \widehat{\sigma}_p^2)$ with proposed sensitivity $\Lambda$. This is a generic application of the propose-test-release framework, so we refer the reader to [10] for a proof that this final step of the algorithm is $(\epsilon, \delta)$-differentially private. Therefore, again using the composition theorem, Algorithm 3 is $(3\epsilon, 2\delta)$-DP with respect to this final set of users.

For the remainder of this section, we will focus on outlining the proof of the utility claims. Let us focus first on the impact of the use of propose-test-release (PTR). The two relevant components for the how the PTR component of Algorithm 3 affects the utility are the scale of $\Lambda/\epsilon$ and the probability that the proposed sensitivity is too small resulting in the algorithm ending in line (19), rather than line (22). The impact of the former is easy to analyse since the noise added is simply output perturbation. In order to show that the PTR ends in line (22) with high probability, we need to show that with high probability (over the randomness in the samples), $\kappa^*$ as defined in line (16) is large enough. Since this claim is really about $\mathcal{M}(\cdot; k_{\max}, n, \hat{p}, \hat{\sigma}_p^2)$, we will state and prove this claim in the notation of Algorithm 4.

**Lemma D.2.** *Given* $k_{\min} < k_{\max}$, $n \in \mathbb{N}$, $\hat{p} \in [0,1]$, $\hat{\sigma}_p^2 \in [0,1]$, $\Upsilon \in \mathbb{N}$, *and a database* $D = \{(\widehat{p}_i, k_i)\}_{i=1}^n$ *such that* $\widehat{p}_i \sim \mathcal{D}(k_i)$. *If the conditions of Theorem D.1 hold then with probability* $1 - \beta$, *D is such that for any $\kappa$-neighbour of D, $D'$ where $0 \le \kappa \le \Upsilon$ then*

$$\Delta(\mathcal{M}(\cdot; k_{\max}, m, \hat{p}, \hat{\sigma}_p^2, \alpha); D') \le 8B \frac{v_{k_{\max}} f_{\mathcal{D}}^{k_{\max}}(n, \hat{\sigma}_p^2, \beta)}{\sum_{i=1}^n v_i},$$

*where* $B = \sqrt{\Upsilon + 1} \log n + \frac{\log(1/\beta)}{\sqrt{\Upsilon + 1}}$.

Up to the factor $B$, the local sensitivity is exactly what we expect from the Algorithm 2. In order for our application of propose-test-release in Algorithm 3 to end in the desired result, we need to set $\Upsilon = \frac{\log(1/\delta)}{\epsilon} + \frac{\ln(1/\delta)\ln(1/\gamma)}{\epsilon}$. This means that up to logarithmic terms, the additional factor $B$ is $O(1/\sqrt{\epsilon})$. If $\epsilon$ is constant then this factor is also constant, although it may be a notable factor when $\epsilon$ is small. This inclusion of this extra factor is actually a result of a union bound in the proof of Lemma D.2, and we leave as an open question whether this factor can be improved or perhaps removed entirely.

*Proof of Lemma D.2.* Let $\sigma_{\max}^2 = \frac{1}{k_{\min}}(\hat{p} - (\hat{p})^2) + (1 - \frac{1}{k_{\min}})\hat{\sigma}_p^2$, $\sigma_{\min}^2 = \frac{1}{k_{\max}}(\hat{p} - (\hat{p})^2) + (1 - \frac{1}{k_{\max}})\hat{\sigma}_p^2$, $v_{\max} = 1/\sigma_{\min}^2$ and $v_{\min} = 1/\sigma_{\max}^2$, and $A = \frac{1}{2}\frac{n - \Upsilon - 1}{\Upsilon + 1}$. Note that as in Equation (7), the condition that $k_{\max}/k_{\min} \le A$ implies that $\sigma_{\max}^2 \le A\sigma_{\min}^2$ and, equivalently, $v_{\max} \le Av_{\min}$.

Let $D = \{(\widehat{p}_i, k_i)\}_{i=1}^n$ be a dataset of size $n$ where each $\widehat{p}_i \sim \mathcal{D}(k_i)$ where $\mathcal{D}$ has mean $p$ and variance $\sigma_p^2$. It suffices to show that for any database $D'$, which is a $\kappa$-neighbour of $D$ where $0 \le \kappa \le \Upsilon + 1$, and any $j \in [n]$, if $D'_{-j}$ is $D'$ where the data of the $j$th data subject has been removed, then,

$$\left| \mathcal{M}(D'; k_{\max}, n, \hat{p}, \hat{\sigma}_p^2, \alpha) - \mathcal{M}(D'_{-j}; k_{\max}, n, \hat{p}, \hat{\sigma}_p^2, \alpha) \right|$$
$$\le 4 \left( \sqrt{\Upsilon + 1} \log n + \frac{\log(1/\beta)}{\sqrt{\Upsilon + 1}} \right) \frac{v_{k_{\max}} f_{\mathcal{D}}^{k_{\max}}(n, \hat{\sigma}_p^2, \beta)}{\sum_{i=1}^n v_i}. \tag{12}$$

The final result is then a simple application of the triangle inequality.

Our proof that Equation (12) holds with high probability for all $\kappa$-neighbours of $D$ relies on the fact that the data points in $D$ are samples from $\mathcal{D}(k_i)$. In particular, it will rely on the fact that with

probability $1 - \beta$, $D$ is such that all subsets $S$ of $D$ of size at least $n - \Upsilon - 1$, $\mathcal{M}(S; k_{\max}, n, \hat{p}, \hat{\sigma_p}^2)$ is concentrated around $p$. Let $I$ be a subset of $[n]$ of size $n - \kappa$ where $\kappa \leq \Upsilon + 1$. Then

$$
\begin{aligned}
\mathrm{Var}(\mathcal{M}(S; k_{\max}, n, \hat{p}, \hat{\sigma_p}^2, \alpha)) &= \mathrm{Var}\left( \frac{\sum_{i \in I} v_i \widehat{p}_i}{\sum_{i \in I} v_i} \right) \\
&= \frac{\sum_{i \in I} \frac{1}{(\hat{\sigma}_i^2)^2} \sigma_i^2}{\left( \sum_{i \in I} \frac{1}{\hat{\sigma}_i^2} \right)^2} \\
&\leq 2 \frac{1}{\sum_{i \in I} \frac{1}{\hat{\sigma}_i^2}} \\
&\leq 2 \frac{1}{\frac{n - \kappa}{A} \frac{1}{\sigma_{\min}^2}} \\
&= \frac{2A}{n - \kappa} \sigma_{\min}^2 \\
&\leq \frac{\sigma_{\min}^2}{\Upsilon + 1}
\end{aligned}
$$

where the first inequality follows from $\sigma_i^2 \leq 2\hat{\sigma}_i^2$ (by Lemma 4.2), the second from the fact that $\hat{\sigma}_i^2 \leq A\sigma_{\min}^2$ for all $i \in [n]$ and the final from the definition of $A$. Let $\Gamma = \sum_{\kappa=0}^{\Upsilon+1} \binom{n}{\kappa}$ be the number of subsets of $D$ of size greater than $n - \upsilon - 1$. Therefore, by the concentration on the assumption of $\mathcal{M}(S; k_{\max}, n, \hat{p}, \hat{\sigma_p}^2)$, with probability $1 - \frac{\beta}{\Gamma}$,

$$
\left| \mathcal{M}(S; k_{\max}, n, \hat{p}, \hat{\sigma_p}^2, \alpha)) - p \right| \leq \frac{2\sigma_{\min} \log \frac{\Gamma}{\beta}}{\sqrt{\Upsilon + 1}} \tag{13}
$$

Applying a union bound, with probability $1 - \beta$, eqn (13) holds simultaneously for all subsets of $D$ of sufficiently large size. For the remainder of the proof, let us assume that this holds.

Let $D'$ be a $\kappa$-neighbour of $D$ where $0 \leq \kappa \leq \Upsilon + 1$. Without loss of generality, assume that $D' = \{(\widehat{p}_i', k_i')\}_{i=1}^n$ where $(\widehat{p}_i', k_i') = (\widehat{p}_i, k_i)$ for $i \in [n - k]$. In order to use this simplification, we will not assume that the $k_i'$ are in descending order. Let the $v_i$ be the un-normalised weights corresponding to $D'$, as defined in line (8) of Algorithm 4. Note that the $v_i$ depends only on the data of user $i$, not the data of any other individual in the data set. Then

$$
\begin{aligned}
\left| \mathcal{M}(D'; k_{\max}, n, \hat{p}, \hat{\sigma_p}^2, \alpha) - \mathcal{M}(D_{-j}'; k_{\max}, n, \hat{p}, \hat{\sigma_p}^2, \alpha) \right| &= \left| \frac{\sum_{i=1}^n v_i \widehat{p}_i'}{\sum_{i=1}^n v_i} - \frac{\sum_{i=1,i\neq j}^n v_i \widehat{p}_i'}{\sum_{i=1,i\neq j}^n v_i} \right| \\
&= \frac{v_j}{\sum_{i=1}^n v_i} \left| \widehat{p}_j' - \frac{\sum_{i=1,i\neq j}^n v_i \widehat{p}_i'}{\sum_{i=1,i\neq j}^n v_i} \right| \\
&\leq \frac{v_j}{\sum_{i=1}^n v_i} \left( \left| \widehat{p}_j' - \frac{\sum_{i=1}^{n-\kappa} v_i \widehat{p}_i}{\sum_{i=1}^{n-\kappa} v_i} \right| + \left| \frac{\sum_{i=1}^{n-\kappa} v_i \widehat{p}_i}{\sum_{i=1}^{n-\kappa} v_i} - \frac{\sum_{i=1,i\neq j}^n v_i \widehat{p}_i'}{\sum_{i=1,i\neq j}^n v_i} \right| \right).
\end{aligned}
\tag{14}
$$

We will bound the two terms separately. For the first term in Equation (14), we will use the fact that $\frac{\sum_{i=1}^{n-\kappa} v_i \widehat{p}_i'}{\sum_{i=1}^{n-\kappa} v_i}$ is concentrated around $p$, and $\hat{p}_j'$ is truncated to within $\alpha + f_{\mathcal{D}}^{k_j'}(n, \sigma_p^2, \beta)$ of $p$. So

$$
\left| \widehat{p}_j' - \frac{\sum_{i=1}^{n-\kappa} v_i \widehat{p}_i}{\sum_{i=1}^{n-\kappa} v_i} \right| \leq \max\left\{ 2(\alpha + f_{\mathcal{D}}^{k_j'}(n, \sigma_p^2, \beta)), \frac{2\sigma_{\min} \log \frac{\Gamma}{\beta}}{\sqrt{\Upsilon + 1}} \right\} \leq 4 f_{\mathcal{D}}^{k_j'}(n, \sigma_p^2, \beta) \frac{\log \frac{\Gamma}{\beta}}{\sqrt{\Upsilon + 1}},
$$

where the second inequality follows since $\max\{\alpha, \sigma_{\min}\} \leq f_{\mathcal{D}}^{k_j'}(n, \sigma_p^2, \beta)$.

Next, let us handle the second term in Equation (14):

$$\left| \frac{\sum_{i=1}^{n-1} v_i \widehat{p_i'}}{\sum_{i=1}^{n-1} v_i} - \frac{\sum_{i=1}^{n-\kappa} v_i \widehat{p_i'}}{\sum_{i=1}^{n-\kappa} v_i} \right| = \left| \frac{\sum_{i=n-\kappa+1}^{n-1} v_i}{\sum_{i=1}^{n-1} v_i} \left( \frac{\sum_{i=n-\kappa+1}^{n-1} v_i \widehat{p_i'}}{\sum_{i=n-\kappa+1}^{n-1} v_i} - \frac{\sum_{i=1}^{n-\kappa} v_i \widehat{p_i'}}{\sum_{i=1}^{n-\kappa} v_i} \right) \right|$$

$$\leq \left( \frac{\sum_{i=n-\kappa+1}^{n-1} v_i}{\sum_{i=1}^{n-1} v_i} \right) \left( \frac{\sum_{i=n-\kappa+1}^{n-1} v_i \left| \widehat{p_i'} - \frac{\sum_{i=1}^{n-\kappa} v_i \widehat{p_i'}}{\sum_{i=1}^{n-\kappa} v_i} \right|}{\sum_{i=n-\kappa+1}^{n-1} v_i} \right)$$

$$\leq \left( \frac{\sum_{i=n-\kappa+1}^{n-1} v_i}{\sum_{i=1}^{n-1} v_i} \right) \left( \frac{\sum_{i=n-\kappa+1}^{n-1} v_i \max\{(2\alpha + 2f_{\mathcal{D}}^{k_i'}(n, \widehat{\sigma}_p^2, \beta)), 2\sigma_{\min} \log \frac{\Gamma}{\beta}\}}{\sum_{i=n-\kappa+1}^{n-1} v_i} \right)$$

$$\leq \left( \frac{\sum_{i=n-\kappa+1}^{n-1} v_i}{\sum_{i=1}^{n-1} v_i} \right) \left( \frac{\sum_{i=n-\kappa+1}^{n-1} v_i 4 f_{\mathcal{D}}^{k_i'}(n, \widehat{\sigma}_p^2, \beta) \frac{\log \frac{\Gamma}{\beta}}{\sqrt{\Upsilon+1}}}{\sum_{i=n-\kappa+1}^{n-1} v_i} \right)$$

$$\leq \left( \frac{4 \sum_{i=n-\kappa+1}^{n-1} v_i f_{\mathcal{D}}^{k_i'}(n, \widehat{\sigma}_p^2, \beta) \frac{\log \frac{\Gamma}{\beta}}{\sqrt{\Upsilon+1}}}{\sum_{i=1}^{n-1} v_i} \right)$$

$$\leq 4\kappa \frac{\log \frac{\Gamma}{\beta}}{\sqrt{\Upsilon+1}} \left( \frac{\max_i v_i f_{\mathcal{D}}^{k_i'}(n, \widehat{\sigma}_p^2, \beta)}{\sum_{i=1}^{n-1} v_i} \right)$$

Since $v_i = \frac{1}{\sigma_{k_i'}^2}$ and $f_{\mathcal{D}}^{k_i'}(n, \sigma_p^2, \beta) \leq 2\sigma_{k_i'}^2 \log n$, $\max_i v_i f_{\mathcal{D}}^{k_i'}(n, \sigma_p^2, \beta) \leq v_{k_{\max}} f_{\mathcal{D}}^{k_{\max}}(n, \sigma_p^2, \beta)$.
Also, $\sigma_{k_i'}^2 \leq A \sigma_{k_{\max}}^2$ so we have $\sum_{i=1}^{n-1} v_i \geq \frac{n-1}{A} v_{k_{\max}}$. Therefore,

$$\left| \frac{\sum_{i=1}^{n-1} v_i \widehat{p_i'}}{\sum_{i=1}^{n-1} v_i} - \frac{\sum_{i=1}^{n-\kappa} v_i \widehat{p_i'}}{\sum_{i=1}^{n-\kappa} v_i} \right| \leq 4\kappa \frac{\log \frac{\Gamma}{\beta}}{\sqrt{\Upsilon+1}} \frac{v_{k_{\max}} f_{\mathcal{D}}^{k_{\max}}(n, \sigma_p^2, \beta)}{\frac{n-1}{A} v_{k_{\max}}}$$

$$\leq \frac{4\sqrt{\Upsilon+1} A \log \frac{\Gamma}{\beta}}{n-1} f_{\mathcal{D}}^{k_{\max}}(n, \sigma_p^2, \beta)$$

$$\leq 2 \frac{\log \frac{\Gamma}{\beta}}{\sqrt{\Upsilon+1}} f_{\mathcal{D}}^{k_{\max}}(n, \sigma_p^2, \beta)$$

where the second inequality follows from $\kappa \leq \Upsilon+1$ and the third inequality follows from $A \leq \frac{n-1}{2(\Upsilon+1)}$ by assumption. Therefore,

$$\left| \mathcal{M}(D'; k_{\max}, n, \hat{p}, \hat{\sigma_p}^2, \alpha) - \mathcal{M}(D'_{-j}; k_{\max}, n, \hat{p}, \hat{\sigma_p}^2, \alpha) \right|$$

$$\leq \frac{v_j}{\sum_{i=1}^n v_i} \cdot 4 \frac{\log \frac{\Gamma}{\beta}}{\sqrt{\Upsilon+1}} (f_{\mathcal{D}}^{k_j'}(n, \hat{\sigma_p}^2, \beta) + f_{\mathcal{D}}^{k_{\max}}(n, \hat{\sigma_p}^2, \beta))$$

$$\leq 4 \frac{\log \frac{\Gamma}{\beta}}{\sqrt{\Upsilon+1}} \frac{v_j}{\sum_{i=1}^n v_i} \cdot f_{\mathcal{D}}^{k_j'}(n, \hat{\sigma_p}^2, \beta)$$

$$\leq 4 \frac{\log \frac{\Gamma}{\beta}}{\sqrt{\Upsilon+1}} \frac{v_j}{\sum_{i=1}^n v_i} \cdot f_{\mathcal{D}}^{k_j'}(n, \hat{\sigma_p}^2, \beta)$$

Note that $\Gamma \leq n^{\Upsilon+1}$. Taking the max over $j$, we again have that $\max_j v_j f_{\mathcal{D}}^{k'_j}(n, \hat{\sigma_p}^2, \beta) \leq v_{k_{\max}} f_{\mathcal{D}}^{k_{\max}}(n, \hat{\sigma_p}^2, \beta)$ so for all $j$,

$$\left| \mathcal{M}(D'; k_{\max}, n, \hat{p}, \hat{\sigma_p}^2, \alpha) - \mathcal{M}(D'_{-j}; k_{\max}, n, \hat{p}, \hat{\sigma_p}^2, \alpha) \right|$$

$$\leq 4 \frac{\log \frac{\Gamma}{\beta}}{\sqrt{\Upsilon + 1}} \frac{v_{k_{\max}} f_{\mathcal{D}}^{k_{\max}}(n, \hat{\sigma_p}^2, \beta)}{\sum_{i=1}^n v_i}$$

$$\leq 4 \left( \sqrt{\Upsilon + 1} \log n + \frac{\log(1/\beta)}{\sqrt{\Upsilon + 1}} \right) \frac{v_{k_{\max}} f_{\mathcal{D}}^{k_{\max}}(n, \hat{\sigma_p}^2, \beta)}{\sum_{i=1}^n v_i}$$

$\square$

*Proof of Theorem D.1.* The main component remaining to prove is that truncating at $\frac{1}{\widehat{\sigma_{\min}}^2}$ rather than the optimal truncation does not affect the utility by more than a constant factor, provided $\epsilon$ is not too small. Recall $k_1 \geq k_2 \geq \cdots \geq k_n$. Firstly, we need to show that $\widehat{k_T}$ is a sufficiently good estimate of $k_{\frac{2 \log n}{\epsilon}}$. A careful instantiation of the exponential mechanism provides us with a $\epsilon$-DP estimator of the $\frac{2 \log n}{\epsilon}$-th order statistic that has the guarantee that with probability $1 - \beta$,

$k_{\frac{2 \log n}{\epsilon} + \frac{1}{\epsilon}(\ln k_{\max} + \ln(1/\beta))} \leq \widehat{k_T} \leq k_{\frac{2 \log n}{\epsilon} - \frac{1}{\epsilon}(\ln k_{\max} + \ln(1/\beta))}$. Since by assumption $k \leq \sqrt{n}$ and $\beta \leq 1/\sqrt{n}$, this implies that with probability $1 - \beta$, $k_{\frac{3 \log n}{\epsilon}} \leq \widehat{k_T} \leq k_{\frac{\log n}{\epsilon}}$. That is, only $\frac{\log n}{\epsilon}$ more data points than desired will be truncated.

Next, we need to show that truncating at any point within this range provides an estimator with accuracy competitive with the optimal truncation. The variance of $\hat{p}_\epsilon^{\text{realistic, unknown size}}$ can be written as two terms, the variance that exists in the non-private setting, and the additional noise due to privacy;

$$\text{Var}(\widehat{p}_\epsilon^{\text{realistic}}) = \underbrace{\frac{\sum_{i=\log n+1}^{9n/10} \min\left\{ \frac{T^2}{\widehat{\sigma}_i^2}, \frac{1}{\widehat{\sigma}_i^4} \right\} \text{Var}([\widehat{p}_i]_{\widehat{a}_i}^{\widehat{b}_i})}{\left( \sum_{i=\log n+1}^{9n/10} \min\left\{ \frac{T}{\widehat{\sigma}_i}, \frac{1}{\widehat{\sigma}_i^2} \right\} \right)^2}}_{\text{non-private term}} + \underbrace{\frac{\left( 8 \left( \sqrt{\Upsilon + 1} \log n + \frac{\log(1/\beta)}{\sqrt{\Upsilon+1}} \right) \frac{f_{\mathcal{D}}^{\widehat{k_T}}(n, \hat{\sigma_p}^2, \beta)}{\widehat{\sigma_{\min}}^2 \widehat{N}} \right)^2}{\epsilon^2}}_{\text{private term}},$$

where $\Upsilon = \frac{\log(1/\delta)}{\epsilon} + \frac{\ln(1/\delta)\ln(1/\gamma)}{\epsilon}$. The truncation has opposite effects on each of these terms. As $T$ increases, the private term decreases while the non-private term increases. When we set $T = 1/\text{Var}(\mathcal{D}(k_{\frac{2 \log n}{\epsilon} + K}))$, where $K \in [-\frac{\log n}{\epsilon}, \frac{\log n}{\epsilon}]$ then if $K$ is negative, no truncation occurs and the non-private term is optimal. If $K$ is positive then only a small number of data points are truncated so the non-private term is close to it's optimal value:

$$\frac{\sum_{i=\log n+1}^{9n/10} \min\left\{ \frac{T^{*2}}{\widehat{\sigma}_i^2}, \frac{1}{\widehat{\sigma}_i^4} \right\} \text{Var}([\widehat{p}_i]_{\widehat{a}_i}^{\widehat{b}_i})}{\left( \sum_{i=\log n+1}^{9n/10} \min\left\{ \frac{T^*}{\widehat{\sigma}_i}, \frac{1}{\widehat{\sigma}_i^2} \right\} \right)^2} \leq O\left( \frac{\sum_{i=\log n+K}^{9n/10} \frac{1}{\widehat{\sigma}_i^4} \text{Var}([\widehat{p}_i]_{\widehat{a}_i}^{\widehat{b}_i})}{\left( \sum_{i=\log n+K}^{9n/10} \frac{1}{\widehat{\sigma}_i^2} \right)^2} \right)$$

$$\leq O\left( \frac{\sum_{i=\log n+1}^{9n/10} \frac{1}{\widehat{\sigma}_i^4} \text{Var}([\widehat{p}_i]_{\widehat{a}_i}^{\widehat{b}_i})}{\left( \sum_{i=\log n+1}^{9n/10} \frac{1}{\widehat{\sigma}_i^2} \right)^2} \right)$$

where the first inequality follows from the fact that if $k_{\max}/k_{\text{med}} \leq \frac{n/2 - \frac{\log n}{\epsilon}}{\frac{\log n}{\epsilon}} \leq \frac{n/2 - \frac{2 \log n}{\epsilon} - K}{\frac{2 \log n}{\epsilon} - K}$ and $\frac{\log n}{\epsilon} \leq n/2$ then deleting $K$ points has only a constant factor impact on the variance (this argument is identical to that made in eqn (10) in the proof of Theorem 4.1). The second inequality follows from the fact that adding more high quality data points only improves the variance of the estimator. Therefore, the non-private term in the variance is within a constant factor of optimal.

Next, we will show under the conditions outlines in the theorem, the non-private term is dominates the variance. The normalisation term also appears in the private term but as an approximation:

$$\widehat{N} = \sum_{j=\log n+1}^{n/2} \min\left\{ \frac{T}{\widehat{\sigma}_i}, \frac{1}{\widehat{\sigma}_i^2} \right\} + \text{Lap}\left( \frac{1}{\epsilon \widehat{\sigma_{\min}}^2} \right) - \frac{1}{\epsilon \widehat{\sigma_{\min}}^2} \ln(2\delta).$$

With probability $1 - \beta$,

$$\tilde{N} \geq \sum_{j=\log n+1}^{9n/10} \min\left\{\frac{T}{\widehat{\sigma}_i}, \frac{1}{\widehat{\sigma}_i^2}\right\} - 2\frac{1}{\epsilon\widehat{\sigma_{\min}}^2}\ln(2\delta)$$

$$\geq \sum_{j=\log n+1}^{n/4} \min\left\{\frac{T}{\widehat{\sigma}_i}, \frac{1}{\widehat{\sigma}_i^2}\right\} + \sum_{j=n/4+1}^{n/2} \min\left\{\frac{T}{\widehat{\sigma}_i}, \frac{1}{\widehat{\sigma}_i^2}\right\} - 2\frac{1}{\epsilon\widehat{\sigma_{\min}}^2}\ln(2\delta)$$

$$\geq \sum_{j=\log n+1}^{n/4} \min\left\{\frac{T}{\widehat{\sigma}_i}, \frac{1}{\widehat{\sigma}_i^2}\right\} + (n/4-1)\frac{1}{\widehat{\sigma}_{k_{\text{med}}}^2} - 2\frac{1}{\epsilon\widehat{\sigma_{\min}}^2}\ln(2\delta)$$

$$\geq \sum_{j=\log n+1}^{n/4} \min\left\{\frac{T}{\widehat{\sigma}_i}, \frac{1}{\widehat{\sigma}_i^2}\right\}$$

$$\geq \frac{1}{2}\sum_{j=\log n+1}^{9n/10} \min\left\{\frac{T}{\widehat{\sigma}_i}, \frac{1}{\widehat{\sigma}_i^2}\right\}$$

where the final inequality comes from high probability bounds on the Laplacian distribution, the second inequality is simply separating the sum into two pieces and removing the contribution of users $i \in [n/2+1, 9n/10]$, the third inequality comes from the fact that any user with more than $k_{\text{med}}$ data points has weight larger than $1/\widehat{\sigma}_{k_{\text{med}}}^2$. The fourth inequality follows from $\frac{\widehat{\sigma}_{k_{\text{med}}}}{\widehat{\sigma_{\min}}^2} \leq \frac{(n/4-1)\epsilon}{3\ln(2/\delta)}$. Now, let us turn to the proof that the non-private noise is dominant when $\epsilon$ is not too large. To see this note that the non-private term satisfies

$$\frac{\sum_{i=\log n+1}^{9n/10} \min\left\{\frac{T^2}{\widehat{\sigma}_i^2}, \frac{1}{\widehat{\sigma}_i^4}\right\} \text{Var}([\widehat{p}_i]_{\widehat{a}_i}^{\widehat{b}_i})}{\left(\sum_{i=\log n+1}^{9n/10} \min\left\{\frac{T}{\widehat{\sigma}_i}, \frac{1}{\widehat{\sigma}_i^2}\right\}\right)^2} \geq \Omega\left(\frac{\sum_{i=\log n+1}^{9n/10} \min\{T^2, \frac{1}{\sigma_i^2}\}}{\left(\sum_{i=\log n+1}^{9n/10} \min\{\frac{T}{\sigma_i}, \frac{1}{\sigma_i^2}\}\right)^2}\right)$$

where the inequality comes from $\text{Var}([\widehat{p}_i]_{\widehat{a}_i}^{\widehat{b}_i}) \geq \frac{1}{2}\sigma_i^2$ and $\widehat{\sigma}_i^2$ is within a constant multiplicative factor of $\sigma_i^2$. Further, the private term satisfies

$$\frac{\left(8((\Upsilon+1)\log n + \log(1/\beta))\frac{f_{\mathcal{D}}^{\widehat{k_T}}(n,\widehat{\sigma_p}^2,\beta)}{\widehat{\sigma_{\min}}^2\widehat{N}}\right)^2}{\epsilon^2} = O\left(\frac{(\log n \log(1/\delta)\log(1/\beta))^2}{\sigma_{\min}^2 N^2 \epsilon^3}\right)$$

$$= O\left(\frac{(\log n \log(1/\delta)\log(1/\beta))^2}{\sigma_T^2 \epsilon^3 (\sum_{i=\log n+1}^{9n/10} \min\{\frac{T}{\sigma_i}, \frac{1}{\sigma_i^2}\})^2}\right)$$

Now, comparing these two terms we can see that the non-private term dominates when:

$$\frac{\sum_{i=\log n+1}^{9n/10} \min\{T, \frac{1}{\sigma_i^2}\}}{\left(\sum_{i=\log n+1}^{9n/10} \min\{\frac{T}{\sigma_i}, \frac{1}{\sigma_i^2}\}\right)^2} = \Omega\left(\frac{(\log n \log(1/\delta)\log(1/\beta))^2}{\sigma_T^2 \epsilon^3 (\sum_{i=\log n+1}^{9n/10} \min\{\frac{T}{\sigma_i}, \frac{1}{\sigma_i^2}\})^2}\right).$$

That is, when:

$$\sum_{i=\log n+1}^{9n/10} \min\left\{T, \frac{1}{\sigma_i^2}\right\} \geq \left(\frac{\log n \log(1/\delta)\log(1/\beta)}{\sigma_{\min}\epsilon^{1.5}}\right)^2.$$

Noting that,

$$\sum_{i=\log n+1}^{9n/10} \min\left\{T^*, \frac{1}{\sigma_i^2}\right\} \geq (n/2 - \log n - 1)\frac{1}{\sigma_{\text{med}}^2} \geq \left(\frac{\log n \log(1/\delta)\log(1/\beta)}{\sigma_{\min}\epsilon^{1.5}}\right)^2$$

where the first inequality is simply because more than $(n/2 - \log n)$ of the user have weight larger than the median weight, and the second inequality follows from the assumption that $\frac{k_{\max}}{k_{\text{med}}} \leq \frac{\epsilon^3(n/2 - \log n - 1)}{\log n \log(1/\delta) \log(1/\beta)}$. Therefore, with high probability (based on the accuracy of $\widehat{k_T}$), truncating at $1/\sigma_{\min}^2$ rather than the optimal truncation $T$ does not affect the variance of the estimator by more than a constant factor.

Now that we have established that the noise added for privacy is not too large, the only remaining potential point of failure for the algorithm is that the PTR component fails and the algorithm outputs $\widehat{p}_\epsilon^{\text{initial}}$ rather than the more accurate weighted estimate. The fact that this does not happen with high probability is a direct corollary of Lemma D.2. $\qquad\square$

# E  Proofs from Section 5

**Lemma 5.2.** *Given* $\widehat{p}_i \sim \mathcal{D}_p(k_i)$ *with variance* $\sigma_i^2$ *for all* $i \in [n]$ *and* $w \in [0,1]^n$ *such that* $\sum_{i=1}^n w_i = 1$, *let* $\widehat{p} = \sum_{i=1}^n w_i \widehat{p}_i + \text{Lap}(\frac{\max_i w_i \sigma_i}{\epsilon})$. *The variance of* $\widehat{p}$ *is minimized by the following weights:* $\tilde{w}_i^* = \frac{\min\{1/\sigma_i^2, T/\sigma_i\}}{\sum_{j=1}^n \min\{1/\sigma_j^2, T/\sigma_j\}}$ *for some* $T$.

*Proof of Lemma 5.2.* Let

$$w^* = \arg\min_{\substack{w \in [0,1]^n \\ \sum_{i=1}^n w_i = 1}} \text{Var}(\widehat{p}) = \arg\min_{\substack{w \in [0,1]^n \\ \sum_{i=1}^n w_i = 1}} \sum_{i=1}^n w_i^2 \sigma_i^2 + \frac{\max_k w_k^2 \sigma_k^2}{\epsilon^2}$$

be an optimal weight vector that minimizes variance of $\widehat{p}$. We start with a few observations on structural properties of the optimal weights. Let $M = \{\arg\max_k w_k^* \sigma_k\}$ be the set of all users with maximum weighted-variance contribution to the estimate $\widehat{p}$.

First, notice that for all $i, j \in [n]$, if $w_i^* > w_j^*$ then $\sigma_i^2 \leq \sigma_j^2$. This follows since if $\sigma_i^2 > \sigma_j^2$ then $w_i^* \sigma_j^2 + w_j^* \sigma_i^2 < w_i^* \sigma_i^2 + w_j^* \sigma_j^2$ and $\max\{w_i^* \sigma_j^2, w_j^* \sigma_i^2\} \leq w_i^* \sigma_i$ which implies that swapping the weights of $i$ and $j$ would result in an estimator with lower variance. This is a contradiction given the definition of $w^*$.

Next, we show that if $i, j \notin M$ then $w_i^* \sigma_i^2 = w_j^* \sigma_j^2$. Suppose towards a contradiction that $w_i^* \sigma_i^2 < w_j^* \sigma_j^2$. Let $\alpha = \min\{\frac{w_j^* \sigma_j^2 - w_i^* \sigma_i^2}{\sigma_i^2 + \sigma_j^2}, \frac{\max_k w_k^* \sigma_k - w_i^* \sigma_i}{\sigma_i}, w_j^*\}$. Then $\alpha > 0$, and $(w_j^* - \alpha)\sigma_j, (w_i^* + \alpha)\sigma_i \in [0, \max_k w_k^* \sigma_k]$. Also,

$$\begin{aligned}
(w_j^* - \alpha)^2 \sigma_j^2 + (w_i^* + \alpha)^2 \sigma_i^2 &= w_j^{*2} \sigma_j^2 + w_i^{*2} \sigma_i^2 + \alpha^2(\sigma_i^2 + \sigma_j^2) - 2\alpha(w_j^* \sigma_j^2 - w_i^* \sigma_i^2) \\
&= w_j^{*2} \sigma_j^2 + w_i^{*2} \sigma_i^2 + \alpha(\alpha(\sigma_i^2 + \sigma_j^2) - 2(w_j^* \sigma_j^2 - w_i^* \sigma_i^2)) \\
&< w_j^{*2} \sigma_j^2 + w_i^{*2} \sigma_i^2.
\end{aligned}$$

This implies that shifting $\alpha$ weight from $w_i^*$ to $w_j^*$ would reduce the variance of the estimator $\widehat{p}$ without changing the maximum weighted-variance, which is a contradiction of the optimality of $w^*$.

Define $H = \max_k w_k^* \sigma_k$ and note that there exists $R > 0$ such that $w_i^* = R/\sigma_i^2$ for all $i \notin M$. From these observations, there must exist some threshold $T$ such that if $\sigma_i \geq 1/T$, then $w_i^* = R/\sigma_i^2$, and if $\sigma_i < 1/T$, then $w_i^* = H/\sigma_i$. By continuity, $H = RT$, and we can write the optimal weights as: $w_i^* = \min\{1/\sigma_i^2, T/\sigma_i\}R$. Since the weights $w_i^*$ must sum to 1, we know that $R = \frac{1}{\sum_{j=1}^n \min\{1/\sigma_j^2, T/\sigma_j\}}$.

Thus the optimal weights are:

$$w_i^* = \frac{\min\{1/\sigma_i^2, T/\sigma_i\}}{\sum_{j=1}^n \min\{1/\sigma_j^2, T/\sigma_j\}},$$

for some appropriate threshold $T$.

$\qquad\square$

Let us recall some notation. Let $\mathcal{P}$ be a parameterized family of distributions $p \mapsto \mathcal{D}_p$, so $\mathbb{E}[\mathcal{D}_p]$. Given an estimator $M$, vector $\boldsymbol{q} \in [0,1]^n$ and set $I \subset [n]$, let

$$\mu_M(x_{[n] \setminus I}; \boldsymbol{q}) = \mathbb{E}_{\forall i \in I, x_i \sim \mathcal{D}_{q_i}(k_i), M}[M(x_1, \cdots, x_n)]$$

be the expectation taken only over the randomness of $I$ and $M$. Note that in this notation, user $i$ is sampling from a meta-distribution with mean $q_i$, which may be different for each user. We will abuse notation slightly and for $p \in [0,1]$, we will let $\mu_M(x_{[n]\setminus I}; p) = \mu_M(x_{[n]\setminus I}; (p, \cdots, p))$. Let $\mu_M(\boldsymbol{q}) = \mu_M(\emptyset; \boldsymbol{q})$. When the estimator $M$ is clear from context, we will simply use the notation $\mu(x_{[n]\setminus I}; \boldsymbol{p})$. Recall that for $p \in [0,1]$ and $k \in \mathbb{N}$, $\phi_{p,k}$ is the probability density function of $\mathcal{D}_p(k)$.

**Lemma 5.3.** *Let $\mathcal{P}$ be a parameterized family of distributions $p \mapsto \mathcal{D}_p$ and suppose that $M : [0,1]^n \to [0,1]$ is an $\epsilon$-DP estimator such that for all $p \in [1/3, 2/3]$, (1) $M$ is unbiased, $\mu_M(p) = p$, and (2) the Fisher information of $\phi_{p,k_i}$ is inversely proportional to the variance $\mathrm{Var}(\mathcal{D}_p(k_i))$, $\int (\frac{\partial}{\partial p} \log \phi_{p,k_i}(x_i))^2 \phi_{p,k_i}(x_i) dx_i = O(\frac{1}{\mathrm{Var}(\mathcal{D}_p(k_i))})$, then there exists an estimator $M_{\mathtt{NL}} \in \mathtt{NLE}$ such that*

$\max_{p \in [1/3,2/3]}[\mathrm{Var}_{\forall i \in [n], x_i \sim \mathcal{D}(k_i), M_{\mathtt{NL}}}(M_{\mathtt{NL}})] \leq O(\max_{p \in [1/3,2/3]}[\mathrm{Var}_{\forall i \in [n], x_i \sim \mathcal{D}(k_i), M}(M)])$.

Before we formally prove Lemma 5.3, let us start with some intuition for the proof. Given an estimator $M_{\mathtt{NL}} \in \mathtt{NLE}$, the variance of $M_{\mathtt{NL}}$ can be written as

$$\mathrm{Var}(M_{\mathtt{NL}}) \leq \sum_{i=1}^{n} w_i^2 \mathrm{Var}(\mathcal{D}(k_i)) + O(\frac{\max w_i \sigma_i}{\epsilon})^2. \tag{15}$$

That is, it can be decomposed as the variance contribution of each individual coordinate, and the variance contribution of the additional noise due to privacy. Lemma E.1 (proved in Appendix E) shows that the variance of any estimator $M$ can be lower bounded by a similar decomposition. Since this involves considering the impact of each coordinate individually, the following notation will be useful. Given an estimator $M$, vector $\boldsymbol{q} \in [0,1]^n$ and set $I \subset [n]$, let $\mu_M(x_{[n]\setminus I}; \boldsymbol{q}) = \mathbb{E}_{\forall i \in I, x_i \sim \mathcal{D}_{q_i}(k_i), M}[M(x_1, \cdots, x_n)]$ be the expectation over only randomness in $I$ and $M$. Note that in this notation, user $i$ is sampling from a meta-distribution with mean $q_i$, which may be different for each user. We will abuse notation slightly to let $\mu_M(\boldsymbol{q}) = \mu_M(\emptyset; \boldsymbol{q})$, and for $p \in [0,1]$, we will let $\mu_M(x_{[n]\setminus I}; p) = \mu_M(x_{[n]\setminus I}; (p, \cdots, p))$. When the estimator $M$ is clear from context, we will omit it.

The following lemma is proved later, outside the proof of Lemma 5.3.

**Lemma E.1.** *For any randomised mechanism $M : [0,1]^n \to [0,1]$,*

$$\mathrm{Var}_{\forall i \in [n], x_i \sim \mathcal{D}_p(k_i), M}(M) \tag{16}$$
$$= \mathbb{E}_{\forall i \in [n], x_i \sim \mathcal{D}_p(k_i), M}[(M(x_1, ..., x_n) - \mu(p))^2]$$
$$\geq \sum_{i=1}^{n} \mathbb{E}_{x_i \sim \mathcal{D}_p(k_i)}[(\mu(x_i; p) - \mu(p))^2] + \mathbb{E}_{\forall i \in [n], x_i \sim \mathcal{D}_p(k_i), M}[(M(x_1, ..., x_n) - \mu(x_1, ..., x_n; p))^2]$$

In Equation (16), the first term is the sum of contributions to the variance of the individual terms $x_i$, and the second term is the contribution to the variance of the noise added for privacy. Now we want to define a weight vector $\mathbf{w}$ such that the terms in Equation (16) are lower bounded by the corresponding terms in Equation (15). The key component of the proof is the observation that if we let

$$w_i(p) = \frac{\partial}{\partial q_i} \mu(\boldsymbol{q}) \big|_{\boldsymbol{q} = (p, \cdots, p)} \tag{17}$$

then we can show that there exists a constant $c$ such that

$$\mathbb{E}_{x_i \sim \mathcal{D}_p(k_i)}[(\mu(x_i; p) - \mu(p))^2] \geq c \cdot w_i(p)^2 \mathrm{Var}(\mathcal{D}_p(k_i)). \tag{18}$$

This controls the contribution of each individual coordinate to the variance of $M$. It remains only to control the contribution of the noise due to privacy. We show that there exists $x_i$, $x_i'$ such that $|\mu(x_i; p) - \mu(x_i'; p)| \geq \Omega(w_i(p) \cdot \sqrt{\mathrm{Var}(\mathcal{D}_p(k_i))})$, which we show implies that,

$$\mathbb{E}_{\forall i \in [n], x_i \sim \mathcal{D}_p(k_i), M}[(M(x_1, \cdots, x_n) - \mu(x_1, \cdots, x_n; p))^2] \geq \Omega(\frac{w_i(p)^2 \mathrm{Var}(\mathcal{D}_p(k_i))}{\epsilon^2}). \tag{19}$$

Intuitively, the worst-case $|\mu(x_i; p) - \mu(x_i'; p)|$ plays an analogous role to the sensitivity, since it captures the impact of changing one user's data. Since $M$ is an $\epsilon$-DP mechanism and $|\mu(x_i; p) - \mu(x_i'; p)|$ is at least $\Omega(w_i(p) \cdot \sqrt{\mathrm{Var}(\mathcal{D}_p(k_i))})$, we show that it must include noise with standard deviation of at least this magnitude over $\epsilon$. This is consistent with, e.g., the Laplace Mechanism that adds noise with standard deviation $\Theta(\Delta f / \epsilon)$.

Combining Lemma E.1 with Equations (18) and (19) gives that the variance of $M$ is at least,
$\mathrm{Var}_{\forall i \in [n], x_i \sim \mathcal{D}_p(k_i), M}(M) \geq \sum_{i=1}^{n} c \cdot w_i(p)^2 \mathrm{Var}(\mathcal{D}_p(k_i)) + \Omega(\frac{w_i(p)^2 \mathrm{Var}(\mathcal{D}_p(k_i))}{\epsilon^2})$.

Finally, we must create a corresponding $M_{\text{NL}} \in \text{NLE}$ for comparison, using the same weights. Since $\sum_{i=1}^{n} w_i(p)$ as defined in Equation (17) need not equal 1, these weights will need to be normalized to sum to 1 to create an estimator in $\text{NLE}$. We need to show this normalisation does not substantially increase the variance of the resulting estimator. In order to show this, we show that there exists a $p^* \in [1/3, 2/3]$ such that $\sum_{i=1}^{n} w_i(p^*) \geq 1$, since normalizing the estimator by a factor of $\frac{1}{\sum_{i=1}^{n} w_i(p^*)}$ will affect the variance by a factor of $\frac{1}{(\sum_{i=1}^{n} w_i(p^*))^2}$, and thus if $\sum_{i=1}^{n} w_i(p^*) \geq 1$, then this will decrease variance. This desired fact follows from the definition of $w_i$, and the fact that $M$ is unbiased. Now, if we define

$$M_{\text{NL}}(\mathbf{x}) = \frac{\sum_{i=1}^{n} w_i(p^*) x_i + \text{Lap}(\frac{\max_i w_i(p^*) \sqrt{\text{Var}(\mathcal{D}_p(k_i))}}{\epsilon})}{\sum_{i=1}^{n} w_i(p^*)},$$

then $M_{\text{NL}} \in \text{NLE}$ and $\text{Var}_{\forall i \in [n], x_i \sim \mathcal{D}_p(k_i), M_{\text{TNL}}}(M_{\text{NL}}) = \Theta\left(\text{Var}_{\forall i \in [n], x_i \sim \mathcal{D}_p(k_i), M}(M)\right)$.

*Proof of Lemma 5.3.* We first apply Lemma E.1 to decompose the variance of the estimate computed by $M$ as:

$\text{Var}_{\forall i \in [n], x_i \sim \mathcal{D}_p(k_i), M}(M)$

$$\geq \sum_{i=1}^{n} \mathbb{E}_{x_i \sim \mathcal{D}_p(k_i)}[(\mu(x_i; p) - \mu(p))^2] + \mathbb{E}_{\forall i \in [n], x_i \sim \mathcal{D}_p(k_i), M}[(M(x_1, \cdots, x_n) - \mu(x_1, \cdots, x_n; p))^2]$$

The first term is the sum of contributions to the variance of the individual terms $x_i$, and the second term is the contribution to the variance of the noise added for privacy. We will proceed by bounding these terms separately, starting with the first term.

First note that by definition,

$$\int (\mu(x_i; \boldsymbol{q}) - \mu(\boldsymbol{q})) \phi_{q_i, k_i}(x_i) dx_i = \mathbb{E}_{x_i \sim \mathcal{D}_{q_i}(k_i)}[\mu(x_i; \boldsymbol{q})] - \mu(\boldsymbol{q}) = 0.$$

Therefore, by taking the partial derivative with respect to $q_i$ we have

$$\int \left[ \left( \frac{\partial}{\partial q_i} (\mu(x_i; \boldsymbol{q}) - \mu(\boldsymbol{q})) \right) \phi_{q_i, k_i}(x_i) + (\mu(x_i; \boldsymbol{q}) - \mu(\boldsymbol{q})) \frac{\partial}{\partial q_i} \phi_{q_i, k_i}(x_i) \right] dx_i = 0.$$

Note that $\mu(x_i; \boldsymbol{q})$ is constant in $q_i$ so rearranging, and noting that $\frac{\partial}{\partial q_i} \phi_{q_i, k_i}(x_i) = \phi_{q_i, k_i}(x_i) \left( \frac{\partial}{\partial q_i} \log \phi_{q_i, k_i}(x_i) \right)$ we have,

$$\int \left( \frac{\partial}{\partial q_i} \mu(\boldsymbol{q}) \right) \phi_{q_i, k_i}(x_i) dx_i$$

$$= \int (\mu(x_i; \boldsymbol{q}) - \mu(\boldsymbol{q})) \phi_{q_i, k_i}(x_i) \left( \frac{\partial}{\partial q_i} \log \phi_{q_i, k_i}(x_i) \right) dx_i \qquad (20)$$

$$\leq \sqrt{\left( \int (\mu(x_i; \boldsymbol{q}) - \mu(\boldsymbol{q}))^2 \phi_{q_i, k_i}(x_i) dx_i \right) \left( \int \left( \frac{\partial}{\partial q_i} \log \phi_{q_i, k_i}(x_i) \right)^2 \phi_{q_i, k_i}(x_i) dx_i \right)}. \qquad (21)$$

Let

$$w_i(p) = \int \left( \frac{\partial}{\partial q_i} \mu(\boldsymbol{q}) \right) \phi_{q_i, k_i}(x_i) dx_i \Bigg|_{\boldsymbol{q} = (p, \cdots, p)} = \frac{\partial}{\partial q_i} \mu(\boldsymbol{q}) \Bigg|_{\boldsymbol{q} = (p, \cdots, p)}$$

and note that by assumption there exists a constant $c$ such that for all $i \in [n]$ and $q_i \in [1/3, 2/3]$,

$$\int \left( \frac{\partial}{\partial q_i} \log \phi_{q_i, k_i}(x_i) \right)^2 \phi_{q_i, k_i}(x_i) dx_i \leq \frac{1}{c \cdot \text{Var}(\mathcal{D}_{q_i}(k_i))}.$$

Then evaluating both sides of Equation (21) at the constant vector $\boldsymbol{q} = (p, \cdots, p)$, we have

$$\left( \int (\mu(x_i; p) - \mu(p))^2 \phi_{p, k_i}(x_i) dx_i \right) \geq \frac{w_i(p)^2}{\int \left( \frac{\partial}{\partial p} \log \phi_{p, k_i}(x_i) \right)^2 \phi_{p, k_i}(x_i) dx_i} \geq c \cdot w_i(p)^2 \text{Var}(\mathcal{D}_p(k_i)).$$

Now we have controlled the contribution of each individual coordinate to the variance of $M$, and it remains to control the contribution of the noise due to privacy.

We will show that for two independent samples $x_i$, $x_i'$ drawn from $\mathcal{D}_p(k_i)$,

$$\mathbb{E}[(\mu(x_i; p) - \mu(x_i'; p))^2] \geq \Omega\Big(w_i(p)^2 \cdot \text{Var}(\mathcal{D}_p(k_i))\Big). \tag{22}$$

Letting

$$\alpha = \sqrt{\mathbb{E}[(\mu(x_i; p) - \mu(x_i'; p))^2]},$$

we can write

$$w_i(p) = \frac{\partial \mu(\boldsymbol{q})}{\partial q_i}\bigg|_{\boldsymbol{q}=(p,\cdots,p)}$$

$$= \frac{\partial(\mu(\boldsymbol{q}) - \mu(x_i'; \boldsymbol{q}))}{\partial q_i}\bigg|_{\boldsymbol{q}=(p,\cdots,p)}$$

$$= \frac{\partial}{\partial q_i} \int_{x_i} (\mu(x_i; \boldsymbol{q}) - \mu(x_i'; \boldsymbol{q}))\phi_{q_i, k_i}(x_i)dx_i \bigg|_{\boldsymbol{q}=(p,\cdots,p)}$$

$$= \int_{x_i} (\mu(x_i; p) - \mu(x_i'; p))\left(\frac{\partial \phi_{q_i, k_i}(x_i)}{\partial q_i}\bigg|_{\boldsymbol{q}=(p,\cdots,p)}\right)dx_i$$

$$= \int_{x_i} (\mu(x_i; p) - \mu(x_i'; p))\left(\frac{\partial \log \phi_{q_i, k_i}(x_i)}{\partial q_i}\bigg|_{\boldsymbol{q}=(p,\cdots,p)}\right)\phi_{p, k_i}(x_i)dx_i$$

$$\leq \sqrt{\left(\int_{x_i} (\mu(x_i; p) - \mu(x_i'; p))^2\phi_{p, k_i}(x_i)dx_i\right)\left(\int_{x_i}\left(\frac{\partial \log \phi_{q_i, k_i}(x)}{\partial q_i}\bigg|_{\boldsymbol{q}=(p,\cdots,p)}\right)^2\phi_{p, k_i}(x)dx_i\right)}$$

$$\leq \alpha \cdot \sqrt{\int_{x_i}\left(\frac{\partial \log \phi_{p_i, k_i}(x_i)}{\partial p_i}\bigg|_{\boldsymbol{p}=(p,\cdots,p)}\right)^2\phi_{p_i, k_i}(x_i)dx_i}$$

$$\leq \alpha \cdot \sqrt{\frac{1}{c \cdot \text{Var}(\mathcal{D}_p(k_i))}}$$

The first equality is by definition. The second equality follows from the fact that $\mu(x_i'; \mathbf{q})$ is constant with respect to $q_i$, so its derivative is 0. The third inequality simply expands out the definition of $\mu(\mathbf{q})$. The fourth equality follows from the linearity of derivatives, the fact $\mu(x_i; \mathbf{q}) - \mu(x_i', \mathbf{q})$ is constant with respect to $q_i$, and the fact that $(\mu(x_i; \mathbf{q}) - \mu(x_i', \mathbf{q}))|_{\mathbf{q}=(p,\cdots,p)} = (\mu(x_i; p) - \mu(x_i', p))$. The fifth equality follows from the formula $\frac{\partial}{\partial x} \ln f(x) = \frac{\frac{\partial}{\partial x} f(x)}{f(x)}$, which holds for any differentiable function $f$. The first inequality is a result of the Cauchy-Schwarz inequality. The second inequality follows from the definition of $\alpha$, and the final inequality follows from Assumption 2 of Lemma 5.3.

Therefore,

$$\alpha \geq w_i(p) \cdot \sqrt{c \cdot \text{Var}(\mathcal{D}_p(k_i))}.$$

We now argue that any $(\epsilon, \epsilon^2/100)$-differentially private mechanism should have variance $\Omega(\alpha^2 \log \frac{1}{\epsilon}/10\epsilon^2)$. Suppose that we had a mechanism that violated this property. Then by running this mechanism $\frac{1}{\epsilon^2 \log \frac{1}{\epsilon}}$ times and averaging, the advanced composition theorem implies that this average is $(1, 1/100)$-DP. This averaged output however has variance $O(\alpha^2/10)$. Thus given samples

$x_i$, and $x_i'$ such that $|\mu(x_i; p) - \mu(x_i'; p)| \geq \alpha/2$, if the noise had variance $O(\alpha^2/10)$ on $(x_i, x_{-i})$ as well as on $(x_i', x_{-i})$ (when $x_{-i}$ is drawn randomly), then these two inputs would be distinguishable with probability at least $9/10$. This however violates the $(1, 1/100)$-DP of the averaged algorithm. This implies that for random $x_i$, the noise added by the DP algorithm is at least $\Omega(\alpha^2 \log \frac{1}{\epsilon}/20\epsilon^2)$

Thus the variance of $M$ is,

$$\text{Var}_{\forall i \in [n], x_i \sim \mathcal{D}_p(k_i), M}(M) \geq \sum_{i=1}^{n} c \cdot w_i(p)^2 \text{Var}(\mathcal{D}_p(k_i)) + \Omega\left(\frac{w_i(p)^2 \text{Var}(\mathcal{D}_p(k_i))}{\epsilon^2}\right) \quad (23)$$

Finally, since the weights $w_i(p)$ that we defined need not sum to 1, they will need to be normalized to sum to 1 to satisfy the conditions of NLE. We need to show this normalisation does not substantially increase the variance of the estimator in NLE defined by these weights. This is equivalent to showing that the normalisation term, $\sum_{i=1}^{n} w_i(p)$ is large for some $p$. For $p \in [1/3, 2/3]$, let $\gamma : [1/3, 2/3] \to [0, 1]^n$, defined by $\gamma(p) = (p, \cdots, p)$, be a path in $[0, 1]^n$ then by the fundamental theorem of line integrals,

$$3 \int_{1/3}^{2/3} \sum_{i=1}^{n} w_i(p) dp = 3 \int_{1/3}^{2/3} \left( \sum_{i=1}^{n} \left( \frac{\partial}{\partial q_i} \mu(\boldsymbol{q}) \right) \Big|_{\boldsymbol{q}=(p,\cdots,p)} \right) dp$$

$$= 3 \int_{\gamma} \nabla \mu(\boldsymbol{q}) \cdot \mathbf{1} d\boldsymbol{q}$$

$$= 3(\mu(2/3, \cdots, 2/3) - \mu(1/3, \cdots, 1/3))$$

$$= 1$$

This implies that there exists $p^* \in [1/3, 2/3]$ such that $\sum_{i=1}^{n} w_i(p^*) \geq 1$. Define

$$M_{\text{NL}}(x_1, \cdots, x_n) = \sum_{i=1}^{n} \frac{w_i(p^*)}{\sum_{j=1}^{n} w_j(p^*) x_j} + \text{Lap}\left( \frac{\max_i \frac{w_i(p^*)}{\sum_{j=1}^{n} w_j(p^*)} \sqrt{\text{Var}(\mathcal{D}_p(k_i))}}{\epsilon} \right)$$

$$= \frac{1}{\sum_{i=1}^{n} w_i(p^*)} \left( \sum_{i=1}^{n} w_i(p^*) x_i + \text{Lap}\left( \frac{\max_i w_i(p^*) \sqrt{\text{Var}(\mathcal{D}_p(k_i))}}{\epsilon} \right) \right),$$

where the second equality follows from properties of the Laplace distribution. Now,

$$\text{Var}_{\mathcal{D}_p}(M_{\text{NL}}) \leq \frac{1}{(\sum_{i=1}^{n} w_i(p^*))^2} \left( \sum_{i=1}^{n} w_i(p^*)^2 \text{Var}(\mathcal{D}_p(k_i)) + O\left( \frac{\max_i w_i(p^*)^2 \text{Var}(\mathcal{D}_p(k_i))}{\epsilon^2} \right) \right)$$

$$\leq \sum_{i=1}^{n} w_i(p^*)^2 \text{Var}(\mathcal{D}_p(k_i)) + O\left( \frac{\max_i w_i(p^*)^2 \text{Var}(\mathcal{D}_p(k_i))}{\epsilon^2} \right),$$

where the second inequality comes from the fact that $\sum_{i=1}^{n} w_i(p^*) \geq 1$. Comparing this with Equation 23, we see that specifically, at $p = p^*$,

$$\text{Var}_{\mathcal{D}_{p^*}}(M_{\text{NL}}) \leq O\left( \text{Var}_{\mathcal{D}_{p^*}}(M) \right).$$

Now, if $p, p^* \in [1/3, 2/3]$ then $\text{Var}(\mathcal{D}_p(k_i)) = \Theta\left( \text{Var}(\mathcal{D}_{p^*}(k_i)) \right)$ so $\text{Var}_{D_p}(M_{\text{NL}}) = \Theta(\text{Var}_{D_{p^*}}(M_{\text{NL}}))$. Therefore, the worst case variance of $M_{\text{NL}}$ is less than the worst case variance of $M$ over all $p \in [1/3, 2/3]$, as required. $\qquad \square$

**Lemma E.1.** *For any randomised mechanism* $M : [0, 1]^n \to [0, 1]$,

$$\text{Var}_{\forall i \in [n], x_i \sim \mathcal{D}_p(k_i), M}(M) \quad (16)$$

$$= \mathbb{E}_{\forall i \in [n], x_i \sim \mathcal{D}_p(k_i), M}[(M(x_1, ..., x_n) - \mu(p))^2]$$

$$\geq \sum_{i=1}^{n} \mathbb{E}_{x_i \sim \mathcal{D}_p(k_i)}[(\mu(x_i; p) - \mu(p))^2] + \mathbb{E}_{\forall i \in [n], x_i \sim \mathcal{D}_p(k_i), M}[(M(x_1, ..., x_n) - \mu(x_1, ..., x_n; p))^2]$$

*Proof of Lemma E.1* Let $M : [0, 1]^n \to [0, 1]$ be a randomised mechanism and suppose that each $x_i \sim \mathcal{D}(p_i, k_i)$ where $p_i \sim \mathcal{D}$. Now, our goal is to decompose the variance of $M$ into

the variance conditioned on each coordinate, and the variance inherent in the mechanism itself. Let $\mu = \mathbb{E}_{x_1 \sim \mathcal{D}(k_1), \cdots, x_n \sim \mathcal{D}(k_n), M}[M(x_1, \cdots, x_n)]$ be the expectation and for any $I \subset [n]$, let $\mu(x_{[n] \setminus I}) = \mathbb{E}_{\forall i \in I, x_i \sim \mathcal{D}(k_i), M}[M(x)]$ be the expectation conditioned only on the randomness in $I$. So,

$$
\begin{aligned}
\text{Var}(M) &= \mathbb{E}_{x_1 \sim \mathcal{D}(k_1), \cdots, x_n \sim \mathcal{D}(k_n), M}[(M(x_1, \cdots, x_n) - \mu)^2] \\
&= \mathbb{E}_{x_1 \sim \mathcal{D}(k_1)} \mathbb{E}_{x_2 \sim \mathcal{D}(k_2), \cdots, x_n \sim \mathcal{D}(k_n), M}[(M(x_1, \cdots, x_n) - \mu_1(x_1) + \mu_1(x_1) - \mu)^2] \\
&= \mathbb{E}_{x_1 \sim \mathcal{D}(k_1)} \mathbb{E}_{x_2 \sim \mathcal{D}(k_2), \cdots, x_n \sim \mathcal{D}(k_n), M}[(M(x_1, \cdots, x_n) - \mu_1(x_1))^2 \\
&\qquad + 2(M(x_1, \cdots, x_n) - \mu_1(x_1))(\mu_1(x_1) - \mu) + (\mu_1(x_1) - \mu)^2] \\
&= \mathbb{E}_{x_1 \sim \mathcal{D}(k_1)}[(\mu_1(x_1) - \mu)^2] + \mathbb{E}_{x_1 \sim \mathcal{D}(k_1)} \mathbb{E}_{x_2 \sim \mathcal{D}(k_2), \cdots, x_n \sim \mathcal{D}(k_n), M}[(M(x_1, \cdots, x_n) - \mu_1(x_1))^2].
\end{aligned}
$$

Now, by induction we obtain the following decomposition of the variance of $M$,

$$
\begin{aligned}
\text{Var}(M) &= \sum_{i=1}^{n} \mathbb{E}_{x_1 \sim \mathcal{D}(k_1), \cdots, x_i \sim \mathcal{D}(k_i)}[(\mu(x_{j \leq i}) - \mu(x_{j < i}))^2] \\
&\qquad\qquad + \mathbb{E}_{x_1 \sim \mathcal{D}(k_1), \cdots, x_n \sim \mathcal{D}(k_n), M}[(M(x_1, \cdots, x_n) - \mu(x_1, \cdots, x_n))^2] \\
&\geq \sum_{i=1}^{n} \mathbb{E}_{x_i \sim \mathcal{D}(k_i)}[(\mu(x_i) - \mu)^2] + \mathbb{E}_{x_1 \sim \mathcal{D}(k_1), \cdots, x_n \sim \mathcal{D}(k_n), M}[(M(x_1, \cdots, x_n) - \mu(x_1, \cdots, x_n))^2]
\end{aligned}
$$

where the second inequality follows from Jensen's inequality:

$$
\begin{aligned}
\mathbb{E}_{x_1 \sim \mathcal{D}(k_1), \cdots, x_i \sim \mathcal{D}(k_i)}[(\mu(x_{j \leq i}) - \mu(x_{j < i}))^2] &\geq \mathbb{E}_{x_i \sim \mathcal{D}(k_i)}[(\mathbb{E}_{x_1 \sim \mathcal{D}(k_1), \cdots, x_{i-1} \sim \mathcal{D}(k_i)}[\mu(x_{j \leq i}) - \mu(x_{j < i})])^2] \\
&= \mathbb{E}_{x_i \sim \mathcal{D}(k_i)}[(\mu(x_i) - \mu)^2].
\end{aligned}
$$

$\square$

**Lemma 5.4.** *For any distribution $\mathcal{D}$, $n > 0$ and $\beta \in [0,1]$, if for all $k_i$, $f_{\mathcal{D}}^{k_i}(n, \sigma_p^2, \beta) = \tilde{O}(\text{Var}(\mathcal{D}(k_i)))$ then for any $\mathbf{w} \in [0,1]^n$ such that $\sum_{i=1}^{n} w_i = 1$, we have $\text{Var}(M_{TNL}(\cdot \,; \mathbf{w})) = \tilde{O}(\text{Var}(M_{NL}(\cdot \,; \mathbf{w})))$. Further, the bias of $M_{TNL}$ is at most $\beta$.*

*Proof of Lemma 5.4.* The variance claim follows immediately from noting that $\text{Var}\left([x_i]_{p - f_{\mathcal{D}}^{k_i}(n, \sigma_p^2, \beta)}^{p + f_{\mathcal{D}}^{k_i}(n, \sigma_p^2, \beta)}\right) \leq \text{Var}(x_i)$, and the assumption that $f_{\mathcal{D}}^{k_i}(n, \sigma_p^2, \beta) = \tilde{O}(\text{Var}(\mathcal{D}(k_i)))$. The bias claim follows from noting that with probability $1 - \beta$, $[x_i]_{p - f_{\mathcal{D}}^{k_i}(n, \sigma_p^2, \beta)}^{p + f_{\mathcal{D}}^{k_i}(n, \sigma_p^2, \beta)} = x_i$. This implies that $M_{\text{TNL}}$ is within $\beta$ in total variation distance to an unbiased estimator. Since $M_{\text{TNL}}$ takes values in $[0,1]$, this implies the mean is in $[p - \beta, p + \beta]$. $\square$

**Corollary 5.5.** *Given $k_1, \cdots, k_n \in \mathbb{N}$, and $\sigma_p$, there exists a family of distributions $\mathcal{D}_p$ such that*

$$
\min_{M, \text{ unbiased}} \max_{p \in [1/3, 2/3]} \text{Var}_{\forall i \in [n], x_i \sim \mathcal{D}_p(k_i)}[M(x_1, \cdots, x_n)] \geq \tilde{\Omega}\left(\min\left\{\frac{\frac{k^*}{\epsilon^2} + \sum_{i=1}^{n} \min\{k_i, k^*\}}{(\sum_{i=1}^{n} \min\{k_i, \sqrt{k_i k^*}\})^2}, \frac{\sigma_p^2}{n}\right\}\right).
$$

*Proof of Corollary 5.5.* Firstly, suppose that $\sigma_p = 0$, so the meta-distribution is constant, and $\mathcal{D}_p(k_i) = \text{Bin}(k_i, p)$. Then the Fisher information of $\phi_{p,k_i}$ is $\int \left(\frac{\partial}{\partial p} \log \phi_{p,k_i}(x_i)\right)^2 \phi_{p,k_i}(x_i) dx_i = \frac{k_i}{p(1-p)}$ and $\text{Var}(\mathcal{D}_p(k_i)) = \frac{p(1-p)}{k_i}$, so $\mathcal{D}_p(k_i)$ satisfies Condition 8 of Theorem 5.1. Additionally,

$$
\min_{M, \text{ unbiased}} \max_{p \in [1/3, 2/3]} \text{Var}_{\mathcal{D}_p}[M] = \tilde{\Omega}\left(\max_{p \in [1/3, 2/3]} \text{Var}_{\mathcal{D}_p}[\widehat{p}_{\epsilon}^{\text{ideal}}]\right) \qquad \text{(under conditions of Thm 4.1)}
$$

We can view the truncation as simply choosing a maximum $k^*$ so that $T = \sqrt{\frac{k^*}{p(1-p)}}$. Now, the un-normalised weights of $\widehat{p}_{\epsilon}^{\text{ideal}}$ are

$$
\min\left\{\frac{1}{\text{Var}(\mathcal{D}_p(k_i))}, \frac{T}{\sqrt{\text{Var}(\mathcal{D}_p(k_i))}}\right\} = \min\left\{\frac{k_i}{p(1-p)}, \frac{\sqrt{k_i k^*}}{p(1-p)}\right\}.
$$

Further, $\mathrm{Var}([\widehat{p}_i]_{a_i}^{b_i}) \leq \mathrm{Var}(\mathcal{D}(k_i))$ and we assume thoughout this paper that $\mathrm{Var}([\widehat{p}_i]_{a_i}^{b_i}) \geq [(1/2)\mathrm{Var}(\mathcal{D}(k_i))]$. So, $\mathrm{Var}([\widehat{p}_i]_{a_i}^{b_i}) = \Theta(\mathrm{Var}(\mathcal{D}(k_i))) = \Theta(\frac{p(1-p)}{k_i})$. Finally, since binomials are highly concentrated, $|b_i - a_i| = \Omega(\sigma_i)$, which implies that $\frac{\max_i w_i^* |b_i - a_i|}{\epsilon}$ as defined in eqn (3) is achieved at $k_i = k^*$. Thus,

$$\min_{M,\text{ unbiased}} \max_{p \in [1/3, 2/3]} \mathrm{Var}_{\mathcal{D}_p}[M] = \max_{p \in [1/3, 2/3]} \frac{\Omega\left(\frac{k^*}{p(1-p)\epsilon^2}\right) + \sum_{i=1}^n \left(\min\left\{\frac{k_i}{p(1-p)}, \frac{\sqrt{k_i k^*}}{p(1-p)}\right\}\right)^2 \frac{1}{2}\frac{p(1-p)}{k_i}}{\left(\sum_{i=1}^n \min\left\{\frac{k_i}{p(1-p)}, \frac{\sqrt{k_i k^*}}{p(1-p)}\right\}\right)^2}$$

$$= \tilde{\Omega}\left(\max_{p \in [1/3, 2/3]} p(1-p)\frac{\frac{k^*}{\epsilon^2} + \sum_{i=1}^n \min\{k_i, k^*\}}{(\sum_{i=1}^n \min\{k_i, \sqrt{k_i k^*}\})^2}\right)$$

$$= \tilde{\Omega}\left(\frac{\frac{k^*}{\epsilon^2} + \sum_{i=1}^n \min\{k_i, k^*\}}{(\sum_{i=1}^n \min\{k_i, \sqrt{k_i k^*}\})^2}\right),$$

where the first equality comes from Theorem 5.1, the second equality pulls out common factors, and the third equality is because $p$ is bounded away from 0 and 1.

For the other component of the bound we will let $\mathcal{D}_p$ be a truncated Gaussian distribution. Let $\phi$ and $\Phi$ respectively be the probability density function and cumulative density function of the standard Gaussian $\mathcal{N}(0, 1)$. Let $W$ be such that $\gamma := \Phi(W) - \Phi(-W) \geq 9/10$ and $\lambda := \frac{2W\phi(W)}{\Phi(W) - \Phi(-W)} \leq 1/2$. Define the truncated Gaussian $\mathcal{D}_p$ with mean $p$ on $[p - \frac{\sigma_p}{\sqrt{1-\lambda}}W, p + \frac{\sigma_p}{\sqrt{1-\lambda}}W]$ by the probability density function:

$$\phi_p(q) = \begin{cases} \frac{1}{\gamma}\phi\left((q-p)\frac{\sqrt{1-\lambda}}{\sigma_p}\right) & q \in [p - \frac{\sigma_p}{\sqrt{1-\lambda}}W, p + \frac{\sigma_p}{\sqrt{1-\lambda}}W] \\ 0 & \text{otherwise.} \end{cases}$$

Now, the variance of $\mathcal{D}_p$ is $\sigma_p^2$ and the Fisher information of $\mathcal{D}_p$ is given by [29]

$$\frac{1}{\sigma_p^2}(1-\lambda)^2 \in \left[\frac{1}{4\sigma_p^2}, \frac{1}{\sigma_p^2}\right]. \tag{24}$$

Since any sample from $\mathcal{D}$ can be post-processed into a sampling from $\mathcal{D}(k)$ for any $k \in \mathbb{N}$, we have

$$\min_{M,\text{ unbiased}} \max_{p \in [1/3, 2/3]} \mathrm{Var}_{\forall i \in [n], x_i \sim \mathcal{D}_p(k_i)}[M(x_1, \cdots, x_n)] \geq \min_{M,\text{ unbiased}} \max_{p \in [1/3, 2/3]} \mathrm{Var}_{p_1, \cdots, p_n \sim \mathcal{D}_p}[M(p_1, \cdots, p_n)]$$

$$\geq \max_{p \in [1/3, 2/3]} O\left(\frac{\sigma_p^2}{n}\right)$$

$$= O(\frac{\sigma_p^2}{n}),$$

where the second inequality follows from the Cramer-Rao bound [30] and Equation (24). □

# F  Example Initial Estimators

In this section we give example initial mean and variance estimation procedures that can be used in the framework described in Section 4. For both estimators, we show that they satisfy the conditions of Theorem 4.1, and thus can be used as initial estimators in Algorithm 2, assuming all other technical conditions are satisfied. This also immediately implies that the set of initial mean and variance estimators which satisfy the conditions of Theorem 4.1 is non-empty.

We note again that the estimators described in this section are examples of estimators that achieve the conditions of Theorem 4.1, and that any private mean and variance estimators that satisfy these conditions could be used instead. As discussed in Section 4.2, one may choose to use different estimators of these initial quantities in different settings (for example, if local differential privacy is required or if different distributional assumptions are known).

### F.1 Initial Mean Estimation

We will begin with the initial mean estimate $\widehat{p}_\epsilon^{\text{initial}}$. We consider the simplest mean estimation subroutine, where the analyst collects a single data point from the $n/10$ users with the smallest $k_i$, then privately computes the empirical mean of these points using the Laplace Mechanism. The following lemma shows that this process is differentially private and satisfies the accuracy conditions of Theorem 4.1, i.e., that with high probability, $\widehat{p}_\epsilon^{\text{initial}}$ is close to $p$ and $\widehat{p}_\epsilon^{\text{initial}}(1 - \widehat{p}_\epsilon^{\text{initial}})$ is close to $p(1-p)$.

**Lemma F.1.** *Fix any* $\epsilon > 0$ *and let* $\widehat{p}_\epsilon^{\text{initial}}(x_{(9n/10)+1}^1, \cdots, x_n^1) = \frac{1}{n/10} \sum_{i=(9n/10)+1}^n x_i +$ Lap $\left(\frac{10}{\epsilon n}\right)$. *Then* $\widehat{p}_\epsilon^{\text{initial}}$ *is* $\epsilon$-*differentially private,* $\mathbb{E}[\widehat{p}_\epsilon^{\text{initial}}(x_{(9n/10)+1}^1, \cdots, x_n^1)] = p$ *and if* $p \geq \frac{20 \log(1/\beta)}{n}$, *then for* $n$ *sufficiently large,*

$$\Pr[|\widehat{p}_\epsilon^{\text{initial}} - p| \leq \alpha] \leq \beta \text{ for } \alpha = 2 \max\left\{\sqrt{\frac{12\widehat{p}_\epsilon^{\text{initial}} \log(4/\beta)}{n/10} + \frac{36 \log^2(4/\beta)}{n^2/100}} + \frac{6 \log(4/\beta)}{n/10}, \frac{\log(2/\beta)}{\epsilon n/10}\right\} \leq f_{\mathcal{D}}^{k_i}(n, \sigma_p^2, \beta).$$

*Further, if* $\min\{p, 1-p\} \geq 12 \max\left\{\frac{3 \log(4/\beta)}{n/10}, \frac{\log(2/\beta)}{\epsilon n/10}\right\}$ *then with probability* $1 - \beta$, $\widehat{p}_\epsilon^{\text{initial}} \in [\frac{1}{2}p, \frac{3}{2}p]$ *and* $\widehat{p}_\epsilon^{\text{initial}}(1 - \widehat{p}_\epsilon^{\text{initial}}) \in [\frac{p(1-p)}{2}, \frac{3p(1-p)}{2}]$.

The concentration bound follows from noticing that $\mathcal{D} = \text{Ber}(p)$ and using the concentration of binomial random variables.

Note that the expression of $\alpha$ depends only on quantities known to the analyst – including $\widehat{p}_\epsilon^{\text{initial}}$, which will be observed as output – so that $\alpha$ can be computed directly for use in Algorithm 2. Although our presentation of Algorithm 2 requires $\alpha$ to be specified up front as input to the algorithm, it could equivalently be computed internally by the algorithm as a function of $\widehat{p}_\epsilon^{\text{initial}}$ and other input parameters.

*Proof.* Firstly, the privacy guarantees follows immediately from the Laplace Mechanism in differential privacy [12] noting that $\frac{10}{n} \sum_{i=(9n/10)+1}^n x_i^1$ has sensitivity $\frac{10}{n}$.

Now, let us turn to the two accuracy guarantees. We will start with the guarantee that $\widehat{p}_\epsilon^{\text{initial}}$ is close to $p$ with high-probability. Note that $\mathcal{D}$ is simply a Bernoulli random variable with mean $p$ so since each sample is independent, $\frac{10}{n} \sum_{i=(9n/10)+1}^n x_i^1 = \text{Bin}(n/10, p)$. Thus, if $n \geq \frac{20 \log(1/\beta)}{p}$, a Chernoff bound gives

$$\Pr\left[\left|\frac{10}{n} \sum_{i=(9n/10)+1}^n x_i^1 - p\right| \geq \sqrt{\frac{3 \min\{p, 1-p\} \log(4/\beta)}{n/10}}\right] \leq \beta/2.$$

Therefore, combining with a high probability bound on the Laplace distribution,

$$\Pr\left[|\widehat{p}_\epsilon^{\text{initial}} - p| \geq \sqrt{\frac{3 \min\{p, 1-p\} \log(4/\beta)}{n/10}} + \frac{\log(2/\beta)}{\epsilon n/10}\right] \leq \beta.$$

We will condition on the following event for the remainder of the proof, which will occur with probability $1 - \beta$:

$$|\widehat{p}_\epsilon^{\text{initial}} - p| \leq 2 \max\left\{\sqrt{\frac{3 \min\{p, 1-p\} \log(4/\beta)}{n/10}}, \frac{\log(2/\beta)}{\epsilon n/10}\right\}.$$

Now if $|\widehat{p}_\epsilon^{\text{initial}} - p| \leq 2\sqrt{\frac{3 \min\{p, 1-p\} \log(4/\beta)}{n/10}}$. Since we need $\alpha$ in terms of $\widehat{p}_\epsilon^{\text{initial}}$ rather than $p$ (since $\widehat{p}_\epsilon^{\text{initial}}$ is known to the algorithm), we need to rework this formula. Squaring both sides and bringing all the terms to the same side, we obtain

$$p^2 - 2\left(\widehat{p}_\epsilon^{\text{initial}} + \frac{6 \log(4/\beta)}{n/10}\right)p + (\widehat{p}_\epsilon^{\text{initial}})^2 \leq 0.$$

Completing the square we obtain

$$\left(p - \widehat{p}_\epsilon^{\text{initial}} - \frac{6 \log(4/\beta)}{n/10}\right)^2 + (\widehat{p}_\epsilon^{\text{initial}})^2 - \left(\widehat{p}_\epsilon^{\text{initial}} + \frac{6 \log(4/\beta)}{n/10}\right)^2 \leq 0.$$

Now, rearranging and taking the square root, we obtain

$$\left| p - \widehat{p}_\epsilon^{\text{initial}} - \frac{6 \log(4/\beta)}{n/10} \right| \leq \sqrt{\left( \widehat{p}_\epsilon^{\text{initial}} + \frac{6 \log(4/\beta)}{n/10} \right)^2 - (\widehat{p}_\epsilon^{\text{initial}})^2}$$

then by squaring both sides, using the fact that $\min\{p, 1 - p\} \leq p$, and rearranging we have

$$|\widehat{p}_\epsilon^{\text{initial}} - p| \leq \sqrt{\frac{12 \widehat{p}_\epsilon^{\text{initial}} \log(4/\beta)}{n/10} + \frac{36 \log^2(4/\beta)}{n^2/100} + \frac{6 \log(4/\beta)}{n/10}}$$

which implies that,

$$|\widehat{p}_\epsilon^{\text{initial}} - p| \leq 2 \max \left\{ \sqrt{\frac{12 \widehat{p}_\epsilon^{\text{initial}} \log(4/\beta)}{n/10} + \frac{36 \log^2(4/\beta)}{n^2/100} + \frac{6 \log(4/\beta)}{n/10}}, \frac{\log(2/\beta)}{\epsilon n/10} \right\}.$$

We need to show that this expression is less than or equal to $f_{\mathcal{D}}^{k_i}(n, \sigma_p^2, \beta)$ because $\alpha = O(1/\sqrt{n})$. To see this, note that $\alpha = O(1/\sqrt{n})$ and $f_{\mathcal{D}}^{k_i}(n, \sigma_p^2, \beta)$ is increasing towards 1 as $n$ grows large. Thus for $n$ sufficiently large, $\alpha \leq f_{\mathcal{D}}^{k_i}(n, \sigma_p^2, \beta)$ will be satisfied.

Next we turn to proving the second accuracy claim, that $\widehat{p}_\epsilon^{\text{initial}}(1 - \widehat{p}_\epsilon^{\text{initial}})$ is concentrated around $p(1 - p)$. Let $\mathcal{E} = \widehat{p}_\epsilon^{\text{initial}} - p$ so

$$\widehat{p}_\epsilon^{\text{initial}}(1 - \widehat{p}_\epsilon^{\text{initial}}) = (p + \mathcal{E})(1 - p - \mathcal{E}) = p(1 - p) + (1 - 2p)\mathcal{E} - \mathcal{E}^2$$

Now, if $\min\{p, 1 - p\} \geq K \max \left\{ \frac{3 \log(4/\beta)}{n/10}, \frac{\log(2/\beta)}{\epsilon n/10} \right\}$ for some constant $K$, then

$$|\mathcal{E}| \leq \sqrt{\frac{3 \min\{p, 1 - p\} \log(4/\beta)}{n/10}} + \frac{\log(2/\beta)}{\epsilon n/10}$$

$$\leq \sqrt{\frac{\min\{p, 1 - p\} \min\{p, (1 - p)\}}{K}} + \frac{\min\{p, (1 - p)\}}{K}$$

$$\leq \frac{2 \min\{p, 1 - p\}}{K}.$$

Thus, combining this with the fact that $1 - 2p \leq \max\{p, 1 - p\}$ for $p \in [0, 1]$,

$$|(1 - 2p)\mathcal{E} - \mathcal{E}^2| \leq \max\{p, 1 - p\} \frac{2 \min\{p, 1 - p\}}{K} + \left( \frac{2 \min\{p, 1 - p\}}{K} \right)^2$$

$$\leq \frac{6p(1 - p)}{K}$$

Finally, choosing $K = 12$ gives,

$$\widehat{p}_\epsilon^{\text{initial}}(1 - \widehat{p}_\epsilon^{\text{initial}}) \in \left[ \frac{p(1 - p)}{2}, \frac{3p(1 - p)}{2} \right].$$

$\square$

## F.2 Initial Variance Estimation

We now turn to estimating $\sigma_p^2$. Let us first provide some background on privately estimating the standard deviation of well-behaved distributions. Lemma F.2 guarantees the existence of a differentially private algorithm for estimating standard deviation within a small constant factor with high probability, as long as the sample size is sufficiently large. The following is a slight generalisation of the estimation of the standard deviation of a Gaussian given in [21] due to [26].

**Lemma F.2** (DP standard deviation estimation). *For all $n \in \mathbb{N}$, $\sigma_{min} < \sigma_{max} \in [0, \infty]$, $\epsilon > 0$, $\delta \in (0, \frac{1}{n}]$, $\beta \in (0, 1/2)$, $\zeta > 0$, there exists an $(\epsilon, \delta)$-differentially private algorithm $\mathcal{M}$ that satisfies: if $x_1, \ldots, x_n$ are i.i.d. draws from a distribution $P$ which has standard deviation $\sigma \in [\sigma_{min}, \sigma_{max}]$ and absolute central third moment $\rho = \mathbb{E}[|x - \mu(P)|^3]$ such that $\frac{\rho}{\sigma^3} \leq \zeta$, then if $n \geq c\zeta^2 \min\{\frac{1}{\epsilon} \ln(\frac{\ln \frac{\sigma_{max}}{\sigma_{min}}}{\beta}), \frac{1}{\epsilon} \ln(\frac{1}{\delta\beta})\}$, (where $c$ is a universal constant), then $\mathcal{M}$ produces an estimate $\widehat{\sigma}$ of the standard deviation such that $\Pr_{x_1, \ldots, x_n \sim P, \mathcal{M}}(\sigma^2 \leq \widehat{\sigma}^2 \leq 8\sigma^2) \geq 1 - \beta$.*

In order to estimate $\sigma_p^2$, we will use the estimator promised by Lemma F.2 on the data of the $\log n$ users with the largest $k_i$. Let $k = k_{\log n}$, so the top $\log n$ individuals all have at least $k$ data points. We will have these individuals report $\widehat{p}_i^k := \frac{1}{k}\sum_{j=1}^{k} x_j^i$, which is the empirical mean of their first $k$ data points. Thus, we are running the estimator promised in Lemma F.2 on $\mathcal{D}(k)$ with $\log n$ data points. In order to utilise Lemma F.2, we first need to ensure that $\mathcal{D}(k)$ satisfies the moment condition that $\rho/\sigma^3$ is bounded, which is shown in Lemma F.3.

**Lemma F.3.** *For $k \in \mathbb{N}$, suppose $p \in [\frac{1}{k}, 1 - \frac{1}{k}]$, $\sigma_p \geq \frac{1}{k}$, $k \geq 2$, and there exists $\gamma > 0$ such that $\frac{\rho_{\mathcal{D}}}{\sigma_p^3} \leq \gamma$ where $\rho_{\mathcal{D}}$ denotes the absolute central third moment of $\mathcal{D}$. Then $\frac{\rho_{\mathcal{D}(k)}}{\mathrm{Var}(\mathcal{D}(k))^{3/2}} \leq 8(3\sqrt{3} + \gamma)$.*

*Proof.* Note that $\mathbb{E}[\mathcal{D}(k)] = p$. Then we can bound the absolute third central moment as follows,

$$\mathbb{E}_{x \sim \mathcal{D}(k)}[|x - p|^3] = \mathbb{E}_{p_i \sim \mathcal{D}}\mathbb{E}_{y \sim \mathrm{Bin}(k,p_i)}[|(\frac{1}{k}y - p_i) - (p - p_i)|^3]$$

$$\leq 4\left( \mathbb{E}_{p_i \sim \mathcal{D}}\mathbb{E}_{y \sim \mathrm{Bin}(k,p_i)}[|\frac{1}{k}y - p_i|^3] + \mathbb{E}_{p_i \sim \mathcal{D}}[|p - p_i|^3] \right)$$

$$\leq 4\left( \frac{1}{k^3}\mathbb{E}_{p_i \sim \mathcal{D}}\left[ \sqrt{\mathbb{E}_{y \sim \mathrm{Bin}(k,p_i)}[|y - k \cdot p_i|^2]\mathbb{E}_{y \sim \mathrm{Bin}(k,p_i)}[|y - p_i|^4]} \right] + \gamma\sigma_p^3 \right)$$

$$\text{(by Cauchy-Schwarz inequality)}$$

$$\leq 4\left( \frac{1}{k^3}\mathbb{E}_{p_i \sim \mathcal{D}}\left[ \sqrt{k^2(p_i(1 - p_i))^2(1 + 3kp_i(1 - p_i))} \right] + \gamma\sigma_p^3 \right)$$

$$\leq 4\left( \frac{1}{k^3}\mathbb{E}_{p_i \sim \mathcal{D}}[k(p_i(1 - p_i))] + \frac{1}{k^3}\mathbb{E}_{p_i \sim \mathcal{D}}[\sqrt{3k^3(p_i(1 - p_i))^3}] + \gamma\sigma_p^3 \right)$$

$$\leq 4\left( \frac{1}{k^2}p(1 - p) + \frac{\sqrt{3}}{k^{3/2}}\mathbb{E}_{p_i \sim \mathcal{D}}[\sqrt{(p_i(1 - p_i))^3}] + \gamma\sigma_p^3 \right)$$

$$\text{(by Jensen's inequality)}$$

$$\leq 4\left( \frac{1}{k^{3/2}}\sqrt{(p(1 - p))^3} + \frac{\sqrt{3}}{k^{3/2}}\mathbb{E}_{p_i \sim \mathcal{D}}[\sqrt{(p_i(1 - p_i))^3}] + \gamma\sigma_p^3 \right),$$

where the first inequality follows from the following inequality that holds for all real valued $a$ and $b$: $|a - b|^3 \leq 4(|a|^3 + |b|^3)$. The second to last inequality follows from Jensen's inequality since $h(x) = x(1-x)$ is concave, and the last inequality follows since $\frac{1}{\sqrt{k}} \leq \sqrt{p(1-p)}$. Now, we will use a generalised form of Jensen's inequality to bound $\mathbb{E}_{p_i \sim \mathcal{D}}[\sqrt{(p_i(1 - p_i))^3}]$. Let $h(x) = (x(1-x))^{3/2}$ and

$$\phi(x) = \frac{h(x) - h(p)}{(x - p)^2} - \frac{h'(p)}{x - p}.$$

Since $p \in [\frac{1}{k}, 1 - \frac{1}{k}]$,

$$\max_{x \in [\frac{1}{2k}, 1 - \frac{1}{2k}]} \phi(x) \leq (1/2) \max_{x \in [\frac{1}{2k}, 1 - \frac{1}{2k}]} h''(x) \leq h''\left(\frac{1}{2k}\right) = \frac{3(8(\frac{1}{2k})^2 - 8(\frac{1}{2k}) + 1)}{4\sqrt{(1 - \frac{1}{2k})\frac{1}{2k}}} = \frac{3(8 - 16k + 4k^2)}{8k\sqrt{(2k - 1)}} \leq \frac{3}{2}\sqrt{k}.$$

If $x \notin [\frac{1}{2k}, 1 - \frac{1}{2k}]$ then $|x - p| \geq \frac{1}{2k}$ and $h(x) < h(p)$, so

$$\phi(x) \leq \frac{|h'(p)|}{|x - p|} = \frac{3|1 - 2p|\sqrt{p(1 - p)}}{2|p - x|} \leq \frac{3}{2}\frac{\sqrt{p(1 - p)}}{|p - x|} \leq \max\left\{ \frac{3}{2}\frac{\sqrt{\frac{1}{k}(1 - \frac{1}{k})}}{|\frac{1}{k} - x|}, \frac{3}{2}\frac{\sqrt{\frac{1}{k}(1 - \frac{1}{k})}}{|1 - \frac{1}{k} - x|} \right\} \leq 3\sqrt{k - 1} \leq 3\sqrt{k}.$$

Therefore, by the generalised Jensen's inequality,

$$\mathbb{E}_{p_i \sim \mathcal{D}}[\sqrt{(p_i(1 - p_i))^3}] \leq \sqrt{(p(1 - p))^3} + \sigma_p^2 \cdot 3\sqrt{k} \leq \sqrt{(p(1 - p))^3} + \sigma_p^2 \cdot 3\sqrt{k}.$$

Continuing to bound the absolute central third moment as above,

$$\mathbb{E}_{x\sim\mathcal{D}(k)}[|x-p|^3] \le 4\left(\frac{1}{k^{3/2}}\sqrt{(p(1-p))^3} + \frac{\sqrt{3}}{k^{3/2}}\mathbb{E}_{p_i\sim\mathcal{D}}[\sqrt{(p_i(1-p_i))^3}] + \gamma\sigma_p^3\right)$$

$$\le 4\left(\frac{1}{k^{3/2}}\sqrt{(p(1-p))^3} + \frac{\sqrt{3}}{k^{3/2}}\sqrt{(p(1-p))^3} + 3\sqrt{3}\frac{\sigma_p^2}{k} + \gamma\sigma_p^3\right)$$

$$\le 4\left(\frac{1}{k^{3/2}}\sqrt{(p(1-p))^3} + \frac{\sqrt{3}}{k^{3/2}}\sqrt{(p(1-p))^3} + 3\sqrt{3}\sigma_p^3 + \gamma\sigma_p^3\right)$$

$$\le 4(3\sqrt{3}+\gamma)\left(\frac{1}{k^{3/2}}\sqrt{(p(1-p))^3} + \sigma_p^3\right)$$

$$\le 4(3\sqrt{3}+\gamma)\left(\frac{1}{k}p(1-p) + \sigma_p^2\right)^{3/2}$$

$$\le 8(3\sqrt{3}+\gamma)\left(\frac{1}{k}p(1-p) + \frac{k-1}{k}\sigma_p^2\right)^{3/2},$$

where the first and second inequalities follow from above, the third inequality follows because $k \ge 1$, the fourth is simply rearranging the terms, the fifth follows from the fact that for all positive, real numbers $a$ and $b$: $a^{3/2} + b^{3/2} < (a+b)^{3/2}$, and the last inequality follows since if $k \ge 2$ then $(k-1)/k > 1/2$. $\qquad\square$

With this result, we can apply Lemma F.2 to our setting to privately achieve an estimate $\widehat{\sigma}_{p,k}^2$ that is close to the true population-level variance $\sigma_p^2$, as shown in Lemma F.4. Note that as $k$ grows large, the allowable range for $p$ approaches the full support $[0,1]$ and the allowable standard deviation $\sigma_p$ approaches any non-negative number.

Lemma F.4 combines these two results to show that Lemma F.2 can be applied to the individual reports $\widehat{p}_i^k$ from the top $\log n$ users, and the resulting variance estimate will satisfy the accuracy conditions of Theorem 4.1.

**Lemma F.4.** *Given $\sigma_{min} < \sigma_{max} \in [0,\infty], \epsilon > 0, \delta \in (0, \frac{1}{n}], \beta \in (0,1/2)$, and $\zeta > 0$, let $\mathcal{M}$ be the $(\epsilon,\delta)$-differentially private mechanism given by Lemma F.2, and let $\widehat{\sigma}_{p,k}^2 = \mathcal{M}(\widehat{p}_1^k, \cdots, \widehat{p}_{\log n}^k)$, where $\widehat{p}_1^k, \cdots, \widehat{p}_{\log n}^k \sim \mathcal{D}(k)$. If there exists $\zeta > 0$ such that $\frac{\rho_{\mathcal{D}}}{\sigma_p^3} \le \zeta$ where $\rho_{\mathcal{D}} = \mathbb{E}_{x\sim\mathcal{D}}[|x - p|^3]$, $\sqrt{\frac{1}{k}p(1-p) + \frac{k-1}{k}\sigma_p^2} \in [\sigma_{min}, \sigma_{max}]$, $\sigma_p > \frac{1}{k}$, $p \in \left[\frac{1}{k}, 1 - \frac{1}{k}\right]$, and $\log n \ge c(8(3\sqrt{3} + \zeta))^2 \min\{\frac{1}{\epsilon}\ln(\frac{\ln(\frac{\sigma_{max}}{\sigma_{min}})}{\beta}), \frac{1}{\epsilon}\ln(\frac{1}{\delta\beta})\}$, then with probability $1 - \beta$, $\widehat{\sigma}_{p,k}^2 \in [\mathrm{Var}(\mathcal{D}(k)), 8\mathrm{Var}(\mathcal{D}(k))]$.*

*Proof.* Note that the conditions are sufficient to ensure from Lemma F.3 that $\frac{\rho_{\mathcal{D}(k)}}{\mathrm{Var}(\mathcal{D}(k))^{3/2}} \le 8(3\sqrt{3} + \gamma)$. Then Lemma F.2 and Lemma 2.1 imply that

$$\mathrm{Var}(\mathcal{D}(k)) = \frac{1}{k}p(1-p) + \frac{k-1}{k}\sigma_p^2 \le \widehat{\sigma}_{p,k}^2 \le 8\left(\frac{1}{k}p(1-p) + \frac{k-1}{k}\sigma_p^2\right) = 8\mathrm{Var}(\mathcal{D}(k)).$$

$\qquad\square$

# G Interpretation and Estimation of Concentration Functions

Recall that $f_{\mathcal{D}}^{k_i}(n, \sigma_p^2, \beta)$ describes the concentration of $\widehat{p}_i \sim \mathcal{D}(k_i)$ and is defined as $f_{\mathcal{D}}^{k_i}(n, \sigma_p^2, \beta) = \arg\inf\{\alpha \mid \Pr_{\widehat{p}_1, \cdots, \widehat{p}_n \sim \mathcal{D}(k_i)}(\max_i |\widehat{p}_i - p| \ge \alpha) \le \beta\}$. In the main body of the paper, we assumed that this function was known to the analyst, even if the input value $\sigma_p^2$ was unknown and had to be estimated. In this appendix, we interpret the structure of this concentration function and show that even when this informational assumption is relaxed, our Algorithm 2 can still be implemented with some minor modifications.

We start by introducing two additional functions: $f_{\mathcal{D}}(n, \sigma_p^2, \beta)$, which describes the concentration of $p_i \sim \mathcal{D}$, and $f_{\text{Bin}}(k_i, p_i, \beta)$, which describes the high probability tail bound on the binomial $\text{Bin}(k_i, p_i)$:

$$f_{\mathcal{D}}(n, \sigma_p^2, \beta) = \arg\inf\{\alpha \mid \Pr_{p_1, \cdots, p_n \sim \mathcal{D}}(\max_i |p - p_i| \geq \alpha) \leq \beta\}.$$

$$f_{\text{Bin}}(k_i, p_i, \beta) = \arg\inf\{\alpha \mid \Pr_{x \sim \text{Bin}(k_i, p_i)}(|\frac{1}{k_i}x - p_i| \geq \alpha) \leq \beta\}$$

In this appendix, we will assume that only the function $f_{\mathcal{D}}(n, \cdot, \beta)$ is known to the analyst, but the input variance parameter $\sigma_p^2$ of the distribution are not known. For example, the analyst may know that $\mathcal{D}$ is Gaussian with unknown mean and variance, and thus she can express the concentration of $p_i$ as a function of the variance. Also note that for any values $k_i, p_i$ and $\beta$, we can empirically compute $f_{\text{Bin}}(k_i, p_i, \beta)$.

The following lemma shows how we can translate high probability bounds on $\mathcal{D}$ to high probability bounds on $\mathcal{D}(k)$, using this binomial tail bound of $\text{Bin}(k_i, p_i)$. Specifically, it shows that our quantity of interest $f_{\mathcal{D}}^{k_i}(n, \sigma_p^2, \beta)$ of the $\hat{p}_i$s can be upper and lower bounded by concentration of the $p_i$s (as described by $f_{\mathcal{D}}(n, \sigma_p^2, \beta)$) plus a binomial tail bound.

**Lemma G.1.** *Suppose that $\mathcal{D}$ is supported on $[0, 1/2]$. Given $k_i, n \in \mathbb{N}$, $\sigma_p^2$, and $\beta \in [0, 1]$, define $\beta' = 2\sqrt{1 - \sqrt[n]{1 - \beta}} = \Theta(\sqrt{\beta/n})$ and assume that for all $p_i$ in the support of $\mathcal{D}$,*

$$\Pr_{\hat{p}_i \sim \text{Bin}(k_i, p_i)}(p_i - \hat{p}_i \geq f_{\text{Bin}}(k_i, p_i, \beta')) \geq \frac{1}{2}\beta' \quad and \quad \Pr_{\hat{p}_i \sim \text{Bin}(k_i, p_i)}(\hat{p}_i - p_i \geq f_{\text{Bin}}(k_i, p_i, \beta')) \geq \frac{1}{4}\beta'.$$

*Then for all $\beta \in [0, 1]$, for all $i \in [n]$,*

$$f_{\mathcal{D}}^{k_i}(n, \sigma_p^2, \beta) \leq f_{\mathcal{D}}(n, \sigma_p^2, \beta/2) + f_{\text{Bin}}(k_i, p_{\max}, \beta/n),$$

*where $p_{\max} = \min\{1/2, p + f_{\mathcal{D}}(n, \sigma_p, \beta/2)\}$. Further, for all $i \in [n]$,*

$$f_{\mathcal{D}}^{k_i}(n, \sigma_p^2, \beta) \geq f_{\mathcal{D}}(1, \sigma_p^2, \beta') + f_{\text{Bin}}(k_i, p_{\max}, \beta').$$

We note that the conditions on $\mathcal{D}$ and $\text{Bin}(k_i, p_i)$ are mild. The condition on the tails of $\text{Bin}(k_i, p_i)$ is intuitively claiming that $\text{Bin}(k_i, p_i)$ is symmetric. This occurs whenever $k_i$ is large enough, and $p_i$ is bounded away from 0 or 1. We conjecture that the condition that $\mathcal{D}$ is supported on $[0, 1/2]$ can be relaxed but leave the relaxation to future work.

*Proof of Lemma G.1.* Notice that if $p < q < 1/2$ then $f_{\text{Bin}}(k_i, p, \beta) \leq f_{\text{Bin}}(k_i, q, \beta)$. Let us first consider the upper bound first. With probability $1 - \frac{\beta}{2}$,

$$\text{for all } i, |p - p_i| \leq f_{\mathcal{D}}(n, \sigma_p, \beta/2). \tag{25}$$

Further if Equation 25 holds then we have that with probability $1 - \frac{\beta}{2n}$,

$$|\hat{p}_i - p_i| \leq f_{\text{Bin}}(k_i, p_i, \frac{\beta}{2n}) \leq f_{\text{Bin}}(k_i, p_{\max}, \frac{\beta}{2n}).$$

Thus, for all $i$,

$$|p - p_i| \leq f_{\mathcal{D}}(n, \sigma_p, \beta/2) + f_{\text{Bin}}(k_i, p_{\max}, \frac{\beta}{2n}).$$

Now, for the lower bound, let $\beta' = \sqrt{8}\sqrt{1 - \sqrt[n]{1 - \beta}}$ and $\alpha = f_{\mathcal{D}}(1, \sigma_p^2, \beta')$. Note that either

$$\Pr_{p_i \sim \mathcal{D}}(p_i - p \geq f_{\mathcal{D}}(1, \sigma_p^2, \beta')) \geq \frac{1}{2}\beta' \quad \text{or} \quad \Pr_{p_i \sim \mathcal{D}}(p - p_i \geq f_{\mathcal{D}}(1, \sigma_p^2, \beta')) \geq \frac{1}{2}\beta'.$$

Assume without loss of generality that $\Pr_{p_i \sim \mathcal{D}}(p_i - p \geq f_{\mathcal{D}}(1, \sigma_p^2, \beta')) \geq \frac{1}{2}\beta'$. Then by assumption,

$$\Pr_{\hat{p}_i \sim \text{Bin}(k_i, p_i)}(\hat{p}_i - p_i \geq f_{\text{Bin}}(k_i, p_i, \beta')) \geq \frac{1}{4}\beta'$$

Then

$$\Pr\left(\max_i |\widehat{p}_i - p| \geq f_{\mathcal{D}}(1, \sigma_p^2, \beta') + f_{\text{Bin}}(k_i, p + \alpha, \beta')\right)$$
$$\geq \Pr\left(\exists i \text{ s.t. } p_i - p \geq f_{\mathcal{D}}(1, \sigma_p^2, \beta') \text{ and } \widehat{p}_i - p_i \geq f_{\text{Bin}}(k_i, p_i, \beta')\right)$$
$$= 1 - \Pr\left(\forall i, p_i - p \leq f_{\mathcal{D}}(1, \sigma_p^2, \beta') \text{ or } \widehat{p}_i - p_i \leq f_{\text{Bin}}(k_i, p + \alpha, \beta')\right)$$
$$= 1 - \left(\Pr\left(p_i - p \leq f_{\mathcal{D}}(1, \sigma_p^2, \beta') \text{ or } \widehat{p}_i - p_i \leq f_{\text{Bin}}(k_i, p + \alpha, \beta')\right)\right)^n.$$

Now,

$$\Pr\left(p_i - p \leq f_{\mathcal{D}}(1, \sigma_p^2, \beta') \text{ or } \widehat{p}_i - p_i \leq f_{\text{Bin}}(k_i, p + \alpha, \beta')\right)$$
$$= 1 - \Pr\left(p_i - p \geq f_{\mathcal{D}}(1, \sigma_p^2, \beta') \text{ and } \widehat{p}_i - p_i \geq f_{\text{Bin}}(k_i, p + \alpha, \beta')\right)$$
$$= 1 - \Pr\left(p_i - p \geq f_{\mathcal{D}}(1, \sigma_p^2, \beta')\right) \Pr\left(\widehat{p}_i - p_i \geq f_{\text{Bin}}(k_i, p + \alpha, \beta') \mid p_i - p \geq f_{\mathcal{D}}(1, \sigma_p^2, \beta')\right)$$
$$\leq 1 - \Pr\left(p_i - p \geq f_{\mathcal{D}}(1, \sigma_p^2, \beta')\right) \Pr\left(\widehat{p}_i - p_i \geq f_{\text{Bin}}(k_i, p_i, \beta') \mid p_i - p \geq f_{\mathcal{D}}(1, \sigma_p^2, \beta')\right)$$
$$\leq 1 - \Pr\left(p_i - p \geq f_{\mathcal{D}}(1, \sigma_p^2, \beta')\right) \Pr\left(\widehat{p}_i - p_i \geq f_{\text{Bin}}(k_i, p_i, \beta')\right)$$
$$\leq 1 - \frac{1}{8}(\beta')^2$$

where the first inequality comes from $p_i \geq p + \alpha$, so $f_{\text{Bin}}(k_i, p + \alpha, \beta') \leq f_{\text{Bin}}(k_i, p_i, \beta')$ So, finally,

$$\Pr\left(\max_i |\widehat{p}_i - p| \geq f_{\mathcal{D}}(1, \sigma_p^2, \beta') + f_{\text{Bin}}(k_i, p + \alpha, \beta')\right) \geq 1 - (1 - (\beta'/\sqrt{8})^2)^n = \beta,$$

which implies the result. □

## G.1 Extending Our Results to Unknown $f_{\mathcal{D}}^{k_i}(n, \sigma_p^2, \beta)$ settings

Lemma G.1 gives both upper bound and lower bounds on $f_{\mathcal{D}}^{k_i}(n, \sigma_p^2, \beta)$, which can be used to modify Algorithm 2 and extend Theorem 4.1 to apply in the setting where $f_{\mathcal{D}}^{k_i}(n, \sigma_p^2, \beta)$ is unknown, but $f_{\mathcal{D}}(n, \sigma_p^2, \beta)$ is known instead.

Recall that the concentration bound $f_{\mathcal{D}}^{k_i}(n, \sigma_p^2, \beta)$ is used in Algorithm 2 to define the truncation parameters $\widehat{a}_i$ and $\widehat{b}_i$, and that we would like to define a truncation window $[\widehat{a}_i, \widehat{b}_i]$ that both contains $[a_i, b_i]$ (so that with high probability none of the $\widehat{p}_i$ are truncated), and is not too wide, so $|\widehat{b}_i - \widehat{a}_i| \leq 6|b_i - a_i|$ (in order to invoke Lemma 4.2).

The following lemma proposes new values for $\widehat{a}_i$ and $\widehat{b}_i$ for the setting where only $f_{\mathcal{D}}(n, \sigma_p^2, \beta)$ is known, but not $f_{\mathcal{D}}^{k_i}(n, \sigma_p^2, \beta)$. It combines the bounds on $f_{\mathcal{D}}^{k_i}(n, \sigma_p^2, \beta)$ from Lemma G.1, with the bounds on $\widehat{p}^{initial}$ from Lemma F.1 to show that $|\widehat{b}_i - \widehat{a}_i| \leq 6|b_i - a_i|$, as desired.

**Lemma G.2.** *For $\alpha > 0$, let*

$$\widehat{a}_i = \max\left\{0, \widehat{p} - \alpha - f_{\mathcal{D}}(n, \widehat{\sigma_p^2}, \beta/2) - f_{\text{Bin}}(k_i, \widehat{p} + \alpha + f_{\mathcal{D}}(n, \widehat{\sigma_p}, \beta/2), \beta/n)\right\}$$

*and*

$$\widehat{b}_i = \min\left\{1, \widehat{p} + \alpha + f_{\mathcal{D}}(n, \widehat{\sigma_p^2}, \beta/2) + f_{\text{Bin}}(k_i, \widehat{p} + \alpha + f_{\mathcal{D}}(n, \widehat{\sigma_p^2}, \beta/2), \beta/n)\right\}.$$

*If $\widehat{\sigma_p^2} \geq \sigma_p^2$, and $|p - \widehat{p}| \leq \alpha$, then for all $i \in [n]$,*

$$[a_i, b_i] \subset [\widehat{a}_i, \widehat{b}_i].$$

*Further, if $\alpha \leq f_{\mathcal{D}}^{k_i}(n, \sigma_p^2, \beta)$ and $f_{\mathcal{D}}^{k_i}(n, \sigma_p^2, \beta) \geq \Omega(f_{\mathcal{D}}(n, \sigma_p^2, \beta) + f_{\text{Bin}}(k_i, \min\{1/2, p + f_{\mathcal{D}}(n, \sigma_p, \beta/2)\}, \beta/n))$ then*

$$|\widehat{b}_i - \widehat{a}_i| \leq 6|b_i - a_i|.$$

*Proof of Lemma G.2.* Let us first show that $[\widehat{a}_i, \widehat{b}_i] \subset [a_i, b_i]$. Using our modified definition of $\widehat{a}_i$ given above, we have,

$$
\begin{aligned}
\widehat{a}_i &= \widehat{p}_\epsilon^{\text{initial}} - \alpha - f_{\mathcal{D}}(n, \widehat{\sigma_p^2}, \beta/2) - f_{\text{Bin}}(k_i, \widehat{p}_\epsilon^{\text{initial}} + \alpha + f_{\mathcal{D}}(n, \widehat{\sigma_p^2}, \beta/2), \beta/n) \\
&\leq p - f_{\mathcal{D}}(n, \widehat{\sigma_p^2}, \beta/2) - f_{\text{Bin}}(k_i, p + f_{\mathcal{D}}(n, \widehat{\sigma_p^2}, \beta/2), \beta/n) \\
&\leq p - f_{\mathcal{D}}(n, \sigma_p^2, \beta/2) - f_{\text{Bin}}(k_i, p + f_{\mathcal{D}}(n, \sigma_p, \beta/2), \beta/n) \\
&\leq p - f_{\mathcal{D}}^{k_i}(n, \sigma_p^2, \beta) \\
&= a_i.
\end{aligned}
$$

The first two inequalities respectively follow from the accuracy conditions on $\texttt{mean}_{\epsilon,\delta}$ and $\texttt{variance}_{\epsilon,\delta}$ in Theorem 4.1, the third inequality comes from Lemma G.1, and the final equality is by the definition of $a_i$. A symmetric result that $\widehat{b}_i \geq b_i$ follows similarly.

The second statement of this lemma ensures that the width of the truncation parameter is not more than a constant factor larger than the ideal. Specifically,

$$
\begin{aligned}
|\widehat{b}_i - \widehat{a}_i| &\leq 2\alpha + 2\left( f_{\mathcal{D}}(n, \widehat{\sigma_p^2}, \beta/2) + f_{\text{Bin}}(k_i, \widehat{p} + \alpha + f_{\mathcal{D}}(n, \widehat{\sigma_p^2}, \beta/2), \beta/n) \right) \\
&\leq 2f_{\mathcal{D}}^{k_i}(n, \sigma_p^2, \beta) + O(f_{\mathcal{D}}(1, \sigma_p^2, \beta') - f_{\text{Bin}}(k_i, p + \alpha, \beta')) \\
&\leq 2f_{\mathcal{D}}^{k_i}(n, \sigma_p^2, \beta) + 2\left( 2f_{\mathcal{D}}^{k_i}(n, \sigma_p^2, \beta) \right) \\
&\leq 6f_{\mathcal{D}}^{k_i}(n, \sigma_p^2, \beta) \\
&= 6|b_i - a_i|
\end{aligned}
$$

$\square$

We note that Lemma 4.2 as stated requires $|\widehat{b}_i - \widehat{a}_i| \leq 4|b_i - a_i|$, rather than $6|b_i - a_i|$, this difference of constants will only affect the constant $C$ in Theorem 4.1, and the main claim of a constant approximation in variance will still hold with these new $\hat{a}_i$ and $\hat{b}_i$ values.

We will, however, have to add an additional assumption to Theorem 4.1 in this setting. We will need to assume that $\mathcal{D}$ is s.t. $f_{\mathcal{D}}^{k_i}(n, \sigma_p^2, \beta) \geq \Omega(f_{\mathcal{D}}(n, \sigma_p^2, \beta) + f_{\text{Bin}}(k_i, \min\{1/2, p + f_{\mathcal{D}}(n, \sigma_p, \beta/2)\}, \beta/n))$, to satisfy the condition of Lemma G.2. This condition is related to the high probability bound on $\mathcal{D}(k)$. The right hand side of this condition is the high probability bound on $\mathcal{D}(k)$ that is inherited directly from the high probability bounds on $\mathcal{D}$ and $\text{Bin}(k, p)$. Without further assumptions on $\mathcal{D}$, this is the best upper bound on $f_{\mathcal{D}}^{k_i}(n, \sigma_p^2, \beta)$ that we can obtain, and hence is the bound used in the truncation in $\widehat{p}_\epsilon^{\text{realistic}}$. The condition states that this upper bound is within a constant multiplicative factor of the true value $f_{\mathcal{D}}^{k_i}(n, \sigma_p^2, \beta)$. We note that this condition is guaranteed by the lower bound on $f_{\mathcal{D}}^{k_i}(n, \sigma_p^2, \beta)$ in Lemma G.1 for $\mathcal{D}$ with support on $[0, 1/2]$, and we conjecture that it holds more broadly.