# OpenReview forum: "Mean Estimation with User-level Privacy under Data Heterogeneity"
_NeurIPS.cc/2022/Conference — NeurIPS 2022 Accept_

### Official Review · Reviewer_mJEC · 2022-07-11

**Rating:** 6
**Confidence:** 2
**Soundness:** 2 fair
**Presentation:** 2 fair
**Contribution:** 2 fair

**Summary:**

The paper proposes an approach to estimate the population-level mean in a heterogenous data setting without compromising user privacy. The core objective is to solve the global task for utility with better privacy for local data of each user.  The main result is that a non-private optimal estimator is linear while private estimator has to be non-linear.

**Questions:**

Provide an example where the exact proposed algorithm satisfying the assumptions will be used in any application.

**Limitations:**

There is no section clearly identifying or discussing the limitations of the paper and ways to address them. The required assumptions that can be seen as limitations if not satisfied are spread throughout the paper making it difficult to understand their impact on the results.

**Strengths And Weaknesses:**


- The paper does not provide sufficient motivation or real-world scenarios where private estimating the mean will be useful in a heterogenous setting. The example of clicking on ads satisfies the assumption of Bernoulli distribution but the importance of estimating the mean privately in such a setting is not clear.

- The paper is a purely theoretical paper and does not provide experiments to demonstrate the effectiveness of the proposed algorithm for real datasets. The empirical privacy-utility trade-off is an important factor to understand for any new differentially private algorithm to evaluate their impact.

- The presented results are limited to binary problems but the abstract and intro motivates heterogeneity in complex data types such as speech and language data.

- There are many assumptions made in the paper. It is not quite clear if these assumptions are rational and hold true in practice or are only considered for better theoretical analysis but may not be possible in reality. Please state all the assumptions upfront and their feasibility clearly.

---

> ### Author Response · Authors · 2022-08-02
> **Response**
>
> We thank the reviewer for their thoughtful review and comments.
>
> We focus on mean estimation as we believe that mean estimation is the most fundamental statistical task, and is the starting point for more complex tasks such as high-dimensional mean estimation (used to compute gradients when training ML models) and histogram computation.
>
> The ads example is one setting where privacy has become increasingly important in recent times, e.g. the W3C’s Private Advertising Technology Community Group (https://patcg.github.io) is working on the question of designing an advertising ecosystem which is more respectful of user privacy.
>
> There are many features where the user generates Boolean signals by engaging or not engaging with a feature (e.g. phone keyboard next word suggestion, crisis helpline link, search engine knowledge panels, sponsored link on yelp or amazon, etc.). As a concrete example, a language model is used to make the next word suggestions on a phone keyboard. A new version of this model would be first tested to measure the average over users of the suggestion acceptance rate. From each user, we would thus get a set of independent Bernoulli r.v.’s with each individual mean p_i corresponding to the model accuracy for the user. Heterogeneity comes from different users typing differently (and hence model accuracy varying for different users) and using the keyboard with different frequency. It would also be reasonable to assume that a general model has similar accuracy for the vast majority of users, or formally, that the model accuracy is well-concentrated. Note the distribution of model accuracies among users is the meta distribution D in our work. Such distributions are typically modeled as (sub)Gaussian. More generally, measuring the average accuracy of a classification model among a large group of users is an important task in itself. Such models are deployed in privacy-sensitive applications such as health and finance. The resulting statistics may need to be shared with third parties or other teams within a company raising potential user privacy concerns. We believe that the algorithm given in our work is a good starting point for solving this problem in practice. We agree that an empirical evaluation of this algorithm would be valuable, but due to the complexity of finding realistic heterogeneous public datasets we leave this evaluation for future work.
>
> Assumptions: We state our results in a very general setting, with the goal of assuming as little of the meta-distribution as possible. This has the side effect of having to state a few different assumptions. We believe our assumptions are fairly weak, and are satisfied, e.g. when the meta-distribution is a (truncated) Gaussian and the distribution of number of examples per-user is not absurdly extreme (i.e. log n users should not contribute more than half of the examples). We will add a better discussion of these assumptions and limitations.

---

> > ### Comment · Reviewer_mJEC · 2022-08-08
> > **Example is helpful!**
> >
> > Thank you for the clear example and explanation about the assumptions. I have increased my score to a weak accept as it would still be helpful to see empirical results on may be a synthetic dataset to demonstrate the practicality of the algorithm. It will be useful to provide a better discussion of the assumptions and limitations as mentioned in the rebuttal.

---

### Official Review · Reviewer_8HMZ · 2022-07-11

**Rating:** 7
**Confidence:** 2
**Soundness:** 3 good
**Presentation:** 2 fair
**Contribution:** 3 good

**Summary:**

This paper proposes a method to estimate the population level mean under user-level differential privacy constraints when the user data is different in both distribution and quantity for different users. The main contribution is to build a differentially private method that estimates the mean and variance of the data distribution in a setting where there is an unknown global meta-distribution D and for each user i, a personal data distribution D_i. There is an additional assumption that D_i is simply a Bernoulli distribution.



**Questions:**

Could authors explain why the given example (next-word prediction for a keyboard.) is important for privacy perspective? What other applications can be given as example? I'm curious because since there is no empirical evaluation of the proposed method, it is hard to imagine how the proposed approach is applied for different problems.

**Limitations:**

Between lines 250-253, the authors discuss one limitations of the proposed method. The issue issue when n is not large enough (that is not specified), the log n users with the most data can not estimate the mean of meta-distribution alone. Up to some logarithmic factors, this condition simply requires that the number of data points held by the user with the most data is at most n times the number of data points of the median user. To me, the n could be specified for a better understanding and it could be discussed how that issue can be soled.


**Strengths And Weaknesses:**

Strengths: The problem defined in paper where there is distributed datasets from different distributions is an important problem that needs to be addressed. The paper theoretically sounds and mostly well written.

Weaknesses: In my opinion, the discussion in the paper needs to be extended. It is not so clear to me if this problem has been studied before and what are the differences of the proposed approach. The application field can be discussed more broadly as well. In addition to that, the utility analysis of the proposed method is not provided and I'd be curious to see that.

---

> ### Author Response · Authors · 2022-08-02
> **Response**
>
> We thank the reviewer for their thoughtful review and comments.
>
> Beyond next word prediction our setting applies to measuring the average accuracy of a classification model deployed to a large group of users. Such models are deployed in privacy-sensitive applications such as health and finance. The resulting statistics may need to be shared with third parties or other teams within a company raising potential user privacy concerns. More generally, measuring the average accuracy of a classification model among a large group of users is an important task that fits into the model we describe. For more discussion see response to reviewer mJEC.
>
> As we discuss in related work, there is little work on this question in the private setting [21,22]. To our knowledge, ours is the first work in this model that we study (with heterogeneity in both the number of data points per user, and the distribution of each user's data points).
>
> Our statements emphasize the fact that our estimator achieves near optimal utility. There is no simple formula for the variance of our estimator since it is a minimum over a choice of a threshold of the variance expression in equation (5). We will make the utility bound more explicit in the revision. Corollary 5.5 does give some intuition for how the variance depends on the distribution of the k_i.
>
> We will further clarify the discussion on lines 250-253 as the reviewer suggests. Essentially, the assumption is that we are not in the regime where the data of the top log n users (those with the largest k_i’s) gives a more accurate estimate of the mean than the data of all the rest of users. In the setting where these log n users can give a very accurate estimate of the mean, we conjecture that there is little benefit in incorporating the data of the remaining users. We believe this is unlikely to be a limiting condition in practice.

---

> > ### Comment · Reviewer_8HMZ · 2022-08-08
> > **Response to the rebuttal**
> >
> > Thank you for clarifying my concerns. I also read other reviews and responses to them. I increased my score accordingly.

---

### Official Review · Reviewer_mEcE · 2022-07-12

**Rating:** 7
**Confidence:** 3
**Soundness:** 4 excellent
**Presentation:** 3 good
**Contribution:** 3 good

**Summary:**

This work studies user-level differentially private mean estimation with heterogeneous user data. The problem is as follows: each user receives $k_i$ i.i.d. samples from $Ber(p_i)$, and $p_i$ is drawn from some global distribution with mean $p$ and variance $\sigma_p^2$. The authors design an unbiased estimator which achieves the near-optimal min-max rate among all unbiased algorithms. The algorithm can be extended to higher dimensions and private $k_i$'s.

**Questions:**

Questions
- Regarding unbiased estimators: see the first item in the weakness section.
- Is it possible to remove the restriction $p\in[1/3, 2/3]$ in the lower bounds? Do you think the proposed algorithm is min-max optimal for $p\in[0, 1]$?
- Could the authors elaborate a little bit on the extension to higher dimensions and the corresponding guarantees? Say the dimension is $d$, and $p_i\in R^d$, there might be two possible generalizations of the original problem formulation: 1. each coordinate of $p_i$ is independent 2. $p_i$ is drawn from a distribution over the $d$-dimensional probability simplex. I assume that we could simply algorithm the algorithm independently to each coordinate, but what would be the guarantee and optimality in these two cases?

Typos
- Line 199 and 310, $\sigma_i$ should be $\sigma_i^2$?

**Limitations:**

Limitations and potential negative societal impact are adequately addressed.

**Strengths And Weaknesses:**

Strengths
- Understanding user-level privacy with heterogenous user data is a highly important problem in federated learning in both theory and practice. The problem considered in the paper is highly relevant to the FL and DP community.
- The problem formulation is general enough and should be relevant in many applications.
- The unbiased algorithm is nearly optimal among all unbiased estimators.
- The writing is clear and well structured.

Weakness
- The authors mainly consider unbiased estimators. Is it possible that biased estimators achieve better performance than unbiased ones? For example, in the homogenous setting where all $k_i=k$,  $p_i=p$ and $p<1/k$, the algorithm in [22] is biased (but probably not by too much) but min-max optimal up to log factors.
- The lower bound result seems to be restricted for $p\in[1/3, 2/3]$ (i.e., not to close to endpoints)

---

> ### Author Response · Authors · 2022-08-02
> **Response**
>
> We thank the reviewer for their positive assessment of the paper, and helpful comments.
>
> Unbiased estimators: The private estimators that we arrive at have a small amount of bias due to the truncation of the data point, although this bias can be made polynomially small (since the truncation is set so that the data point is not truncated with high probability). Our lower bound can also be extended to include estimators with polynomially small bias. The unbiased (or small bias) assumption is inherent in the techniques we use, and to the best of our knowledge is also a required assumption for Cramer-Rao style proofs in the non-private setting.
>
> Assumption on p^* \in [⅓, ⅔]. This is a great question. In our proof, this assumption is primarily used to relate the variance of Ber(p^*) to that of Ber(p_i) where p_i is drawn from the meta distribution. As p^* gets close to one of the end points, one can imagine meta distributions where typically p_i is a lot smaller, so that on average the variance of Ber(p_i) is a lot less than that of Ber(p^*). While our techniques would extend to a larger range of p^* under additional assumptions (e.g. median = p^*), we leave to future work the question of establishing minimax optimality more generally.
>
> The natural high dimensional generalization of our problem is frequency estimation of k elements. As the reviewer observes, the algorithm we have, when applied to each coordinate/element, will give us some upper bounds in the high-dimensional case. We do not know if the results are tight in high dimensions, and expect that the simplex constraint will help in the high-dimensional case as projection of noisy measurements provably reduces the squared error, potentially at the cost of bias. We leave a deeper exploration of this question to future work.

---

### Official Review · Reviewer_Tq3t · 2022-07-17

**Rating:** 7
**Confidence:** 4
**Soundness:** 3 good
**Presentation:** 2 fair
**Contribution:** 3 good

**Summary:**

This paper considers the mean estimation under use-level differential privacy. Specifically, each user data is sampled from the Bernoulli distribution with a different mean p_i and p_i is sampled from a meta-distribution with unknown variance. The goal is to estimate the mean of the meta-distribution with minimal variance. In the ideal setting where the variance of each user's Bernoulli distribution is known, the optimal estimator is given by the average of each user's sample mean weighted by the inverse variance of each user. In the realistic setting, the authors propose an algorithm to estimate the variance of each user first and then apply the inverse variance method.  In the private setting, the authors assume there exists a known concentration function of the meta-distribution (variance is still unknown). To lower the sensitivity, the authors propose to use this concentration function to truncate the sample mean of each user and use some threshold to truncate the weight of each user too. Finally, the authors show that, for sub-Gaussians, the proposed estimator is nearly optimal up to logarithmic factors and does not have extra privacy costs if epsilon is a constant.

**Questions:**

1. In Theorem 4.1, what would break if the third assumption does not hold? Is this necessary for achieving optimal rates?
2. Is there an upper bound on the estimation rate for sub-Gaussian meta distribution?
3. In the initial variance estimator, it seems that we need to know the \sigma_min, \sigma_max, and third moment as prior information. Is it possible to avoid this? Can you use the concentration function to help you estimate the variance?
4. I understand the non-private estimator is unbiased. But in the private setting, the truncation would cause some bias. What is the bias and how is it compared to the variance?

**Limitations:**

This paper is theoretical and does not have a direct negative societal impact.



**Strengths And Weaknesses:**

Strengths: 1. The problem setting is interesting and well motivated by giving examples.
2. The presentation of this paper is clear. The authors start by giving an algorithm under a non-private and ideal setting and then progressively modify the algorithm for the realistic and private setting. I enjoyed reading this paper.

Weaknesses: 1. It would be better if the authors could highlight some theoretical results or theorems in the introduction section including the assumptions.

---

> ### Author Response · Authors · 2022-08-02
> **Response**
>
> We thank the reviewer for their positive assessment of the paper, and helpful suggestions. Below we address the reviewer’s questions.
>
> In Theorem 4.1, the third assumption implies that we are not in the regime where the data of the top log n users (those with the largest k_i’s) gives a more accurate estimate of the mean than the data of all the rest of users. In the setting where these log n users can give a very accurate estimate of the mean, we conjecture that there is little benefit in incorporating the data of the remaining users. If this assumption does not hold then an estimator that utilizes the top log n users better may be optimal. We believe this assumption is likely to hold in practice.
>
> The variance of our estimator is dependent on the k_i’s and hence it is difficult to write an explicit formula. For a truncated Gaussian meta distribution, the formula in Corollary 5.5 on page 9 of the main submission will be tight and would thus be the variance of our estimator.
>
> Bounds on sigma, and the third moment are required for the initial estimator we describe.
> The dependence on sigma_min and sigma_max is very mild (i.e. log (sigma_max/sigma_min)) so we can set this interval to be very large without much impact on the performance, e.g. sigma_max=1 and sigma_min = 1/poly n.
> We need a bound on the third moment to sigma^3 ratio as well. Such a bound is trivially true for Gaussians, and for most distributions of interest. Subgaussian tails for example will imply a bound on this ratio, which would suffice. We will add more details in the final version. We also note that any initial estimator with appropriate accuracy guarantees can be used, so improvements in variance estimators (in the standard non-heterogeneous model) will immediately imply improvements in our algorithm.
>
> The private estimators that we arrive at have a small amount of bias due to the truncation of the data point, although this bias can be made polynomially small (since the truncation is set so that the data point is not truncated with high probability). The bias can be made very small compared to the variance.

---

> > ### Comment · Reviewer_Tq3t · 2022-08-08
> > **Thanks for your response**
> >
> > I have read the other reviewers' comments and the authors' responses. Thanks to the authors for their feedback. All of my questions are addressed.

---

### Meta-Review · Area_Chair_saGx · 2022-08-21

**Recommendation:** Accept
**Confidence:** Certain

**Metareview:**

Addressing heterogeneity in differentially private aggregation and estimation is an practically important topic that shows up in many real world settings. This paper makes the first step in closing the gap between existing sophisticated algorithms that work under homogeneous setting and the practical scenarios with heterogeneous users. The reviewers agree that this is an important paper that opens up several exciting research directions.

**Award:**

No

---

### Decision · Program_Chairs · 2022-09-14

Accept